# MME-Reasoning: A Broad-Spectrum Benchmark for Evaluating Logical Reasoning in MLLMs

Jiakang Yuan [* 1 2]   Tianshuo Peng [* 3 2]   Yilei Jiang [3]   Yiting Lu [4]   Renrui Zhang [3]   Kaituo Feng [3]   Chaoyou Fu [5]
Tao Chen [1]   Lei Bai [2]   Bo Zhang [2]

## Abstract

Logical reasoning is a fundamental aspect of human intelligence and an essential capability for multimodal large language models (MLLMs). Despite the significant advancement in multimodal reasoning, existing benchmarks fail to comprehensively evaluate their reasoning abilities due to the lack of explicit categorization for logical reasoning types and an unclear understanding of reasoning. To address these issues, we introduce MME-Reasoning, a comprehensive benchmark designed to evaluate the reasoning ability of MLLMs, which covers all three types of reasoning (i.e., inductive, deductive, and abductive). We carefully curate the data to ensure that each question effectively evaluates reasoning ability rather than perceptual skills or knowledge breadth, and extend the evaluation protocols to cover the evaluation of diverse questions. Our evaluation reveals substantial limitations of SoTA MLLMs when subjected to holistic assessments of logical reasoning capabilities. Even the most advanced MLLMs show limited performance in comprehensive logical reasoning, with notable performance imbalances across reasoning types. In addition, we conducted an in-depth analysis of approaches such as "thinking mode" and Rule-based RL, which are commonly believed to enhance reasoning abilities. We hope the community can pay more attention to the comprehensive reasoning capabilities of MLLMs instead of only focusing on its subset.

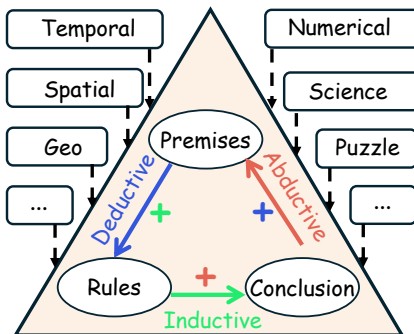

*Figure 1.* Reasoning Types.

*Table 1.* Comparison with other benchmarks.

| Benchmark | Size | Question Types | Reasoning Types |
|---|---|---|---|
| MathVista | 6.1K | MCQ, Free-form | Ind., Ded. |
| LogicVista | 0.4K | MCQ | Ind., Ded. |
| VisualPuzzles | 1.2K | MCQ | Ind. |
| VisuLogic | 1K | MCQ | Ind. |
| **MME-Reasoning** | 1.2K | MCQ, Free-form Rule-based | Ind., Ded., Abd. |

## 1. Introduction

Logical reasoning ([Liu et al., 2025a](#)), a fundamental cognitive process of analyzing premises and evidence to reach valid conclusions, serves as the cornerstone of human intelligence. Multimodal reasoning ([Jaech et al., 2024](#)) enables humans to integrate information from different modalities, such as visual and text, which is essential for tackling complex tasks. Recently, with the emergence of reasoning large language models (LLMs) ([Dubey et al., 2024](#)) such as DeepSeek-R1 ([DeepSeek-AI, 2025](#)), injecting reasoning capability into MLLMs ([OpenAI, 2024](#)) has begun to be explored ([Peng et al., 2025b](#)). Despite the significant progress in reasoning MLLMs, a comprehensive evaluation of their capabilities still remains an open challenge. Therefore, it is important to establish a comprehensive evaluation benchmark to assess the reasoning capabilities of MLLMs and further accelerate the development of this field.

Currently, most benchmarks ([Fu et al., 2023](#); [Wang et al., 2024a](#); [Lu et al., 2023](#); [Yue et al., 2024a](#);b; [Gong et al., 2024](#); [He et al., 2024](#)) designed for multimodal reasoning

---
[*]Equal contribution   [1]Fudan University, Shanghai, China   [2]Shanghai Artificial Intelligence Laboratory, Shanghai, China   [3]The Chinese University of Hong Kong, Hong Kong, China   [4]University of Science and Technology of China   [5]Nanjing University. Correspondence to: Tao Chen <eetchen@fudan.edu.cn>, Bo Zhang <zhangbo@pjlab.org.cn>.

*Proceedings of the $43^{rd}$ International Conference on Machine Learning*, Seoul, South Korea. PMLR 306, 2026. Copyright 2026 by the author(s).

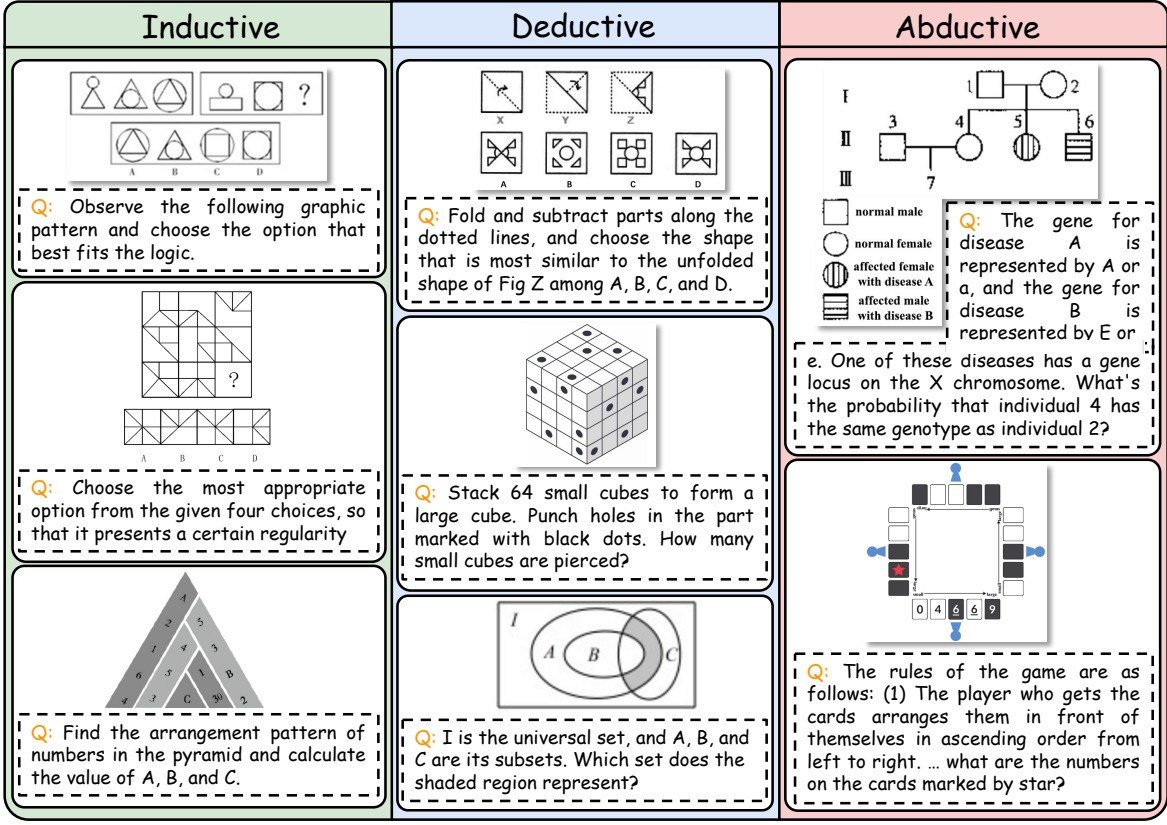

*Figure 2.* Example of questions in MME-Reasoning which covers comprehensive reasoning types.

primarily focus on knowledge-driven tasks. For example, MathVista (Lu et al., 2023) and MathVerse (Zhang et al., 2024) provide comprehensive evaluations of MLLMs' mathematical reasoning abilities. OlympiadBench (He et al., 2024) and EMMA (Hao et al., 2025) expand the scope to include additional subjects, such as physics and chemistry. Apart from knowledge-driven tasks, some works (Chia et al., 2024; Zhang et al., 2025c) have begun to decouple knowledge from reasoning, aiming to assess the reasoning abilities of MLLMs independent of domain knowledge. For instance, VisualPuzzles (Song et al., 2025) focus on reasoning-focused, knowledge-light tasks.

Despite recent advances, existing benchmarks still suffer from several problems as outlined below, which limits a more comprehensive evaluation of multimodal reasoning. *(1) Lacking explicit categorization of reasoning and insufficient coverage of reasoning types.* Logical reasoning is typically classified into three types: *abduction, deduction, and induction* (Peirce, 2014). Most existing benchmarks primarily concentrate on evaluating MLLMs' inductive and deductive reasoning ability. For example, most of the questions in MathVerse (Lu et al., 2023) belong to deductive reasoning, which uses rules and premises to derive conclusions. PuzzleVQA (Chia et al., 2024) only contains

questions of inductive reasoning, which learns rules based on premises and conclusions. However, abductive reasoning ability (*i.e.*, exploring premises to explain a conclusion based on the conclusion and rules) is rarely evaluated. *(2) The concept of reasoning is not clear enough,* which is reflected in confusing perception with reasoning or equating reasoning with the complexity of the required knowledge. For example, MathVista (Lu et al., 2023) contains many questions that can be answered through visual perception, while OlympiadBench (He et al., 2024) includes questions that require advanced domain knowledge, which the model may not have access to. This may lead to an inaccurate evaluation of MLLMs' reasoning ability.

To address these and construct a comprehensive benchmark for multimodal reasoning, we introduce MME-Reasoning, a comprehensive benchmark specifically designed to evaluate the reasoning capability of MLLMs. MME-Reasoning consists of 1,188 carefully curated questions that systematically cover types of logical reasoning (*i.e.*, inductive, deductive, and abductive), while spanning a range of difficulty levels, as illustrated in Fig. 2. Besides, we identify 5 key abilities related to multimodal reasoning, including calculation, planning and exploring, spatial-temporal, pattern analysis, and casual chaining analysis, and annotate the

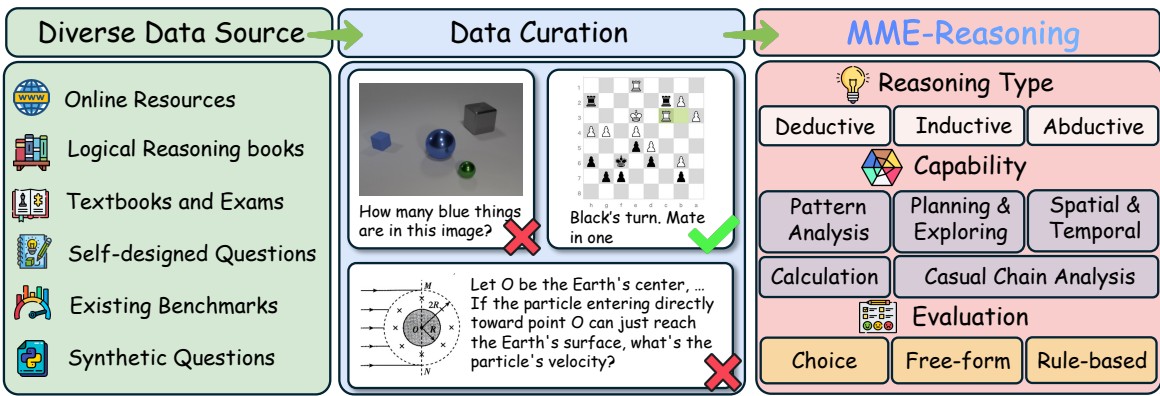

*Figure 3.* The overall construction process of MME-Reasoning.

type of ability assessed by each question. To ensure a true evaluation of reasoning ability, MME-Reasoning eliminates questions that can be answered purely through perception or require complex domain knowledge, thereby focusing on the core reasoning skills of the model. Furthermore, MME-Reasoning incorporates a variety of evaluation methods, including multiple-choice, free-form, and rule-based (*e.g.*, Sudoku Puzzles) questions. Employing multiple evaluation methods enables a wider variety of question types, thereby facilitating a more comprehensive evaluation of models' capabilities. Experiments on diverse MLLMs (*i.e.*, chat/-thinking, open/closed-source) reveal several key findings:

- **MLLMs exhibit significant limitations and pronounced imbalances in reasoning capabilities.** Even the most advanced MLLMs achieve only limited results under holistic logical reasoning evaluation, with Gemini-Pro-2.5-Thinking scoring only 60.19%, followed by Seed1.5-VL (59.85). These results indicate that MME-Reasoning, through its comprehensive evaluation of all the logical reasoning types, establishes a systematic and challenging benchmark for multimodal reasoning.

- **Abductive reasoning remains a major bottleneck for current MLLMs.** While most models demonstrate competent deductive reasoning, their abductive reasoning lags significantly. Closed-source models exhibit an average gap of 5.38 points between deductive and abductive tasks, which further widens to 9.81 among open-source models. Since it underpins many real-world tasks, addressing this gap is crucial for improving overall reasoning.

- **Reasoning length scales with task difficulty, benefiting performance but accompanied by marginal effects and decreasing token efficiency.** Thinking Models exhibit longer reasoning chains, particularly on more difficult questions, demonstrating adaptive inference budgeting and enhanced depth of reasoning. A positive correlation between average token count (ATC) and accuracy supports the effectiveness of extended outputs, especially in

complex tasks. However, this performance gain plateaus beyond a certain length, revealing diminishing returns.

## 2. Related Works

### 2.1. Multimodal Reasoning Benchmarks

Early works such as CLEVR (Johnson et al., 2016) assess compositional and spatial reasoning, while more recent benchmarks (Lu et al., 2024; Zhang et al., 2025c; Chia et al., 2024) focus on symbolic logic reasoning. EMMA (Hao et al., 2025) focuses on ensuring that questions cannot be answered by LLMs using only the image caption. VisualPuzzles (Hu et al., 2024), MM-IQ (Cai et al., 2025), and VisuLogic (Xu et al., 2025) try to evaluate MLLMs' reasoning capabilities through puzzle-solving. Rbench-V (Guo et al., 2025b) is designed to evaluate models' capability of MCoT (*i.e.*, whether a model can perform reasoning visually). However, these benchmarks typically focus on narrow subtypes and fail to offer a comprehensive evaluation across deductive, inductive, and abductive paradigms. To evaluate the reasoning capability, some benchmarks (Rein et al., 2023; He et al., 2024; Jiang et al., 2025) improve the complexity of questions. However, most of them overly depend on domain-specific knowledge rather than logical inference. Evaluation protocols are also limited—most rely on multiple-choice formats and lack support for open-ended or rule-based assessment. In contrast, our benchmark provides a fine-grained evaluation of visual reasoning, explicitly covering the three classical reasoning types.

### 2.2. Multimodal Reasoning

Chain-of-thought (CoT) reasoning (Wei et al., 2022) has emerged as a key paradigm for enhancing the reasoning capability of LLMs, enabling more transparent and accurate decision-making in complex tasks. Inspired by its success in text-only settings, CoT has recently been extended to MLLMs, giving rise to multimodal chain-of-thought (MCoT) reasoning (Jiang et al., 2024; Zhang et al., 2023;

*Table 2.* Statistics of MME-Reasoning.

| Statistics | Number |
|---|---|
| **Total** | 1188 (100%) |
| - Newly-add questions | 84.85% |
| - Sampled questions | 15.15% |
| **Question Type** | |
| - Multi-choice questions | 58.50% |
| - Free-form questions | 31.57% |
| - Rule-based questions | 9.93% |
| **Image Type** | |
| - Single-image questions | 58.50% |
| - Multi-image questions | 31.57% |
| **Disciplinary** | |
| - Disciplinary questions | 31.48% |
| - Non-discipl. questions | 68.52% |

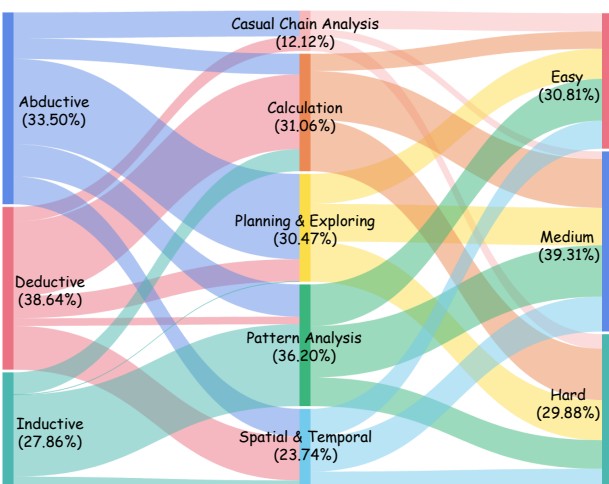

*Figure 4.* Overview of MME-Reasoning.

Chen et al., 2023; Peng et al., 2024; Lu et al., 2025; Xia et al., 2024). Early approaches such as Multimodal-CoT (Zhang et al., 2023) and IPVR (Chen et al., 2023) demonstrate that generating intermediate reasoning steps significantly improves model performance in visual question answering. Other methods (Yao et al., 2023; Zheng et al., 2024; Hu et al., 2024) introduce graph structures, debating agents, and visual intermediate states to further enhance interpretability and reasoning depth. More recently, following the success of Deepseek-R1, the Group Relative Policy Optimization (GRPO) algorithm has gained traction in the development of multimodal models. Several works (Meng et al., 2025; Zhou et al., 2025; Yingzhe et al., 2025; Chen et al., 2025; Feng et al., 2025) adapt GRPO to solve mathematical tasks, demonstrating promising reasoning capabilities. Other works (Shen et al., 2025; Liu et al., 2025b) apply GRPO to enhance visual competencies such as grounding.

## 3. The MME-Reasoning Benchmark

We introduce MME-Reasoning, a comprehensive benchmark for evaluating the reasoning ability of MLLMs. MME-Reasoning consists of 1,188 questions (1,008 newly collected items). MME-Reasoning covers three types of reasoning (*i.e.*, inductive, deductive, abductive) and includes three question types (*i.e.*, MCQ, free-form, rule-based). We further divided MME-Reasoning into three difficulty levels (*i.e.*, easy, medium, hard). The key statistics are shown in Tab. 2 and the construction pipeline is shown in Fig. 3.

### 3.1. Design Principles of MME-Reasoning

To ensure a comprehensive evaluation of multimodal reasoning, MME-Reasoning is guided by the following principles: *1) Comprehensiveness.* According to *Charles Sanders Peirce's classification of reasoning*, deduction, induction, and abduction can be distinguished based on different ar-

rangements of rule, case, and result. Therefore, a comprehensive evaluation of reasoning ability should include all three types of reasoning tasks. *2) Beyond Perception.* Each question should be carefully designed to ensure that the answer is obtained through a reasoning process instead of simple visual recognition. *3) Minimizing Knowledge Reliance.* It is essential to ensure that the questions do not require complex domain knowledge, thereby preventing models from being penalized for the absence of specialized information. In MME-Reasoning, the domain expertise is limited to K12 or below. *4) Diverse evaluation formats.* The benchmark should consist of diverse question types, avoiding incomplete evaluation caused by a narrow range of task types.

### 3.2. Data Collection and Curation

**Data Collection.** We collect questions related to multimodal reasoning from a variety of sources, including *1) Textbooks* can provide subject exam questions (*e.g.*, mathematics, physics, chemistry, and biology). *2) Online resources, books on logical practice, and Chinese Civil Service Examination (Logic Test)* primarily includes IQ test questions, logic games (*e.g.*, Mate-in-one), and other tasks highly related to logical reasoning. *3) Synthetically generated questions.* Some visual reasoning problems, such as Number Bridge and Sudoku, can be generated based on specific rules. We develop code to generate a wide variety of such logic puzzles, covering different types and a range of difficulty levels. *4) Questions from existing benchmarks.* We sample 80 questions from PuzzleVQA (Chia et al., 2024) and 100 questions from MMIQ (Cai et al., 2025), excluding questions based on shape size identification which can be directly answered based on visual perception. *5) Self-designed questions.* We mainly construct questions related to spatial and temporal reasoning. The spatial reasoning questions involve relative spatial relationships and naviga-

tion tasks, with the question design methodology inspired by VSIBench (Yang et al., 2024). For temporal reasoning, the questions mainly focus on sequence judgment. We sample frames from videos in YouCook2 (Zhou et al., 2018) and VideoMME (Fu et al., 2024) as the sources of images. Note that for questions with well-defined rules such as Maze, we include the corresponding rules as part of each question. The composition of MME-Reasoning is shown in Fig. 4, and more details are listed in the Appendix.

**Data Curation.** We initially collect around 4k questions from various sources mentioned above. Following the design principles of MME-Reasoning, we conduct a careful manual curation process to ensure the quality of the benchmark. Specifically, we exclude questions that depend solely on visual recognition, require complex domain-specific knowledge, too easy to evaluate the reasoning ability. This curation process ensures that the remaining questions are well-aligned with our goal of evaluating visual reasoning ability, rather than perceptual skills or the breadth of specialized knowledge. For questions with multiple possible answers, we first try to convert them into rule-based (will be introduced in Sec. 4.2) or multiple-choice questions; otherwise, discard them. Finally, to comprehensively evaluate the multimodal reasoning ability, we balance the distribution of questions across the three reasoning types. This approach prevents the benchmark from being overly biased towards evaluating the ability of any single reasoning type. Through this data curation process, we filter 1,008 questions from the initially collected questions.

**Metadata Annotation.** Further, we annotate questions in MME-Reasoning with information including question type (*i.e.*, multiple-choice, free-form, and rule-based), difficulty (*i.e.*, easy, medium, hard), capability (*i.e.*, pattern analysis, planning and exploring, spatial and temporal, calculation, casual chain analysis), and reasoning type (*i.e.*, deductive, inductive, and abductive). The specific rules are listed in our Appendix.

# 4. Experiments

## 4.1. Experimental Settings

We conduct extensive evaluations on SoTA MLLMs including: **(1) Thinking Models.** We first evaluate thinking MLLMs that focus on improving the models' multimodal reasoning which can be divided into ***Close-source models*** including (1) o1 (Jaech et al., 2024), and o4-mini (, 2025); (2) Gemini-2.5-Flash-Thinking and Gemini-2.5-Pro-Thinking (Gemini et al., 2023); (3) Claude-3.7-Sonnet-Thinking, Claude-4-Sonnet-Thinking (Anthropic, 2022); (4) Seed1.5-VL-Thinking (Guo et al., 2025a); and ***Open-source models*** including (1) QvQ-72B-Preview (Team, 2024); (2) Kimi-VL-A3B-Thinking (Team et al., 2025b); (3)LlamaV-

o1 (Thawakar et al., 2025); (4) Virgo-72B (Du et al., 2025). **(2) Chat Models.** ***Close-source models***: (1) GPT-4o (OpenAI, 2024); (2) Claude-3.7-Sonnet (Anthropic, 2022) (3) Kimi-latest (Team et al., 2025a); (4) Seed1.5-VL (Guo et al., 2025a). ***Open-source models***: (1) Qwen-2.5-VL (7B, 32B, 72B) (Qwen Team, 2025a); (2) InternVL-3 (8B, 38B, 78B) (Zhu et al., 2025); (3) LLaVA-Onevision-72B (Li et al., 2024); (4) Molmo (7B-O, 7B-D, 72B) (Deitke et al., 2024); (5) Kimi-VL-A3B-Instruct (Team et al., 2025b). **(3) Rule-based RL Models.** Rule-based RL is a highly promising strategy for eliciting reasoning paradigms in models. Therefore, we further evaluated MLLMs trained using Rule-based RL, including: (1) R1-VL (Zhang et al., 2025a), (2) R1-Onevision (Yang et al., 2025), (3) Vision-R1 (Huang et al., 2025), (4) MM-Eureka (7B, 32B) (Meng et al., 2025), (5) VL-Rethinker (7B, 72B) (Wang et al., 2025), (6) Bee-8B-RL (Zhang et al., 2025b). We use GPT-4o-mini to extract answers. Due to rate limits, we sample 302 questions to construct mini-set with the same distribution for o1's evaluation, all other models are evaluated on the entire benchmark.

## 4.2. Evaluation Protocols

Following MathVista (Lu et al., 2023), the evaluation consists of two steps: extracting answers and judging answers. For different types of questions, we designed specific prompts for GPT to extract answers that are composed of extraction rules and examples. For MCQ questions, we match the extracted answers with the reference answers. For free-form questions, we use GPT to judge the consistency between the extracted answers and the reference answers. For rule-based questions, we first use GPT to extract answers and convert them into an intermediate format, which is then judged using specific scripts. For example, in a Number Bridge problem, we first use GPT to extract the start and end points of each bridge, then convert the answers into a specific matrix format, and finally determine correctness based on predefined rules. An example of rule-based evaluation is shown in Fig. 5.

## 4.3. Main Results

**MME-Reasoning poses significant challenges for vision-language reasoning.** As shown in Tab. 3, the best-performing model, Gemini-2.5-Pro-Thinking, achieved an average score of 60.2%. The latest MLLM, Seed1.5-VL, achieved a comprehensive score of 59.9. Qwen2.5-VL and Claude-3.7-Sonnet achieved scores of 35.9 and 57.2 on OlympiadBench, yet only reached 34.1 on MME-Reasoning. These results indicate that the benchmark sets stringent standards for evaluating models' logical reasoning capabilities by comprehensively assessing three distinct reasoning types.

**Prominent bias in logical reasoning performance within MLLMs.** In almost all cases, models exhibit dominant de-

*Table 3.* Performance comparison of SoTA MLLMs. The top three are highlighted in blue. † indicates the model was evaluated on the mini-set. "T" represents "Thinking". CAL., P&E., PA., S&T., and CCA. represent calculation, plan and exploration, pattern analysis, spatial and temporal, and casual chain analysis.

| Model | Model Capability | | | | | Reasoning Type | | | AVG. |
|---|---|---|---|---|---|---|---|---|---|
| | CAL. | P& E. | PA. | S&T. | CCA. | DED. | IND. | ABD. | |
| *Human Performance* | | | | | | | | | |
| Human Expert† | 75.0 | 84.4 | 84.9 | 80.3 | 88.1 | 85.8 | 76.9 | 85.6 | 83.4 |
| *Close-source & Thinking* | | | | | | | | | |
| Gemini-3-Pro-T | **77.3** | **77.2** | **72.5** | **54.8** | 84.1 | **69.0** | **65.1** | **69.4** | **68.3** |
| Gemini-2.5-Pro-T | 68.0 | 64.4 | 53.7 | 52.1 | **90.3** | 64.0 | 51.7 | 62.8 | 60.2 |
| Seed1.5-VL-T | 67.2 | 62.7 | 56.0 | 47.2 | 82.6 | 64.5 | 52.3 | 60.8 | 59.9 |
| GPT-5.4 | 62.6 | 47.5 | 50.0 | 48.9 | 62.5 | 61.0 | 45.0 | 48.0 | 52.2 |
| o4-mini | 63.1 | 58.3 | 57.2 | 50.4 | 59.0 | 60.6 | 51.4 | 59.0 | 57.5 |
| o1† | 50.0 | 38.5 | 41.5 | 43.7 | 52.4 | 50.8 | 42.3 | 42.3 | 45.7 |
| Claude-4-Sonnet-T | 59.3 | 53.9 | 49.7 | 43.3 | 74.3 | 54.5 | 43.8 | 53.5 | 51.2 |
| Claude-4-Sonnet-T | 33.3 | 35.9 | 33.0 | 36.2 | 47.9 | 39.4 | 32.0 | 35.7 | 36.1 |
| Claude-3.7-Sonnet-T | 30.4 | 27.6 | 32.3 | 38.3 | 46.5 | 34.6 | 36.2 | 31.7 | 34.1 |
| Gemini-2.5-Flash-T | 19.8 | 21.3 | 20.9 | 33.0 | 38.9 | 28.1 | 22.1 | 24.6 | 25.2 |
| *Close-source & Chat* | | | | | | | | | |
| Seed1.5-VL | 52.0 | 42.0 | 38.4 | 44.0 | 72.9 | 54.9 | 45.0 | 41.0 | 47.5 |
| GPT-4o | 21.4 | 22.1 | 30.5 | 38.6 | 36.8 | 29.0 | 34.7 | 27.9 | 30.2 |
| Claude-3.7-Sonnet | 29.0 | 24.6 | 32.8 | 35.5 | 46.5 | 35.7 | 38.7 | 26.1 | 33.3 |
| Kimi-Latest | 21.4 | 17.4 | 19.8 | 29.1 | 41.0 | 27.7 | 25.4 | 19.9 | 24.4 |
| *Open-source & Thinking* | | | | | | | | | |
| QVQ-72B-Preview | 37.4 | 27.1 | 28.8 | 35.8 | 57.6 | 41.6 | 33.5 | 29.1 | 35.2 |
| Virgo-72B | 30.4 | 22.9 | 26.1 | 36.2 | 47.2 | 37.7 | 32.6 | 24.4 | 31.8 |
| VL-Rethinker-72B | 33.6 | 28.4 | 31.4 | 37.2 | 59.7 | 39.0 | 36.0 | 31.9 | 35.8 |
| VL-Rethinker-7B | 24.7 | 17.7 | 23.5 | 39.4 | 42.4 | 34.4 | 29.9 | 22.9 | 29.3 |
| MM-Eureka-Qwen-32B | 23.0 | 25.7 | 25.6 | 36.2 | 50.7 | 32.9 | 30.5 | 28.1 | 30.6 |
| MM-Eureka-Qwen-7B | 27.1 | 19.3 | 22.3 | 31.9 | 50.0 | 32.7 | 28.7 | 22.6 | 28.2 |
| Bee-8B | 47.4 | 26.0 | 34.8 | 36.5 | 43.1 | 48.6 | 38.7 | 25.9 | 38.2 |
| Vision-R1-7B | 18.2 | 18.0 | 17.9 | 34.4 | 36.1 | 27.4 | 26.3 | 18.1 | 24.0 |
| R1-Onevision-7B-RL | 19.5 | 12.2 | 20.0 | 31.6 | 27.1 | 27.7 | 24.8 | 14.6 | 22.5 |
| Kimi-VL-A3B-T | 28.7 | 16.0 | 19.5 | 32.3 | 35.4 | 33.3 | 25.1 | 18.1 | 25.9 |
| *Open-source & Chat* | | | | | | | | | |
| Qwen2.5-VL-72B | 31.7 | 25.1 | 27.2 | 37.9 | 53.5 | 39.0 | 32.3 | 29.9 | 34.1 |
| Qwen2.5-VL-32B | 32.2 | 26.8 | 24.4 | 39.0 | 52.1 | 40.5 | 27.5 | 29.6 | 33.2 |
| Qwen2.5-VL-7B | 22.2 | 18.2 | 21.9 | 35.1 | 36.1 | 31.4 | 27.5 | 20.9 | 26.8 |
| InternVL3-78B | 26.0 | 24.0 | 26.5 | 41.8 | 50.0 | 35.1 | 33.8 | 27.1 | 32.1 |
| InternVL3-38B | 23.0 | 18.5 | 23.0 | 38.3 | 41.7 | 33.5 | 29.0 | 22.1 | 28.4 |
| InternVL3-8B | 19.5 | 19.6 | 22.6 | 31.6 | 41.0 | 28.1 | 29.9 | 21.4 | 26.4 |
| Molmo-72B | 12.5 | 11.9 | 14.7 | 28.7 | 28.5 | 23.1 | 18.4 | 14.3 | 18.9 |
| Molmo-7B-D | 11.7 | 8.6 | 8.1 | 27.3 | 23.6 | 20.7 | 10.9 | 11.1 | 14.7 |
| LLaVA-OV-72B | 17.1 | 18.0 | 23.9 | 32.3 | 38.9 | 27.4 | 30.5 | 19.9 | 25.8 |
| Kimi-VL-A3B | 18.7 | 11.9 | 21.4 | 34.0 | 27.8 | 25.9 | 26.3 | 17.1 | 23.1 |

ductive reasoning performance, while abductive reasoning is considerably weaker. Closed-source models demonstrate an average deductive advantage of 5.38 over abductive reasoning, which widens to 9.81 among open-source models, making abductive reasoning a significant bottleneck in reasoning performance. Deductive reasoning maintains a high proportion in the training corpus due to its widespread distribution. Abductive reasoning processes usually involve larger exploration spaces and richer assumptions, hypotheses, and reflections, making their data challenging to scale. However, in the real world, non-deductive reasoning plays a crucial role in many scenarios such as scientific discover-

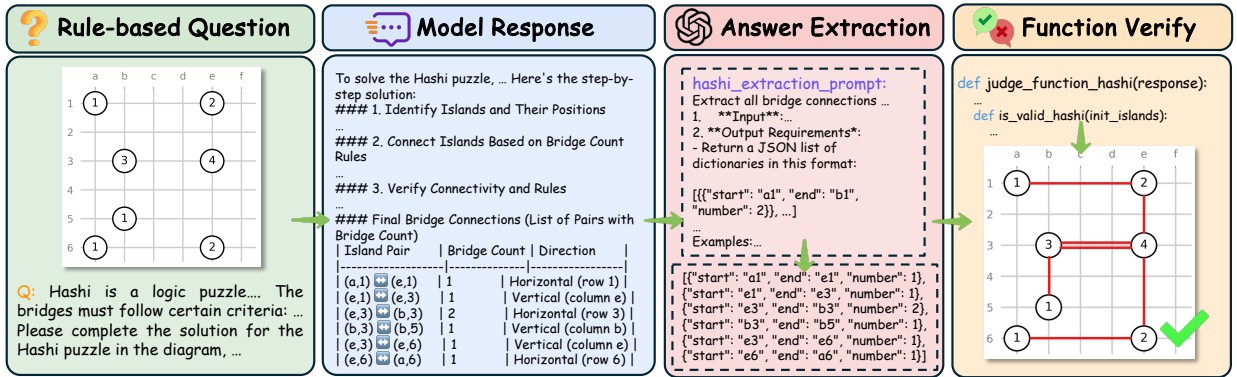

*Figure 5.* Evaluation of rule-based questions.

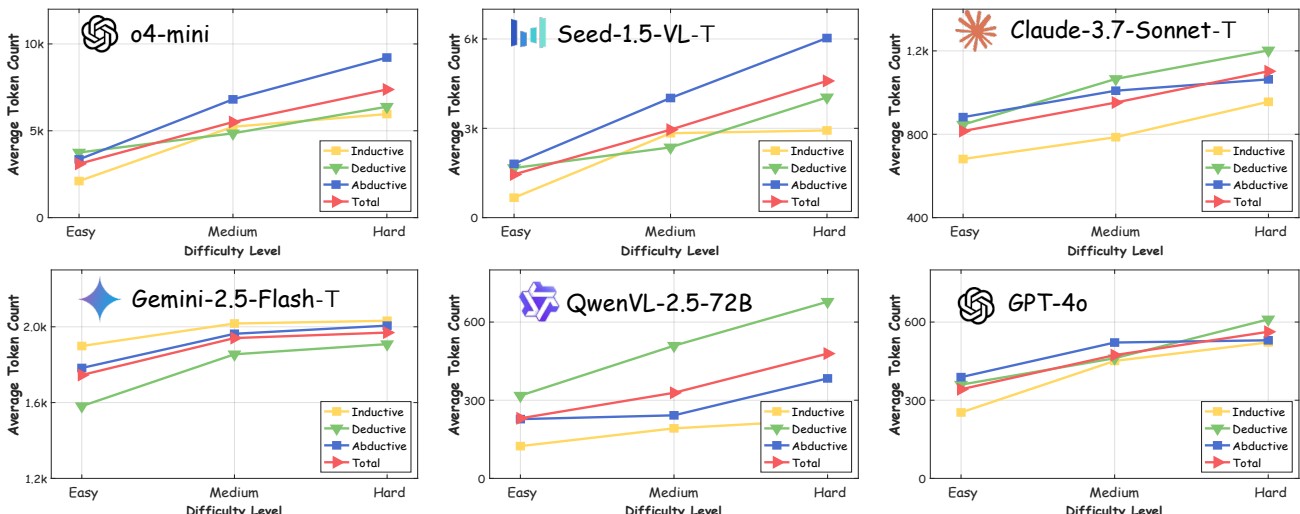

*Figure 6.* Comparison of Difficulty Level and Average Token Count on MME-Reasoning.

ies. These findings highlight the necessity for researchers to develop a more comprehensive understanding of models' logical reasoning abilities to facilitate their application in real-world scenarios.

**Comparison with Human Performance.** To evaluate expert-level performance on MME-Reasoning, we further report human performance on the mini-set of MME-Reasoning. As shown in Tab. 3, the human expert achieved an overall score of 83.4—significantly outperforming the best-performing thinking model. Looking deeper into the reasoning types, the human expert scored 85.8, 76.9, and 85.6 on deductive, inductive, and abductive reasoning, respectively, all of which are notably higher than the scores of the best-performing model. Moreover, the human expert demonstrated a particularly strong ability in abductive reasoning, with performance comparable to that in deductive reasoning—which is the key focus in current multi-modal reasoning research. This strength aligns with a few top-performing models, but stands in contrast to most baseline models, which show clear weaknesses in abductive reasoning. These results highlight the significant gap that still ex-

ists between current models and human-level performance in comprehensive multimodal reasoning evaluation.

**Limited performance in open-ended reasoning scenarios.** Models generally demonstrate relative advantages in Casual Chain Analysis but perform poorly on tasks involving Plan & Exploration. This may benefit from the autoregressive paradigm, continuously aiding models in learning causal dependencies within input sequences. However, it also highlights a critical shortcoming: current models struggle with planning and exploration in open-ended problem-solving spaces. To advance models in solving difficult practical problems, it is critical to innovate learning paradigms and strategies suitable for open scenarios.

**Comprehensive improvement of reasoning ability requires more attention.** In most cases, "thinking models" significantly outperform their base version. For example, Seed1.5-VL-T outperformed Seed1.5-VL by 12.4. However, we observed that methods adopting rule-based RL which activating the "thinking mode" of foundational models do not consistently outperform their base models on MME-Reasoning. Most models at the 7B scale experienced

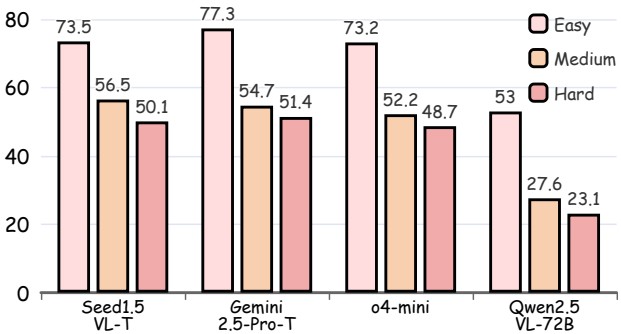

*Figure 7.* Results within different difficulty levels.

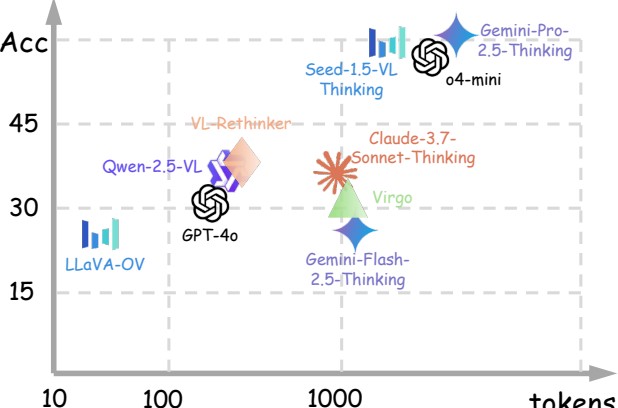

*Figure 8.* Response tokens vs. Performance.

*Table 4.* Human and LLM-based evaluation.

|  | Group 1 | Group 2 | Group 3 | GPT-4o |
|---|---|---|---|---|
| MathVista | 55.7% | 65.3% | 52.2% | 60.7% |
| MME-Reasoning | 96.1% | 97.3% | 95.8% | 94.4% |

performance degradation. This may be due to the training method, data, since existing works often concentrate on a small set of reasoning (*i.e.*, Math). We hope that more researchers can pay attention to a more comprehensive type of reasoning instead of only math.

### 4.4. Fine Grained Analysis

**Does increasing the length of the reasoning process help?** To investigate whether increased output length consistently leads to improved accuracy, we selected 10 representative models, including Chat Models (*e.g.*, GPT-4o) and Thinking Models (*e.g.*, o4-mini). In Fig. 8, we present the semi-log plot of average token count (ATC) versus accuracy. The overall trend reveals that models with longer outputs tend to achieve higher scores, indicating the effectiveness of extending the reasoning process to enhance reasoning performance. However, we observe that the profit brought by the increase of model output tokens is diminishing. Compared to Thinking Models, Chat Models demonstrate higher token efficiency. These findings highlight the computational cost associated with scaling up inference for improved performance. Balancing reasoning efficiency and model effectiveness remains a challenge for future research.

**Is the length of the reasoning process strongly correlated with task difficulty?** To examine whether models spontaneously allocate more inference budget to more challenging questions, we conducted research on representative Thinking and Chat Models. We first analyzed the accuracy of different models across varying levels of difficulty, as shown in Fig. 7. With increasing difficulty, model performance declines significantly, confirming the validity of MME-Reasoning's difficulty stratification. Besides, Fig. 6 illustrates the trend of ATC across different reasoning types and difficulty levels. It reveals a consistent pattern: overall, output length increases steadily with rising difficulty. This trend holds across varying output lengths, model categories, and reasoning types. Compared to Chat Models, Thinking Models exhibit a more pronounced increase in ATC as difficulty rises. For instance, the ATC of Seed1.5-VL increases by up to 3k tokens. In contrast, the ATC increase

for Qwen2.5-VL and GPT-4o remains within 300 tokens.

**Is MME-Reasoning a reasoning-centric benchmark?** We conduct experiments involving both human and LLM-based evaluation to show that MME-Reasoning is a reasoning-centric benchmark. **Human Evaluation**: We enlisted six PhD-level human experts, grouped into pairs (3 groups). Each pair assessed the questions from both MathVista and MME-Reasoning benchmarks to classify them based on the need for reasoning. **LLM-Based Evaluation**: We employ GPT-4o to assess the same classification task, further validating the reasoning-centric nature of our benchmark. Tab. 4 presents the proportion of questions requiring reasoning in each benchmark. The results show that the vast majority of questions in MME-Reasoning (over 95%) were classified as requiring reasoning to solve, which further verifies that MME-Reasoning is a reasoning-centric benchmark. Further, to show that our benchmark requires less knowledge, we follow VisualPuzzles (Song et al., 2025) to prompt GPT-4o to generate "knowledge concept checklists" for 50 randomly selected questions and manually verify each question. The results are shown in the following table, which further demonstrates that questions in MME-Reasoning need less advanced knowledge.

**Further Analysis of Human Performance.** In our manuscript, we report PhD-level performance on MME-Reasoning in Tab. 3. For a further assessment of performance across different backgrounds, five college-level students with varied backgrounds were recruited for testing. The results are presented in Tab. 5. The results show that while there is some performance variability among human participants, likely due to their diverse backgrounds, they consistently outperform the current models. This shows that there is still a performance gap between existing SoTA mod-

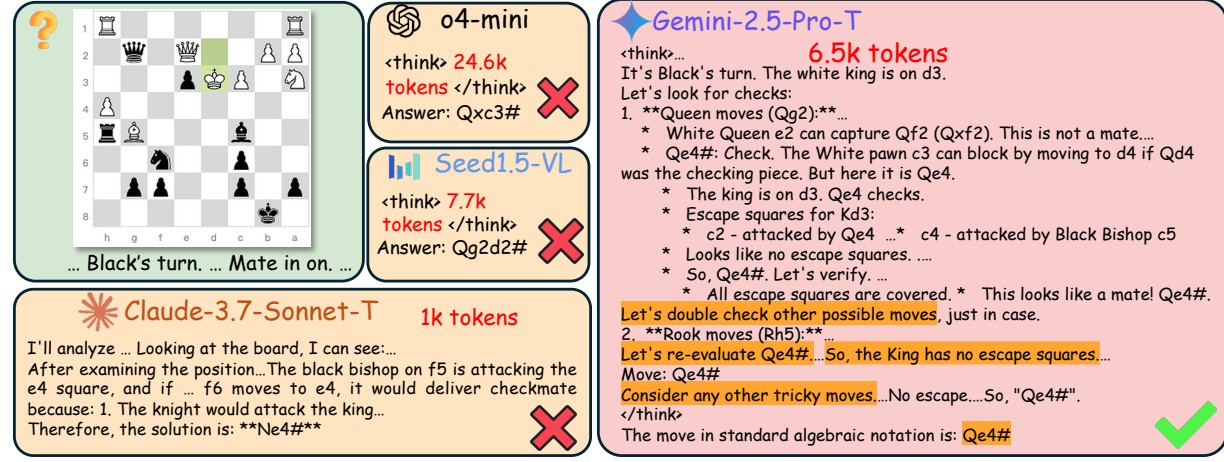

*Figure 9.* Case study of a Mate-in-one problem.

*Table 5.* Performance comparison of chat models with or w/o MCTS.

|  | Deductive | Inductive | Abductive | Average |
|---|---|---|---|---|
| Human Expert 1 | 85.8 | 76.9 | 85.6 | 83.4 |
| Human Expert 2 | 90.1 | 77.5 | 86.1 | 84.6 |
| Human Expert 3 | 82.1 | 71.5 | 86.1 | 79.9 |
| Human Expert 4 | 75.5 | 76.8 | 83.8 | 78.7 |
| Human Expert 5 | 80.8 | 75.2 | 85.4 | 80.5 |

*Table 6.* Knowledge concepts.

| Benchmark | Question numbers |
|---|---|
| MMMU | 3.9 |
| MME-Reasoning | 1.4 |

els and humans. Besides, we observed that the performance variability among human participants was more pronounced on deductive and inductive tasks compared to abductive ones. We attribute this to the fact that performance on puzzles and mathematical reasoning is often influenced by a participant's specific academic background and expertise. In contrast, abductive reasoning relies on more universal human abilities, such as the capacity for trial, critical thinking, and reflection. Compared to the models, humans have a much stronger ability to learn from trial and error. This results in a more stable and consistently high level of performance on abductive reasoning tasks across participants, regardless of their individual backgrounds.

### 4.5. Case Study

In Fig. 9, we present an example of abductive reasoning which demands planning and exploration. From this case, several key observations can be identified: *(1) Long reasoning process*: The selected models generated over 1k tokens in response, with o4-mini producing up to 24.6k tokens. This demonstrates that MME-Reasoning constitutes a highly challenging benchmark for multimodal reasoning. *(2) Planning in the problem-solving process*: The response includes multiple iterations of *"hypothesis generation (possible movement) – feasibility verification (check escape squares) – check "*, indicating that the model spontaneously engages in structured planning and reflection to explore solutions within an open-ended problem-solving space. *(3) Repetitive reflection*: We observed that the model

tends to revisit and reflect on the same reasoning paths multiple times—up to 7 instances in some cases. This behavior may result in significant computational overhead and informational redundancy. Balancing reasoning efficiency with performance remains a critical issue to be addressed.

## 5. Conclusion

We introduce MME-Reasoning, a comprehensive benchmark designed to evaluate MLLMs' logical reasoning abilities across inductive, deductive, and abductive reasoning types. Through careful data curation and an expanded evaluation protocol, our benchmark provides a holistic assessment of reasoning capabilities, beyond simple perception or high-level knowledge. Our experiments reveal that existing MLLMs still face significant challenges and exhibit notable performance imbalances across different reasoning types. These findings underscore the need for further research and development to enhance the reasoning abilities of MLLMs, paving the way for more generalizable AI systems.

## 6. Limitation

Despite our best efforts to cover a wide range of multimodal reasoning question types, it remains challenging to comprehensively collect all possible types of reasoning problems that occur in real-world scenarios. This is primarily because gathering and curating high-quality reasoning questions is often a time-consuming and labor-intensive process. Future work is needed to further enrich the diversity of question types and optimize dataset coverage.

## Impact Statement

This study utilizes samples extracted from existing quiz sources exclusively for the purposes of scholarly analysis and algorithmic testing. We emphasize that the included data samples are neither intended to nor capable of serving as a substitute for the original works. Furthermore, our human evaluation studies were conducted with fair compensation and informed consent from all participants.

## Acknowledgement

This work was supported by the New Generation Artificial Intelligence – National Science and Technology Major Project of China (Grant No. 2025ZD0121803). This work was also supported by the National Key R&D Program of China (No. 2026YFE0101200), Shanghai Natural Science Foundation (No. 23ZR1402900). This work was partially supported by the National Natural Science Foundation of China (No. 62306261), HK RGC-Early Career Scheme (No. 24211525), ITSP Platform Project (No. ITS/600/24FP) and the SHIAE Grant (No. 8115074). This study was supported in part by the Centre for Perceptual and Interactive Intelligence, a CUHK-led InnoCentre under the InnoHK initiative of the Innovation and Technology Commission of the Hong Kong Special Administrative Region Government. This work is partially supported by Hong Kong RGC Strategic Topics Grant (No. STG1/E-403/24-N), and CUHK-CUHK(SZ)-GDST Joint Collaboration Fund (No. YSP26-4760949).

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

# Appendix

# A. More Detailed Comparison Between MME-Reasoning and Other Benchmarks

In this section, we provide a more detailed comparison between MME-Reasoning and other benchmarks as shown in the Tab. 7. We further provide here a mapping between the sub-reasoning types in our dataset and the three major reasoning types in Tab. 8.

The difference can be summarized into the following three aspects:

- Covering More Reasoning Types and Subtypes: MME-Reasoning includes a broader range of reasoning types and subtypes. First, based on the three-stage theory of reasoning (premise, rules, and result), we categorize reasoning into three major classes (*i.e.*, inductive, deductive, and abductive), where two can infer the third. While previous reasoning benchmarks typically only cover inductive and deductive reasoning, our benchmark comprehensively incorporates abductive reasoning as well. Furthermore, compared to existing benchmarks, which usually divide the reasoning types into more fine-grained subtypes such as spatial or mathematical reasoning, MME-Reasoning covers an even wider set of reasoning subtypes including spatial reasoning, temporal reasoning, puzzle (quantitative, spatial, positional, attribute, stylistic), geometry, algebraic, statistical, scientific (physics, biology, chemistry) (as shown in the table), enabling a more comprehensive evaluation of MLLMs' reasoning abilities.

- Diversity of Question Formats: In terms of question types, existing benchmarks usually focus on MCQ, while MME-Reasoning encompasses three formats: MCQ, free-form, and rule-based. This allows us to cover a wider variety of question types, such as Sudoku and Number Bridge, and so on, which are not typically included in previous benchmarks.

- Variety of Image Types: MME-Reasoning also includes a wider range of image types. Compared to earlier reasoning benchmarks, we comprehensively cover images from Math (Geometry, Function), Biology, Chemistry, Physics, Charts, Puzzles (Quantitative, Spatial, Positional, Attribute, Stylistic, Chess and Card, Sudoku), Multi-view images, Temporal sequences, and others (such as chess). This extensive coverage ensures that our benchmark can more thoroughly assess the reasoning capabilities of MLLMs.

# B. More Experimental Results

## B.1. Full Results on MME-Reasoning

We present the performance of more baselines on MME-Reasoning in Tab 9, including OpenVLThinker (Deng et al., 2025), LMM-R1-MGT-PerceReason (Peng et al., 2025a), Mulberry (Yao et al., 2024), LlamaV-o1 (Thawakar et al., 2025) and Qwen2-VL series (Wang et al., 2024b).

## B.2. Full Results on Mini-set of MME-Reasoning

We randomly sampled 25% of the questions and conducted manual review to ensure that the diversity of image types was maintained. These sampled questions were then used to to construct the Mini-set. We also analyzed the question distributions of both the Mini-set and the Full-set to ensure the sampled questions retained the same distribution. The statistical results are presented in Tab 10.

We provide the performance of all baseline models on the Mini-set in Tab. 11. All baseline models achieved similar performance on both the Full-set and the Mini-set, further demonstrating the consistency of Mini-set and the comparability of model performance across different splits.

## B.3. Results on Different Question Types

We also evaluated the model's performance across different question types and present the results in Tab. 12.

## B.4. Results with Test-Time Compute Scaling

To evaluate whether the use of Test-Time Compute Scaling (TTS) methods can improve model performance on MME-Reasoning, we take Qwen2.5-VL-7B as an example and use Qwen2.5-VL-32B as the Reward Model. The evaluation is

*Table 7.* More detailed comparison between MME-Reasoning and other benchmarks.

| Benchmark | Size | Question Types | Answer Types | Reasoning Types | Image Types | Sub reasoning / Task Types |
|---|---|---|---|---|---|---|
| MathVista | 6.1k | MCQ, Free-form | Text | Inductive, Deductive | Math (Geometry, Function), Chart, Table, Puzzle, Others (Synthetic scene, Abstract Scene) | Geometry, Algebraic, Statistical, Scientific, Puzzles |
| MathVerse | 11.5K | MCQ, Free-form | Text | Deductive | Math (Geometry, Function) | Geometry, Algebraic |
| LogicVista | 0.4K | MCQ | Image | Inductive, Deductive | Text-rich image, Table, Chart, Physics, Puzzles, Others (3D shape) | Spatial, Scientific, Puzzles |
| VisualPuzzles | 1.2K | MCQ | Image+Text | Inductive | Puzzles (Quantitative, Spatial) | Puzzles (Quantitative, Spatial) |
| VisuLogic | 1K | MCQ | Image | Inductive | Puzzles (Quantitative, Spatial, Positional, Attribute, Stylistic) | Puzzles (Quantitative, Spatial, Positional, Attribute, Stylistic, Others) |
| Rbench-V | 803 | MCQ, Free-form | Image+Text | Inductive, Deductive | Math, Physics, Counting, Puzzles, Games | Geometry, Physics, Counting, Puzzle, Games |
| MM-IQ | 2.7K | MCQ | Image+Text | Inductive | Puzzles | Puzzles |
| MME-Reasoning | 1.2K | MCQ, Free-form, Rule-based | Image+Text | Inductive, Deductive, Abductive | Math (Geometry, function), Biology, Chemistry, Physics, Chart, Puzzles (Quantitative, Spatial, Positional, Attribute, Stylistic, chess and card, sudoku), Multi-view image, Temporal sequence, Others (chess) | Spatial, Temporal, Puzzle (Quantitative, Spatial, Positional, Attribute, Stylistic, Others), Geometry, Algebraic, Statistical, Scientific (Physics, Chemistry, Biology) |

*Table 8.* The mapping between reasoning sub-types and major reasoning types.

| Reasoning Sub-type | Deductive | Inductive | Abductive |
|---|---|---|---|
| Spatial Reasoning | ✓ | | ✓ |
| Temporal Reasoning | | | ✓ |
| Geometry Reasoning | ✓ | | |
| Algebraic Reasoning | ✓ | | |
| Statistical Reasoning | ✓ | | |
| Scientific Reasoning-Physic | ✓ | | ✓ |
| Scientific Reasoning-Biology | ✓ | | ✓ |
| Scientific Reasoning-Chemistry | ✓ | | ✓ |
| Puzzle-Quantitative Reasoning | ✓ | ✓ | |
| Puzzle-Spatial Reasoning | ✓ | ✓ | ✓ |
| Puzzle-Positional Reasoning | | ✓ | |
| Puzzle-Attribute Reasoning | | ✓ | |
| Puzzle-Stylistic Reasoning | | ✓ | |
| Puzzle-Others Nikoli-style Puzzles and Chess | | | ✓ |

*Table 9.* Performance comparison of state-of-the-art MLLMs on MME-Reasoning. The top three are highlighted in `blue`. "T" represents "Thinking".

| Model | Model Capability | | | | | Reasoning Type | | | AVG. |
|---|---|---|---|---|---|---|---|---|---|
| | CAL. | P& E. | PA. | S&T. | CCA. | DED. | IND. | ABD. | |
| *Close-source & Thinking* | | | | | | | | | |
| Gemini-2.5-Pro-T | **68.0** | **64.4** | 53.7 | **52.1** | **90.3** | 64.0 | 51.7 | **62.8** | **60.2** |
| Seed1.5-VL-T | 67.2 | 62.7 | 56.0 | 47.2 | 82.6 | **64.5** | **52.3** | 60.8 | 59.9 |
| o4-mini | 63.1 | 58.3 | **57.2** | 50.4 | 59.0 | 60.6 | 51.4 | 59.0 | 57.5 |
| Claude-4-Sonnet-T | 33.3 | 35.9 | 33.0 | 36.2 | 47.9 | 39.4 | 32.0 | 35.7 | 36.1 |
| Claude-3.7-Sonnet-T | 30.4 | 27.6 | 32.3 | 38.3 | 46.5 | 34.6 | 36.2 | 31.7 | 34.1 |
| Gemini-2.5-Flash-T | 19.8 | 21.3 | 20.9 | 33.0 | 38.9 | 28.1 | 22.1 | 24.6 | 25.2 |
| *Close-source & Chat* | | | | | | | | | |
| Seed1.5-VL | 52.0 | 42.0 | 38.4 | 44.0 | 72.9 | 54.9 | 45.0 | 41.0 | 47.5 |
| GPT-4o | 21.4 | 22.1 | 30.5 | 38.6 | 36.8 | 29.0 | 34.7 | 27.9 | 30.2 |
| Claude-3.7-Sonnet | 29.0 | 24.6 | 32.8 | 35.5 | 46.5 | 35.7 | 38.7 | 26.1 | 33.3 |
| Kimi-Latest | 21.4 | 17.4 | 19.8 | 29.1 | 41.0 | 27.7 | 25.4 | 19.9 | 24.4 |
| *Open-source & Thinking* | | | | | | | | | |
| QVQ-72B-Preview | 37.4 | 27.1 | 28.8 | 35.8 | 57.6 | 41.6 | 33.5 | 29.1 | 35.2 |
| Virgo-72B | 30.4 | 22.9 | 26.1 | 36.2 | 47.2 | 37.7 | 32.6 | 24.4 | 31.8 |
| VL-Rethinker-72B | 33.6 | 28.4 | 31.4 | 37.2 | 59.7 | 39.0 | 36.0 | 31.9 | 35.8 |
| VL-Rethinker-7B | 24.7 | 17.7 | 23.5 | 39.4 | 42.4 | 34.4 | 29.9 | 22.9 | 29.3 |
| MM-Eureka-Qwen-32B | 23.0 | 25.7 | 25.6 | 36.2 | 50.7 | 32.9 | 30.5 | 28.1 | 30.6 |
| MM-Eureka-Qwen-7B | 27.1 | 19.3 | 22.3 | 31.9 | 50.0 | 32.7 | 28.7 | 22.6 | 28.2 |
| R1-VL-7B | 16.3 | 11.6 | 17.7 | 30.9 | 26.4 | 25.3 | 21.8 | 15.8 | 21.1 |
| Vision-R1-7B | 18.2 | 18.0 | 17.9 | 34.4 | 36.1 | 27.4 | 26.3 | 18.1 | 24.0 |
| R1-Onevision-7B-RL | 19.5 | 12.2 | 20.0 | 31.6 | 27.1 | 27.7 | 24.8 | 14.6 | 22.5 |
| Kimi-VL-A3B-T | 28.7 | 16.0 | 19.5 | 32.3 | 35.4 | 33.3 | 25.1 | 18.1 | 25.9 |
| OpenVLThinker-7B | 19.8 | 14.6 | 19.3 | 35.8 | 34.7 | 30.7 | 24.8 | 17.3 | 24.6 |
| LMM-R1-MGT-PerceReason | 22.2 | 16.0 | 23.7 | 37.9 | 34.0 | 30.3 | 32.3 | 20.1 | 27.4 |
| Mulberry | 14.6 | 13.3 | 18.8 | 33.7 | 31.3 | 23.8 | 25.4 | 17.6 | 22.1 |
| LlamaV-o1 | 14.9 | 7.7 | 16.5 | 28.0 | 25.0 | 22.4 | 21.5 | 12.3 | 18.8 |
| *Open-source & Chat* | | | | | | | | | |
| Qwen2.5-VL-72B | 31.7 | 25.1 | 27.2 | 37.9 | 53.5 | 39.0 | 32.3 | 29.9 | 34.1 |
| Qwen2.5-VL-32B | 32.2 | 26.8 | 24.4 | 39.0 | 52.1 | 40.5 | 27.5 | 29.6 | 33.2 |
| Qwen2.5-VL-7B | 22.2 | 18.2 | 21.9 | 35.1 | 36.1 | 31.4 | 27.5 | 20.9 | 26.8 |
| Qwen2.5-VL-3B | 17.6 | 15.5 | 19.0 | 39.7 | 32.6 | 28.5 | 27.5 | 19.6 | 25.6 |
| Qwen2-VL-72B | 19.2 | 19.3 | 24.9 | 36.2 | 44.4 | 28.8 | 32.3 | 22.1 | 27.5 |
| Qwen2-VL-7B | 15.7 | 12.4 | 19.8 | 37.9 | 30.5 | 25.5 | 25.7 | 19.7 | 23.4 |
| Qwen2-VL-2B | 13.0 | 8.1 | 19.3 | 31.6 | 19.4 | 22.7 | 25.7 | 11.8 | 19.9 |
| InternVL3-78B | 26.0 | 24.0 | 26.5 | 41.8 | 50.0 | 35.1 | 33.8 | 27.1 | 32.1 |
| InternVL3-38B | 23.0 | 18.5 | 23.0 | 38.3 | 41.7 | 33.5 | 29.0 | 22.1 | 28.4 |
| InternVL3-8B | 19.5 | 19.6 | 22.6 | 31.6 | 41.0 | 28.1 | 29.9 | 21.4 | 26.4 |
| Molmo-72B | 12.5 | 11.9 | 14.7 | 28.7 | 28.5 | 23.1 | 18.4 | 14.3 | 18.9 |
| Molmo-7B-D | 11.7 | 8.6 | 8.1 | 27.3 | 23.6 | 20.7 | 10.9 | 11.1 | 14.7 |
| Molmo-7B-O | 8.1 | 5.5 | 11.6 | 22.7 | 15.3 | 16.6 | 16.0 | 7.5 | 13.4 |
| LLaVA-OV-72B | 17.1 | 18.0 | 23.9 | 32.3 | 38.9 | 27.4 | 30.5 | 19.9 | 25.8 |
| Kimi-VL-A3B | 18.7 | 11.9 | 21.4 | 34.0 | 27.8 | 25.9 | 26.3 | 17.1 | 23.1 |

*Table 10.* Comparison of statistics between full and mini-set of MME-Reasoning.

| Split | Reasoning Type | | | Question Type | | | Difficulty Level | | |
|---|---|---|---|---|---|---|---|---|---|
| | DED. | IND. | ABD. | Open | MCQ | Rule. | Easy | Medium | Hard |
| Mini | 39.7% | 25.8% | 34.4% | 32.4% | 58.3% | 9.3% | 31.8% | 39.4% | 28.8% |
| Full | 38.6% | 27.9% | 33.5% | 31.6% | 58.5% | 9.9% | 30.8% | 39.3% | 29.9% |

conducted using the Monte Carlo Tree Search (MCTS) algorithm, with the settings: *branch = 3* and *max-iteration = 18*. The results are shown in Table 13.

*Table 11.* Performance comparison of state-of-the-art MLLMs on mini set of MME-Reasoning. The top three are highlighted in  blue . "T" represents "Thinking".

| Model | Model Capability | | | | | Reasoning Type | | | AVG. |
|---|---|---|---|---|---|---|---|---|---|
| | CAL. | P& E. | PA. | S&T. | CCA. | DED. | IND. | ABD. | |
| *Human Performance* | | | | | | | | | |
| Human Expert | 75.0 | 84.4 | 84.9 | 80.3 | 88.1 | 85.8 | 76.9 | 85.6 | 83.4 |
| *Close-source & Thinking* | | | | | | | | | |
| Gemini-2.5-Pro-T | 66.0 | 63.5 | **58.5** | **49.3** | **85.7** | 60.8 | **55.1** | 65.4 | 60.9 |
| Seed1.5-VL-T | **68.0** | **67.7** | **58.5** | **49.3** | 83.3 | **67.5** | 48.7 | **67.3** | **62.6** |
| o4-mini | 64.0 | 58.3 | 56.6 | 45.1 | 54.8 | 57.5 | 51.3 | 60.6 | 57.0 |
| o1 | 50.0 | 38.5 | 41.5 | 43.7 | 52.4 | 50.8 | 42.3 | 42.3 | 45.7 |
| Claude-4-Sonnet-T | 33.0 | 30.2 | 35.8 | 39.4 | 50.0 | 42.5 | 37.2 | 33.7 | 38.1 |
| Claude-3.7-Sonnet-T | 30.0 | 17.7 | 36.8 | 38.0 | 38.1 | 31.7 | 42.3 | 27.9 | 33.1 |
| Gemini-2.5-Flash-T | 18.0 | 16.7 | 15.1 | 39.4 | 33.3 | 27.5 | 19.2 | 26.0 | 24.8 |
| *Close-source & Chat* | | | | | | | | | |
| Seed1.5-VL | 50.0 | 42.7 | 34.9 | 40.8 | 69.0 | 57.5 | 39.7 | 39.4 | 46.7 |
| GPT-4o | 20.0 | 24.0 | 24.5 | 40.8 | 33.3 | 31.7 | 28.2 | 27.9 | 29.5 |
| Claude-3.7-Sonnet | 27.0 | 22.9 | 34.0 | 31.0 | 42.9 | 31.7 | 38.5 | 27.9 | 32.1 |
| Kimi-Latest | 22.0 | 17.7 | 17.9 | 29.6 | 33.3 | 30.8 | 23.1 | 19.2 | 24.8 |
| *Open-source & Thinking* | | | | | | | | | |
| QVQ-72B-Preview | 36.0 | 24.0 | 34.0 | 33.8 | 47.6 | 38.3 | 37.2 | 29.8 | 35.1 |
| Virgo-72B | 28.0 | 18.8 | 27.4 | 43.7 | 38.1 | 37.5 | 41.0 | 21.2 | 32.8 |
| VL-Rethinker-72B | 23.0 | 25.0 | 29.2 | 39.4 | 42.9 | 34.2 | 32.1 | 31.7 | 32.8 |
| VL-Rethinker-7B | 23.0 | 16.7 | 21.7 | 47.9 | 40.5 | 35.8 | 28.2 | 26.0 | 30.5 |
| MM-Eureka-Qwen-32B | 23.0 | 20.8 | 26.4 | 38.0 | 38.1 | 32.5 | 34.6 | 25.0 | 30.5 |
| MM-Eureka-Qwen-7B | 28.0 | 17.7 | 21.7 | 32.4 | 50.0 | 32.5 | 32.1 | 22.1 | 28.8 |
| R1-VL-7B | 10.0 | 10.4 | 16.0 | 35.2 | 16.7 | 23.3 | 19.2 | 16.3 | 19.9 |
| Vision-R1-7B | 14.0 | 12.5 | 18.9 | 39.4 | 31.0 | 26.7 | 29.5 | 16.3 | 23.8 |
| R1-Onevision-7B-RL | 15.0 | 10.4 | 22.6 | 35.2 | 19.0 | 22.5 | 30.8 | 16.3 | 22.5 |
| Kimi-VL-A3B-T | 30.0 | 9.4 | 19.8 | 26.8 | 31.0 | 28.3 | 26.9 | 16.3 | 23.8 |
| OpenVLThinker-7B | 14.0 | 14.6 | 14.2 | 33.8 | 28.6 | 29.2 | 16.7 | 16.3 | 21.5 |
| LMM-R1-MGT-PerceReason | 27.0 | 14.6 | 23.6 | 38.0 | 33.3 | 35.8 | 33.3 | 18.3 | 29.1 |
| Mulberry | 19.0 | 15.6 | 18.9 | 33.8 | 33.3 | 28.3 | 23.1 | 18.3 | 23.5 |
| LlamaV-o1 | 15.0 | 8.3 | 17.9 | 31.0 | 26.2 | 23.3 | 23.1 | 15.4 | 20.5 |
| *Open-source & Chat* | | | | | | | | | |
| Qwen2.5-VL-72B | 31.0 | 19.8 | 25.5 | 38.0 | 42.9 | 39.2 | 32.1 | 26.0 | 32.8 |
| Qwen2.5-VL-32B | 31.0 | 28.1 | 28.3 | 40.8 | 45.2 | 41.7 | 34.6 | 27.9 | 35.1 |
| Qwen2.5-VL-7B | 19.0 | 16.7 | 24.5 | 38.0 | 33.3 | 32.5 | 30.8 | 21.2 | 28.1 |
| Qwen2.5-VL-3B | 21.0 | 14.6 | 21.2 | 39.4 | 31.0 | 30.0 | 30.8 | 21.2 | 27.2 |
| Qwen2-VL-72B | 20.0 | 19.8 | 28.3 | 38.0 | 38.1 | 34.2 | 39.7 | 19.2 | 30.5 |
| Qwen2-VL-7B | 16.0 | 9.4 | 25.5 | 33.8 | 26.2 | 22.5 | 34.6 | 16.3 | 23.5 |
| Qwen2-VL-2B | 12.0 | 9.4 | 17.9 | 29.6 | 19.0 | 23.3 | 23.1 | 11.5 | 19.2 |
| InternVL3-78B | 25.0 | 22.9 | 33.0 | 42.3 | 40.5 | 36.7 | 43.6 | 24.0 | 34.1 |
| InternVL3-38B | 19.0 | 19.8 | 26.4 | 36.6 | 38.1 | 31.7 | 33.3 | 23.1 | 29.1 |
| InternVL3-8B | 19.0 | 20.8 | 29.2 | 23.9 | 35.7 | 26.7 | 35.9 | 21.2 | 27.2 |
| Molmo-72B | 11.0 | 13.5 | 16.0 | 35.2 | 31.0 | 26.7 | 21.8 | 17.3 | 22.2 |
| Molmo-7B-D | 12.0 | 8.3 | 12.3 | 28.2 | 16.7 | 22.5 | 15.4 | 9.6 | 16.2 |
| Molmo-7B-O | 7.0 | 4.2 | 11.3 | 25.4 | 14.3 | 19.2 | 17.9 | 4.8 | 13.9 |
| LLaVA-OV-72B | 13.0 | 19.8 | 25.5 | 23.9 | 35.7 | 25.0 | 30.8 | 17.3 | 23.8 |
| Kimi-VL-A3B | 18.0 | 8.3 | 18.9 | 29.6 | 9.5 | 23.3 | 23.1 | 11.5 | 19.2 |

*Table 12.* Performance across different question types on MME-Reasoning. The top three are highlighted in `green`. † indicates the model was evaluated on the mini-set. "T" represents "Thinking".

| Model | Choice | | | | Open | | | | Rule |
|---|---|---|---|---|---|---|---|---|---|
| | DED. | IND. | ABD. | ALL | DED. | IND. | ABD. | ALL | ABD.&ALL |
| *Close-source & Thinking* | | | | | | | | | |
| Gemini-2.5-Pro-T | **58.0** | 49.8 | **63.6** | 55.7 | 75.9 | 61.5 | **60.0** | **66.9** | 66.1 |
| Seed1.5-VL-T | 57.3 | **54.2** | 60.2 | **56.5** | **78.5** | 44.2 | 59.4 | 65.3 | 63.5 |
| o4-mini | 57.3 | 48.7 | 61.9 | 54.7 | 67.1 | **67.3** | 48.5 | 58.9 | **71.3** |
| o1† | 46.2 | 42.4 | 53.3 | 46.0 | 60.0 | 45.5 | 36.2 | 46.9 | 40.7 |
| Claude-4-Sonnet-T | 41.1 | 33.2 | 40.7 | 37.9 | 36.5 | 25.0 | 35.2 | 34.3 | 31.3 |
| Claude-3.7-Sonnet-T | 38.0 | 39.7 | 46.6 | 40.1 | 28.5 | 17.3 | 31.5 | 28.3 | 16.5 |
| Gemini-2.5-Flash-T | 31.7 | 23.8 | 37.3 | 29.5 | 21.5 | 11.5 | 23.0 | 20.8 | 13.9 |
| *Close-source & Chat* | | | | | | | | | |
| Seed1.5-VL | 54.0 | 46.2 | 59.3 | 51.8 | 57.0 | 38.5 | 42.4 | 48.0 | 20.0 |
| GPT-4o | 36.3 | 38.3 | 44.1 | 39.1 | 15.2 | 17.3 | 27.9 | 21.1 | 7.0 |
| Claude-3.7-Sonnet | 38.7 | 42.2 | 37.3 | 39.9 | 30.4 | 21.2 | 27.9 | 28.0 | 12.2 |
| Kimi-Latest | 31.3 | 29.6 | 38.1 | 31.8 | 20.9 | 3.8 | 20.0 | 18.1 | 0.9 |
| *Open-source & Thinking* | | | | | | | | | |
| QVQ-72B-Preview | 43.7 | 35.0 | 45.8 | 40.6 | 38.0 | 26.9 | 31.5 | 33.6 | 8.7 |
| Virgo-72B | 39.7 | 36.8 | 47.5 | 39.9 | 34.2 | 11.5 | 22.4 | 25.9 | 3.5 |
| VL-Rethinker-72B | 43.3 | 38.3 | 53.4 | 43.0 | 31.0 | 25.0 | 29.1 | 29.3 | 13.9 |
| VL-Rethinker-7B | 41.3 | 33.9 | 51.7 | 40.1 | 21.5 | 9.6 | 16.4 | 17.6 | 2.6 |
| MM-Eureka-Qwen-32B | 36.7 | 33.2 | 49.2 | 37.4 | 25.9 | 17.3 | 26.1 | 24.8 | 9.6 |
| MM-Eureka-Qwen-7B | 36.7 | 32.9 | 41.5 | 36.0 | 25.3 | 7.7 | 21.8 | 21.3 | 4.3 |
| R1-VL-7B | 31.7 | 24.9 | 34.7 | 29.5 | 13.3 | 5.8 | 12.7 | 12.0 | 0.9 |
| Vision-R1-7B | 32.3 | 29.6 | 33.9 | 31.5 | 18.4 | 9.6 | 16.4 | 16.3 | 4.3 |
| R1-Onevision-7B-RL | 34.3 | 27.8 | 31.4 | 31.2 | 15.2 | 9.6 | 12.1 | 13.1 | 0.9 |
| Kimi-VL-A3B-T | 33.0 | 27.4 | 34.7 | 31.1 | 34.2 | 13.5 | 14.5 | 22.7 | 6.1 |
| OpenVLThinker-7B | 38.7 | 28.2 | 41.5 | 35.0 | 15.8 | 7.7 | 11.5 | 12.8 | 0.9 |
| LMM-R1-MGT-PerceReason | 36.3 | 36.1 | 44.9 | 37.7 | 19.0 | 13.5 | 15.8 | 16.8 | 0.9 |
| Mulberry | 30.0 | 28.9 | 42.4 | 31.7 | 12.0 | 7.7 | 11.5 | 11.2 | 0.9 |
| LlamaV-o1 | 26.7 | 25.3 | 29.7 | 26.6 | 14.6 | 1.9 | 7.9 | 9.9 | 0.9 |
| *Open-source & Chat* | | | | | | | | | |
| Qwen2.5-VL-72B | 41.0 | 34.3 | 55.1 | 40.7 | 34.8 | 23.1 | 26.1 | 29.3 | 9.6 |
| Qwen2.5-VL-32B | 44.0 | 30.0 | 50.0 | 39.4 | 34.2 | 15.4 | 27.3 | 28.5 | 12.2 |
| Qwen2.5-VL-7B | 39.0 | 30.0 | 41.5 | 35.8 | 17.1 | 15.4 | 18.2 | 17.3 | 3.5 |
| Qwen2.5-VL-3B | 33.3 | 30.7 | 46.6 | 34.5 | 19.6 | 11.5 | 13.9 | 16.0 | 0.0 |
| Qwen2-VL-72B | 35.3 | 36.5 | 46.6 | 37.7 | 16.5 | 11.5 | 18.8 | 16.8 | 1.7 |
| Qwen2-VL-7B | 32.7 | 28.9 | 45.8 | 33.4 | 12.0 | 9.6 | 12.7 | 12.0 | 0.9 |
| Qwen2-VL-2B | 30.0 | 30.3 | 31.4 | 30.4 | 8.9 | 1.9 | 6.1 | 6.7 | 0.0 |
| InternVL3-78B | 40.0 | 37.9 | 54.2 | 41.6 | 25.9 | 13.5 | 23.0 | 22.9 | 5.2 |
| InternVL3-38B | 36.7 | 32.1 | 48.3 | 36.8 | 27.8 | 13.5 | 17.0 | 21.1 | 2.6 |
| InternVL3-8B | 29.7 | 35.7 | 47.5 | 35.1 | 25.3 | 0.0 | 17.0 | 18.1 | 0.9 |
| Molmo-72B | 30.0 | 20.2 | 30.5 | 26.2 | 10.1 | 9.6 | 11.5 | 10.7 | 1.7 |
| Molmo-7B-D | 27.0 | 12.6 | 23.7 | 20.7 | 8.9 | 1.9 | 9.7 | 8.3 | 0.0 |
| Molmo-7B-O | 23.0 | 18.4 | 16.9 | 20.1 | 4.4 | 3.8 | 6.1 | 5.1 | 0.0 |
| LLaVA-OV-72B | 33.7 | 35.4 | 39.0 | 35.3 | 15.8 | 5.8 | 18.8 | 15.7 | 1.7 |
| Kimi-VL-A3B | 31.3 | 30.0 | 42.4 | 32.7 | 15.8 | 7.7 | 9.1 | 11.7 | 2.6 |

*Table 13.* Performance comparison of chat models with or w/o MCTS.

| Model | Model Capability | | | | | Reasoning Type | | | AVG. |
|---|---|---|---|---|---|---|---|---|---|
| | CAL. | P& E. | PA. | S&T. | CCA. | DED. | IND. | ABD. | |
| Qwen2.5-VL-7B | 22.2 | 18.2 | 21.9 | 35.1 | 36.1 | 31.4 | 27.5 | 20.9 | 26.8 |
| *+ MCTS* | 20.6 | 13.8 | 18.8 | 30.5 | 35.4 | 28.1 | 23.6 | 17.6 | 23.3 |

Under the MCTS-based setting, the model's performance dropped noticeably across all reasoning types. We attribute this decline to two main factors: (1) Questions in MME-Reasoning often involve complex parallel reasoning, hypothesis generation, and reflection, rather than simple linear logical progression. These characteristics may not be effectively captured by the Reward Model. (2) The limited capabilities of the Reward Model result in guidance that lacks practical utility.

*Table 14.* Performance comparison of SoTA chat models with or w/o CoT prompt.

| Model | Model Capability | | | | | Reasoning Type | | | AVG. |
|---|---|---|---|---|---|---|---|---|---|
| | CAL. | P& E. | PA. | S&T. | CCA. | DED. | IND. | ABD. | |
| Qwen2.5-VL-7B | 22.2 | 18.2 | 21.9 | 35.1 | 36.1 | 31.4 | 27.5 | 20.9 | 26.8 |
| *+ CoT prompt* | 20.3 | 18.5 | 17.9 | 33.3 | 38.9 | 28.3 | 21.4 | 23.6 | 24.7 |
| Qwen2.5-VL-32B | 32.2 | 26.8 | 24.4 | 39.0 | 52.1 | 40.5 | 27.5 | 29.6 | 33.2 |
| *+ CoT prompt* | 29.0 | 24.6 | 23.3 | 40.8 | 52.1 | 40.1 | 28.7 | 26.4 | 32.3 |
| Qwen2.5-VL-72B | 31.7 | 25.1 | 27.2 | 37.9 | 53.5 | 39.0 | 32.3 | 29.9 | 34.1 |
| *+ CoT prompt* | 32.5 | 26.2 | 25.1 | 37.2 | 52.8 | 37.5 | 30.8 | 30.0 | 33.0 |
| InternVL3-8B | 19.5 | 19.6 | 22.6 | 31.6 | 41.0 | 28.1 | 29.9 | 21.4 | 26.4 |
| *+ CoT prompt* | 21.1 | 16.3 | 20.2 | 31.6 | 38.2 | 31.2 | 26.9 | 16.8 | 25.2 |
| InternVL3-38B | 23.0 | 18.5 | 23.0 | 38.3 | 41.7 | 33.5 | 29.0 | 22.1 | 28.4 |
| *+ CoT prompt* | 28.7 | 24.3 | 28.6 | 38.3 | 48.6 | 37.5 | 32.9 | 26.9 | 32.7 |
| InternVL3-78B | 26.0 | 24.0 | 26.5 | 41.8 | 50.0 | 35.1 | 33.8 | 27.1 | 32.1 |
| *+ CoT prompt* | 29.0 | 22.9 | 27.0 | 40.8 | 48.6 | 36.6 | 35.1 | 26.9 | 32.9 |

We leave further exploration of TTS methods for reasoning to future work and hope that MME-Reasoning can serve as a representative benchmark for developing more general and comprehensive TTS algorithms in reasoning tasks.

## B.5. Results with CoT Prompt

| Model | CoT Prompt |
|---|---|
| **Qwen2.5-VL** | Let's think step by step. |
| **InternVL3** | Answer the preceding question. The last line of your response should follow this format: 'Answer: $FINAL_ANSWER' (without quotes), where 'FINAL_ANSWER' is your conclusion based on the reasoning provided. If you are uncertain or the problem is too complex, make a reasoned guess based on the information provided. Avoid repeating steps indefinitely—provide your best guess even if unsure. Think step by step logically, considering all relevant information before answering. |

*Table 15.* Chain-of-Thought Prompts for Different Models

Chain-of-Thought (CoT) prompting increases output length by encouraging explicit output of the thought process, thereby enhancing reasoning performance. To investigate the impact of CoT on performance in MME-Reasoning, we evaluated the Qwen2.5-VL and InternVL3 series using CoT prompts shown in Tab. 15. The results are presented in Tab. 14.

We observed that the Qwen2.5-VL models naturally tend to generate their reasoning process, so adding a CoT prompt did not significantly increase output length. In contrast, InternVL3 models, under default settings, tend to directly output the final answer, and the CoT prompt substantially increased output length.

In terms of performance, adding the CoT prompt consistently led to performance degradation for the Qwen2.5-VL series. For InternVL3, performance dropped for the 7B model but improved for the larger 38B and 78B models. One possible hypothesis is that for models already inclined to produce long outputs, explicit CoT instructions might introduce noise into the reasoning process. Conversely, for models that tend to answer questions directly, smaller models struggle to produce helpful and correct CoT outputs, but as model size increases, they begin to benefit noticeably from relatively accurate reasoning processes.

## B.6. Token Usage of Thinking Models

In Fig. 10, we present the average token length of different thinking models on MME-Reasoning. Overall, there is a clear trend indicating that better model performance is often associated with longer reasoning paths. However, we also observe diminishing returns between output length and performance in both open-source and closed-source models.

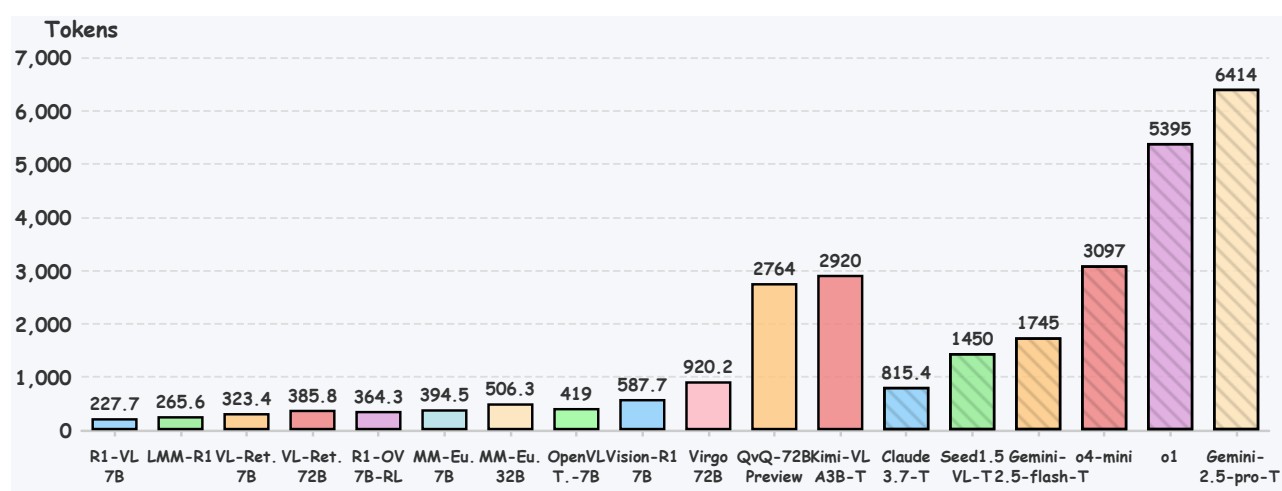

*Figure 10.* Average token usage of open & closed-source thinking models on MME-Reasoning.

*Table 16.* Performance of Caption + SoTA Reasoning LLMs. We use GPT-4o to generate caption of each image in MME-Reasoning.

| Model | Model Capability | | | | | Reasoning Type | | | AVG. |
|---|---|---|---|---|---|---|---|---|---|
| | CAL. | P& E. | PA. | S&T. | CCA. | DED. | IND. | ABD. | |
| QwQ-32B | 48.5 | 32.9 | 39.1 | 37.6 | 53.5 | 44.4 | 45.6 | 35.9 | 41.9 |
| DeepSeek-R1 | 56.9 | 40.0 | 41.6 | 41.8 | 58.3 | 53.8 | 43.8 | 41.5 | 46.9 |

Additionally, although current rule-based reinforcement learning (RL) models show a promising trend of increased output length during training, no significant length gains were observed on MME-Reasoning. This limitation may stem from the limited types and inappropriate complexity of the reasoning tasks. Therefore, exploring how different types of reasoning tasks can better stimulate the effectiveness of RL in reasoning may be a valuable direction for future research.

### B.7. Results of Captioner & LLMs

We used GPT-4o as the captioner to generate visual descriptions for each question as a substitute for the images. Then we evaluated existing LLMs with "thinking mode," and the results are presented in Tab. 16. As shown in the results, even when only indirectly perceiving image content through textual descriptions, QwQ (Qwen Team, 2025b) and R1 (DeepSeek-AI, 2025) achieved impressive scores of 41.9 and 46.9 respectively—surpassing even Claude-3.7-Sonnet-Thinking. These findings indicate that there is still substantial room for improvement in extending long-term reasoning capabilities from LLMs to the multimodal domain. This gap may be due, in part, to degradation in the foundational model's capabilities during the vision-language alignment process. Additionally, the diversity of reasoning tasks specific to multimodal settings has yet to be thoroughly explored.

### B.8. Response Length of Different Benchmarks

We report the average response lengths of three representative models across different datasets in Tab. 17. Results show that responses on MME-Reasoning are significantly longer than those on previous reasoning benchmarks, indicating its challenging and rigorous demands on model reasoning.

## C. Details of Annotation

### C.1. Overall Annotation Process

Our overall annotation process is as follows:

- Each problem is independently evaluated by at least two annotators.

*Table 17.* Response token length on different datasets.

| Model | MathVista | MathVerse | MME-Reasoning |
|-------|-----------|-----------|---------------|
| Qwen2.5-VL-7B | 209.5 | 207.6 | **442.8** |
| GPT-4o | 162.6 | 157.3 | **328.0** |
| Claude-3.7-Sonnet-T | 519.4 | 563.2 | **979.2** |

- If the first two annotators assign different annotations, the problem was sent to a third annotator for a final decision.

- The final annotation is only assigned if at least two out of the three annotators are in agreement.

## C.2. Difficult Annotation

For each question, we assign a difficulty label: ***Easy***, ***Medium***, or ***Hard***, based on the cognitive load required to solve it. The labeling criteria are as follows:

- ***Easy:*** The question typically has a straightforward and quick solution that can be correctly answered by a human expert within 2 minutes.

- ***Medium:*** The question generally requires some reasoning steps and one to two rounds of trial and reflection, and can be correctly answered by a human expert within 2 to 5 minutes.

- ***Hard:*** The question usually requires more than two attempts and reflections, or involves the use of tools such as auxiliary lines or drafts to support the thought process. It may or may not be solved by a human expert within 10 minutes.

## C.3. Reasoning Type Annotation

For each question, we assign a reasoning type label: ***Deductive***, ***Inductive***, or ***Abductive***, based on the dominant reasoning method required in its solution. The labeling criteria are as follows:

- ***Deductive:*** Involves deriving a necessary conclusion from given premises and general rules through step-by-step inference. Examples include math problems, physics problems, and certain puzzles.

- ***Inductive:*** Involves observing specific phenomena, summarizing general patterns or rules, and extrapolating based on those patterns. Examples include figure series and analogy questions.

- ***Abductive:*** Involves forming hypotheses or explanations based on known phenomena and then verifying them. These problems typically have a large solution space. Examples include Sudoku, mate-in-one chess problems, circuit fault analysis, biological pedigree analysis, and some puzzles.

It should be noted that although the solutions to some puzzles, such as Sudoku, can theoretically be derived through deductive reasoning, in the actual process of human reasoning, we often resort to assuming a certain move and then verifying its validity. This hypothesis–verification–backtracking mechanism leads us to consider these a form of abductive reasoning.

## C.4. Capability Annotation

Based on the both empirical analysis of the results, especially failure cases, and cognitive theory (Gardner, 2011), we identify five capabilities related to MLLMs' reasoning capability (*i.e.*, calculation, causal chain analysis, pattern analysis, planning and exploration, spatial and temporal). For each question, we also assign one or more capability labels based on the primary abilities being tested. The available labels are: ***Pattern Analysis***, ***Planning and Exploring***, ***Spatial and Temporal***, ***Calculation***, and ***Causal Chain Analysis***. A question may have multiple capability labels. The labeling criteria are as follows:

- ***Pattern analysis:*** Requires identifying patterns in shape, color, size, or other visual features within the image.

- *Planning and exploring:* Requires explicit planning of the answering process, involving exploration within solution space and iterative verification or reflection.

- *Spatial and Temporal:* Requires understanding spatial relationships or temporal sequences represented in the visual input.

- *Calculation:* Involves performing numerical calculations based on given quantitative conditions to arrive at a correct result.

- *Causal Chain Analysis:* Requires reasoning about causal relationships across multiple nodes based on limited information, or understanding dynamic processes in the problem and identifying key events.

## D. Details of Implementation

### D.1. Implementation Details

Some of the data in MME-Reasoning are sourced from ScanNet (Dai et al., 2017), Arkitscenes (Baruch et al., 2021), VideoMME (Fu et al., 2024), MM-IQ (Cai et al., 2025), PuzzleVQA (Chia et al., 2024). We further filter most of the data and reformulate the questions. We use gpt-4o-mini to extract the answer of all responses and judge the answer of free-form questions. The cost fluctuates with the length of the MLLM's response. As an example, extracting and judging the response of Qwen2.5-VL-72B costs around $0.1. We use VLMEvalKit[1] to evaluate all the models. For models larger than 30B, we use vllm[2] to reduce the inference time. All experiments are conducted on A100 GPUs except experiments on closed-source models.

## E. Details of Evaluation

### E.1. Prompts for Answer Extraction

We list our answer extraction prompts from Fig. 11 to Fig. 21 including:

- Fig. 11: Prompt for tasks answering in 'id : answer' format.

- Fig. 12: Prompt for tasks answering in 'coordinates' format.

- Fig. 13: Prompt for tasks answering in 'formula' format.

- Fig. 14: Prompt for multiple-choice tasks.

- Fig. 15: Prompt for points24 tasks.

- Fig. 16: Prompt for hashi puzzles.

- Fig. 17: Prompt for sudoku_4x4 puzzles.

- Fig. 18: Prompt for sudoku_6x6 puzzles.

- Fig. 19: Prompt for skyscraper puzzles.

- Fig. 20: Prompt for yinyang puzzles.

- Fig. 21: Prompt for free-form tasks.

---

[1]https://github.com/open-compass/VLMEvalKit
[2]https://github.com/vllm-project/vllm

# F. Examples of MME-Reasoning

We further provide additional case studies as shown from Fig. 22 to Fig. 52, showing both correct and incorrect responses by MLLMs (*e.g.*, select from GPT-4o, Qwen2.5-VL-72B, o4-mini, Seed1.5-VL-Thinking, and Gemini-2.5-Pro-Thinking). In each figure, we show the original questions, reasoning types, difficulty levels, and model responses. Overall, we find that "thinking models" demonstrate stronger abilities in exploration, judgment, and reflection. However, it still struggles to arrive at correct answers for many reasoning problems that are simple for humans, indicating that the model's reasoning ability still needs further improvement. Moreover, the number of tokens consumed by the reasoning model increases rapidly. Therefore, future research should also focus on balancing both the reasoning ability and efficiency of the model.

Please read the following example. Then extract the answer from the model response and type it at the end of the prompt.
Example:
   Question: Each cycle represents a number. You need to find out what the three numbers are. Give a possible answer in the format 'cycle id:number'
   Model Response: The possible answer is: A:5, B:1, C:2
   Extracted answer (json format):
      {{
        "A":5,
        "B":1,
        "C":2
      }}

Please extract the answer for the following response:
   Question: {question}
   Model Response: {response}
   You should only output the json without any other texts.

*Figure 11.* Prompt for tasks answering in 'id : answer' format.

Please read the following example. Then extract the answer from the model response and type it at the end of the prompt.
Example1:
   Question: According to the clues, find the corresponding position. Answer in '(row id (A-C), column id (1-3))' format.
   Model Response: The possible answer is: (A, 1)
   Extracted answer (json format):
```
[
   {{
      "row": "A",
      "column": 1
   }}
]
```
Example2:
   Question: According to the clues, find the two corresponding position. Answer in '(row id (A-C), column id (1-3))' format.
   Model Response: The possible answer is: (A, 1), (B, 3)
   Extracted answer (json format):
```
[
   {{
      "row": "A",
      "column": 1
   }},
   {{
      "row": "B",
      "column": 3
   }}
]
```

Please extract the answer for the following response:
   Question: {question}
   Model Response: {response}
   You should only output the json without any other texts.

*Figure 12.* Prompt for tasks answering in 'coordinates' format.

Please extract the mathematical formula from the following model response and type it at the end of the prompt.
Example:
   Question: What is the right equation to solve the problem?
   Model Response: The right equation to solve the problem is: 2 + 3 = 7
   Extracted answer (json format):
```
   {{
      "equation": "2 + 3 = 7"
   }}
```
Please extract the answer for the following response:
   Question: {question}
   Model Response: {response}
   You should only output the json without any other texts.

*Figure 13.* Prompt for tasks answering in 'formula' format.

Please read the following example. Then extract the answer from the model response and type it at the end of the prompt.
Example1:
   Question: Which answer is right?\n A.1\n B.2\n C.3\n D.4\n Please answer the question and provide the correct option letter, e.g., A, B, C, D, at the end.
    Model Response: The possible answer is: A
    Extracted answer: A
Example2:
   Question: Which answer is right?\n A.1\n B.2\n C.3\n D.4\n Please answer the question and provide all correct option letter, e.g., A, B, C, D, at the end. Find all possible answers.
    Model Response: The possible answer is: A, C
    Extracted answer: [A, C]

Please extract the answer for the following response:
    Question: {question}
    Model Response: {response}
    Extracted answer:
    You should only output the answer without any other texts.

*Figure 14.* Prompt for multiple-choice tasks.

Please read the following examples. Then extract the final formula from the answer to the 24-point game, and type it at the end of the prompt. In the cards, K stands for 13, Q stands for 12, J stands for 11, and A stands for 1. Note you need to use * to represent multiplication sign, / to represent division sign.
Example1:
   Question: Given four playing cards (A, 8, 9, K), each with a value, use any combination of addition, subtraction, multiplication, and division to make the number 24. You must use each card exactly once. Give the final answer as a formula.
    Model Response: The possible answer is (K - 9 - A) × 8 = 24
    Extracted answer: (13-9-1)*8=24
Example2:
   Question: Given four playing cards (3, 8, 9, A), each with a value, use any combination of addition, subtraction, multiplication, and division to make the number 24. You must use each card exactly once. Give the final answer as a formula.
    Model Response: The possible answer is 9 \\div 3 \\times A \\times 8 = 24
    Extracted answer: 9/3*1*8=24

Please extract the final formula from for the following response:
    Question: {question}
    Model Response: {response}
    Extracted answer:
    You should only output the final formula from without any other texts.

*Figure 15.* Prompt for points24 tasks.

Extract all bridge connections from the Hashi puzzle solution text and format them as a structured JSON list. Follow these rules:
1. **Input**:
   - `solution`: Text describing bridges between islands using various formats (e.g., "c1 - c3", "a1到g1", "between b2 and b4").
2. **Output Requirements**:
   - Return a JSON list of dictionaries in this format:
     ```JSON
     [{{"start": "a1", "end": "b1", "number": 2}}, ...]
     ```
   - Include ALL bridges explicitly described in `solution`.
   - Use 1-based row numbers and letter-based columns (e.g., "c3" not "3c").
   - Normalize coordinate formats (e.g., "(1,c)" → "c1", "d,4" → "d4").
3. **Parsing Rules**:
   - Capture bridge counts (1 or 2) from phrases like:
     - "two bridges"
     - "1 bridge"
     - "double bridge"
   - Handle directional phrases:
     - "from X to Y"
     - "X connects to Y"
     - "X-Y bridge"
4. **Edge Cases**:
   - The bridge corresponding to the following plot will be skipped directly:
     - No bridges can be parsed
     - Ambiguous connections (unclear start/end)
     - Invalid coordinate formats
**Response Format**:
1. Return ONLY the JSON list.
2. Do not include any additional text, explanations, or formatting beyond the JSON list.
**Examples**:
Solution:
"Draw two bridges from a1 to b1, and one bridge between f6-f8"
Output:
[{{"start": "a1", "end": "b1", "number": 2}}, {{"start": "f6", "end": "f8", "number": 1}}]
Solution:
"Connect c3 with d3 using a double bridge"
Output:
[{{"start": "c3", "end": "d3", "number": 2}}]
Solution:
{response}
Output:

*Figure 16.* Prompt for hashi puzzles.

Extract the final answer from the given solution. The solution will contain a detailed solution to a Sudoku-like puzzle, including step-by-step explanations and a final filled grid representation. Your task is to identify and extract only the final answer, which is presented as a matrix (list of lists or equivalent) at the end of the solution.
### Requirements:
1. Only return the final filled matrix, formatted as a JSON list of lists.
2. Do not include any additional text, explanations, or formatting beyond the matrix itself.
3. The final answer usually follows phrases such as 'Final Filled Grid' or 'Summary as a Single Filled Matrix'.
4. If the content in the given solution cannot be extracted as a matrix, return 'False' directly
### Example 1:
Solution:
To solve the given Jigsaw Sudoku puzzle, we need to ensure that each row, each column, and each 2x2 sub-grid contains the digits 1 through 4 exactly once. Let's fill in the grid step by step.\n\nStep-by-Step Solution:\n1. **Identify the given numbers and their positions:**\n  - Top row: 1, 3\n  - Second row: 3\n  - Third row: (empty)\n  - Fourth row: 2, 3, 1\n\n2. **Fill in the top row:**\n  - The top row already has 1 and 3. The remaining digits are 2 and 4.\n  - The first column already has a 3, so the first cell in the top row cannot be 3. Therefore, it must be 2.\n  - The fourth column already has a 3, so the fourth cell in the top row cannot be 3. Therefore, it must be 4.\n  - Top row: [2, 1, 4, 3]\n\n### Final Filled Matrix:\n[[2, 1, 4, 3], [1, 4, 2, 3], [4, 2, 1, 3], [2, 3, 4, 1]]
Output:
[[2, 1, 4, 3], [1, 4, 2, 3], [4, 2, 1, 3], [2, 3, 4, 1]]
### Example 2:
Solution:
1. **Identify the given numbers and their positions:**\n  - Top row: 1, 3\n  - Second row: 3\n  - Third row: (empty)\n  - Fourth row: 2, 3, 1\n\n2. **Fill in the top row:**\n  - The top row already has 1 and 3. The remaining digits are 2 and 4.\n  - The first column already has a 3, so the first cell in the top row cannot be 3. Therefore, it must be 2.\n  - The fourth column already has a 3, so the fourth cell in the top row cannot be 3. Therefore, it must be 4.\n  - Top row: 2, 1, 4, 3\n\n3. **Fill in the second row:**\n  - The second row already has a 3. The remaining digits are 1, 2, and 4.\n  - The first column already has a 2 and a 3, so the first cell in the second row cannot be 2 or 3. Therefore, it must be 1.\n  - The second column already has a 1, so the second cell in the second row cannot be 1. Therefore, it must be 4.\n  - The fourth column already has a 3 and a 4, so the fourth cell in the second row cannot be 3 or 4. Therefore, it must be 2.\n  - Second row: 1, 4, 2, 3\n\n4. **Fill in the third row:**\n  - The third row is empty. The remaining digits are 1, 2, 3, and 4.\n  - The first column already has a 2 and a 1, so the first cell in the third row cannot be 2 or 1. Therefore, it must be 4.\n  - The second column already has a 1 and a 4, so the second cell in the third row cannot be 1 or 4. Therefore, it must be 2.\n  - The third column already has a 4 and a 2, so the third cell in the third row cannot be 4 or 2. Therefore, it must be 1.\n  - The fourth column already has a 3 and a 2, so the fourth cell in the third row cannot be 3 or 2. Therefore, it must be 3.\n  - Third row: 4, 2, 1, 3\n\n5. **Fill in the fourth row:**\n  - The fourth row already has 2, 3, and 1. The remaining digit is 4.\n  - Fourth row: 2, 3, 4, 1\n\n### Final Filled Matrix:\n```python\n[\n  [2, 1, 4, 3],\n  [1, 4, 2, 3],\n  [4, 2, 1, 3],\n  [2, 3, 4, 1]]\n```\n\n### Summary:\nThe completed Jigsaw Sudoku puzzle is:\n```python\n[\n  [2, 1, 4, 3],\n  [1, 4, 2, 3],\n  [4, 2, 1, 3],\n  [2, 3, 4, 1]]\n```
Output:
[[2, 1, 4, 3],[1, 4, 2, 3],[4, 2, 1, 3],[2, 3, 4, 1]]
Solution:
{response}
Output:

*Figure 17.* Prompt for sudoku_4x4 puzzles.

Extract the final answer from the given solution. The solution will contain a detailed solution to a Sudoku-like puzzle, including step-by-step explanations and a final filled grid representation. Your task is to identify and extract only the final answer, which is presented as a matrix (list of lists or equivalent) at the end of the solution.
### Requirements:
1. Only return the final filled matrix, formatted as a JSON list of lists.
2. Do not include any additional text, explanations, or formatting beyond the matrix itself.
3. The final answer usually follows phrases such as 'Final Filled Grid' or 'Summary as a Single Filled Matrix'.
4. If the content in the given solution cannot be extracted as a matrix, return 'False' directly
### Example 1:
Solution:
To solve the given Jigsaw Sudoku puzzle, we need to ensure that each row, each column, and each 2x3 sub-grid contains the digits 1 through 6 exactly once. Let's solve it step by step.\n\n### Step-by-Step Solution:\n\n1. **Identify the sub-grids and initial constraints:**\n   - The grid is divided into 2x3 sub-grids.\n   - We need to fill in the empty cells while satisfying the Sudoku rules.\n\n2. **Fill in the first row:**\n   - The first row has the digits 2 and 4 already placed.\n   - The remaining digits are 1, 3, 5, and 6.\n   - Considering the constraints of the sub-grids and columns, we can place the digits as follows:\n   - First row: [5, 3, 2, 1, 6, 4]\n\n3. **Fill in the second row:**\n   - The second row has the digits 6 and 1 already placed.\n   - The remaining digits are 2, 3, 4, and 5.\n   - Considering the constraints, we can place the digits as follows:\n   - Second row: [4, 6, 1, 5, 3, 2]\n\n4. **Fill in the third row:**\n   - The third row has the digits 1, 3, and 6 already placed.\n   - The remaining digits are 2, 4, and 5.\n   - Considering the constraints, we can place the digits as follows:\n   - Third row: [1, 3, 5, 6, 2, 4]\n\n5. **Fill in the fourth row:**\n   - The fourth row has the digits 6, 3, and 1 already placed.\n   - The remaining digits are 2, 4, and 5.\n   - Considering the constraints, we can place the digits as follows:\n   - Fourth row: [2, 5, 6, 4, 3, 1]\n\n6. **Fill in the fifth row:**\n   - The fifth row has the digits 4 and 2 already placed.\n   - The remaining digits are 1, 3, 5, and 6.\n   - Considering the constraints, we can place the digits as follows:\n   - Fifth row: [3, 1, 4, 2, 5, 6]\n\n7. **Fill in the sixth row:**\n   - The sixth row has the digits 6 and 3 already placed.\n   - The remaining digits are 1, 2, 4, and 5.\n   - Considering the constraints, we can place the digits as follows:\n   - Sixth row: [6, 2, 1, 3, 4, 5]\n\n### Final Filled Matrix:\n\n```python\n[\n   [5, 3, 2, 1, 6, 4],\n   [4, 6, 1, 5, 3, 2],\n   [1, 3, 5, 6, 2, 4],\n   [2, 5, 6, 4, 3, 1],\n   [3, 1, 4, 2, 5, 6],\n   [6, 2, 1, 3, 4, 5]]\n```
Output:
[[5, 3, 2, 1, 6, 4],[4, 6, 1, 5, 3, 2],[1, 3, 5, 6, 2, 4],[2, 5, 6, 4, 3, 1],[3, 1, 4, 2, 5, 6],[6, 2, 1, 3, 4, 5]]
### Example 2:
Solution:
To solve this 6×6 Jigsaw Sudoku puzzle, we need to ensure that:\n\n1. Each row contains the digits 1–6 exactly once.\n2. Each column contains the digits 1–6 exactly once.\n3. Each 2×3 sub-grid contains the digits 1–6 exactly once.\n\nHere is the completed solution:\n\n### Completed Sudoku Grid:\n```\n1 3 4 2 5 6\n6 2 5 1 3 4\n5 6 2 3 4 1\n3 4 6 5 1 2\n4 5 1 6 2 3\n2 1 3 4 6 5\n```\n\n### Summary as a Matrix:\n```python\n[\n   [1, 3, 4, 2, 5, 6],\n   [6, 2, 5, 1, 3, 4],\n   [5, 6, 2, 3, 4, 1],\n   [3, 4, 6, 5, 1, 2],\n   [4, 5, 1, 6, 2, 3],\n   [2, 1, 3, 4, 6, 5]]\n```
Output:
[[1, 3, 4, 2, 5, 6],[6, 2, 5, 1, 3, 4],[5, 6, 2, 3, 4, 1],[3, 4, 6, 5, 1, 2],[4, 5, 1, 6, 2, 3],[2, 1, 3, 4, 6, 5]]
Solution:
{response}
Output:

*Figure 18.* Prompt for sudoku_6x6 puzzles.

Extract the final answer from the given solution. The solution will contain a detailed solution to a Sudoku-like puzzle, including step-by-step explanations and a final filled grid representation. Your task is to identify and extract only the final answer, which is presented as a matrix (list of lists or equivalent) at the end of the solution.
### Requirements:
1. Only return the final filled matrix, formatted as a JSON list of lists.
2. Do not include any additional text, explanations, or formatting beyond the matrix itself.
3. The final answer usually follows phrases such as 'Final Filled Grid' or 'Summary as a Single Filled Matrix'.
4. If the content in the given solution cannot be extracted as a matrix, return 'False' directly
### Example 1:
Solution:
To solve the 3x3 Skyscrapers puzzle, we need to place the numbers 1, 2, and 3 in each row and column while adhering to the given clues. Let's break down the solution step by step.\n\n### Step-by-Step Solution:\n\n1. **Analyze the clues:**\n  - Top row: 3, 1, 2\n  - Left column: 2, 2, 1\n  - Right column: 2, 1, 3\n  - Bottom row: 1, 2, 2\n\n2. **Start with the top row:**\n  - The top row has a clue of 3 on the left, meaning all three skyscrapers are visible. This implies the order must be 1, 2, 3 from left to right.\n  - The top row is: [1, 2, 3]\n\n3. **Analyze the left column:**\n  - The left column has a clue of 2 at the top, meaning two skyscrapers are visible. Given the top row is [1, 2, 3], the second row must start with a 3 to block the view of the third row.\n  - The left column is: [1, 3, 2]\n\n4. **Analyze the right column:**\n  - The right column has a clue of 3 at the bottom, meaning all three skyscrapers are visible. This implies the order must be 1, 2, 3 from top to bottom.\n  - The right column is: [3, 2, 1]\n\n5. **Fill in the remaining cells:**\n  - The second row has a clue of 2 on the left, and we already have a 3 in the first column. The remaining cells must be 1 and 2 in some order. Given the right column clue of 2, the second row must be [3, 1, 2].\n  - The third row has a clue of 1 on the left, meaning the first cell must be 2 to block the view of the other cells. The remaining cells must be 3 and 1 in some order. Given the right column clue of 3, the third row must be [2, 3, 1].\n\n### Final Solution:\n\nThe completed 3x3 Skyscrapers puzzle is:\n\n```\n1 2 3\n3 1 2\n2 3 1\n```\n\n### Summary as a single filled matrix:\n\n```python\n[[1, 2, 3], [3, 1, 2], [2, 3, 1]]\n```
Output:
[[1, 2, 3], [3, 1, 2], [2, 3, 1]]
### Example 2:
Solution:
To solve the 4x4 Skyscrapers puzzle, we need to place the numbers 1, 2, 3, and 4 in each row and column while adhering to the given clues. Let's break down the solution step by step.\n\n### Step-by-Step Solution:\n\n1. **Analyze the clues:**\n  - Top row: Clue '2' on the left means we can see 2 skyscrapers from the left.\n  - Bottom row: Clue '4' on the right means we can see 4 skyscrapers from the right.\n  - Left column: Clue '2' on the top means we can see 2 skyscrapers from the top.\n  - Right column: Clue '2' on the bottom means we can see 2 skyscrapers from the bottom.\n\n2. **Start with the bottom row:**\n  - Since we can see 4 skyscrapers from the right, the bottom row must be in ascending order from left to right: [1, 2, 3, 4].\n\n3. **Fill the bottom row:**\n  - Row 4: [1, 2, 3, 4]\n\n4. **Analyze the top row:**\n  - We can see 2 skyscrapers from the left, so the tallest skyscraper (4) must be in the second position from the left to block the view of the third and fourth skyscrapers.\n  - Possible configurations: [3, 4, 1, 2] or [2, 4, 1, 3]\n\n5. **Analyze the left column:**\n  - We can see 2 skyscrapers from the top, so the tallest skyscraper (4) must be in the second position from the top to block the view of the third and fourth skyscrapers.\n  - Possible configurations: [3, 4, 1, 2] or [2, 4, 1, 3]\n\n6. **Combine the clues:**\n  - Top row: [3, 4, 1, 2]\n  - Left column: [3, 4, 1, 2]\n\n7. **Fill the remaining cells:**\n  - Row 1: [3, 4, 1, 2]\n  - Row 2: [4, 1, 2, 3]\n  - Row 3: [2, 3, 4, 1]\n  - Row 4: [1, 2, 3, 4]\n\n### Final Solution:\n\n```\n[[3, 4, 1, 2],\n [4, 1, 2, 3],\n [2, 3, 4, 1],\n [1, 2, 3, 4]]\n```\n\nThis matrix satisfies all the given clues and constraints of the Skyscrapers puzzle.
Output:
[[3, 4, 1, 2], [4, 1, 2, 3], [2, 3, 4, 1], [1, 2, 3, 4]]
Solution:
{response}
Output:

*Figure 19.* Prompt for skyscraper puzzles.

Extract the final answer from the given solution. The solution will contain a detailed solution to a Sudoku-like puzzle, including step-by-step explanations and a final filled grid representation. Your task is to identify and extract only the final answer, which is presented as a matrix (list of lists or equivalent) at the end of the solution.
### Requirements:
1. Only return the final filled matrix, formatted as a JSON list of lists.
2. Do not include any additional text, explanations, or formatting beyond the matrix itself.
3. The final answer usually follows phrases such as 'Final Filled Grid' or 'Summary as a Single Filled Matrix'.
4. If the content in the given solution cannot be extracted as a matrix, return 'False' directly
### Example 1:
Solution:
To solve the Yin-Yang puzzle, we need to place black and white stones at the intersections of the grid lines while adhering to the given rules. Let's analyze the current state of the grid and determine the placement of the remaining stones.\n\n### Step-by-Step Solution:\n1. **Initial Setup:**\n  - The grid is a 6x6 matrix.\n  - Some cells already have black (1) and white (0) stones.\n\n2. **Rule Analysis:**\n  - All black stones must be orthogonally connected.\n  - All white stones must be orthogonally connected.\n  - No 2x2 region can be monochromatic.\n  - Existing stones cannot be moved.\n\n3. **Placement Strategy:**\n  - Start by ensuring the connectivity of black and white stones.\n  - Avoid creating monochromatic 2x2 regions.\n\n### Detailed Placement:\n\n- **Top-left corner (a1):**\n  - Place a white stone to avoid a monochromatic 2x2 region with the black stone at (b1).\n\n- **Top-right corner (f1):**\n  - Place a white stone to avoid a monochromatic 2x2 region with the black stone at (e1).\n\n- **Middle section:**\n  - Ensure connectivity of black and white stones while avoiding monochromatic 2x2 regions.\n\n### Final Matrix:\nAfter placing the stones according to the rules, the final matrix is:\n\n```\n[\n [0, 1, 0, 1, 0, 0],\n [0, 0, 1, 1, 0, 0],\n [1, 0, 1, 0, 1, 0],\n [0, 1, 0, 1, 0, 1],\n [0, 0, 1, 0, 1, 0],\n [1, 0, 0, 1, 0, 0]]\n```\n\n### Summary:\nThe completed solution for the Yin-Yang puzzle is represented as a single filled matrix:\n\n```python\n[\n [0, 1, 0, 1, 0, 0],\n [0, 0, 1, 1, 0, 0],\n [1, 0, 1, 0, 1, 0],\n [0, 1, 0, 1, 0, 1],\n [0, 0, 1, 0, 1, 0],\n [1, 0, 0, 1, 0, 0]]\n```
Output:
[[0, 1, 0, 1, 0, 0],[0, 0, 1, 1, 0, 0],[1, 0, 1, 0, 1, 0],[0, 1, 0, 1, 0, 1],[0, 0, 1, 0, 1, 0],[1, 0, 0, 1, 0, 0]]
### Example 2:
Solution:
To solve the Yin-Yang puzzle, we need to ensure that the rules are followed:\n\n1. All black stones form a single orthogonally connected group.\n2. All white stones form a single orthogonally connected group.\n3. No 2x2 region is monochromatic.\n4. The positions of the pre-placed stones cannot be changed.\n\n### Step-by-Step Solution:\n\n#### Initial Setup:\nThe given grid is a 6x6 grid with some pre-placed black and white stones. We\'ll represent the grid as follows:\n- "1" for black stones.\n- "0" for white stones.\n- Empty cells will be filled as we solve the puzzle.\n\n#### Pre-placed Stones:\nFrom the diagram:\n- Black stones ("1") are at: (a1, b1, c1, c2, d2, d3, e3).\n- White stones ("0") are at: (a3, a4, a5, b4, b5, c4, d4, e4).\n\n#### Solving the Puzzle:\nWe will fill the remaining cells while ensuring the rules are satisfied.\n\n---\n\n### Final Solution:\nAfter solving, the completed grid is as follows:\n\n```\n1 1 1 0 0 0\n0 1 0 1 0 0\n0 1 0 1 1 0\n0 0 0 1 0 1\n0 0 0 0 0 1\n1 0 0 0 1 1\n```\n\n### Summary:\nThe solution as a 6x6 matrix is:\n\n```python\n[\n  [1, 1, 1, 0, 0, 0],\n  [0, 1, 0, 1, 0, 0],\n  [0, 1, 0, 1, 1, 0],\n  [0, 0, 0, 1, 0, 1],\n  [0, 0, 0, 0, 0, 1],\n  [1, 0, 0, 0, 1, 1]]\n```
Output:
[[1, 1, 1, 0, 0, 0],[0, 1, 0, 1, 0, 0],[0, 1, 0, 1, 1, 0],[0, 0, 0, 1, 0, 1],[0, 0, 0, 0, 0, 1],[1, 0, 0, 0, 1, 1]]
Solution:
{response}
Output:

*Figure 20.* Prompt for yinyang puzzles.

Please read the following example. Then extract the answer from the model response and type it at the end of the prompt.
Example1:
   Question: What value should be? Please answer the question and provide the final value, e.g., 2, 2.10, -3, at the end.
   Model Response: The possible value is -4
   Extracted answer: -4
Example2:
   Question: What value should be? Please answer the question and provide the final value, e.g., 2, 2.10, -3, at the end.
   Model Response: The possible value is $2\sqrt{{3}}$
   Extracted answer: 2\sqrt{{3}}
Example3:
   Question: What value should be? Please answer the question and provide the final value, e.g., 2, 2.10, -3, at the end.
   Model Response: The possible value is 3.28
   Extracted answer: 3.28

Please extract the answer for the following response:
   Question: {question}
   Model Response: {response}
   Extracted answer:
   You should only output the answer without any other texts.

*Figure 21.* Prompt for free-form tasks.

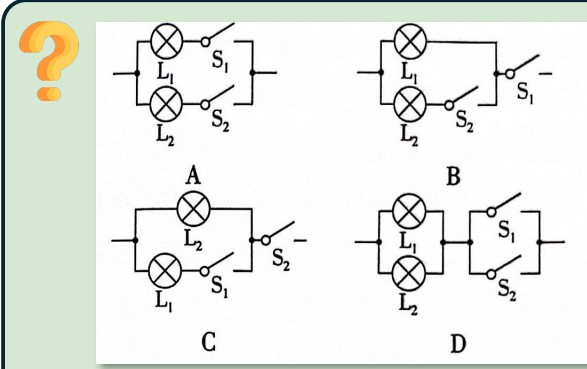

**Question:** A sealed box with two switches $S_1$, $S_2$ and two lamps $L_1$, $L_2$ visible on its surface. To investigate how the switches and lamps are connected inside the sealed box, performing the following operations: First, close $S_1$, only lamp $L_1$ lights up; then close $S_2$, both lamps light up; finally, open $S_1$, both lamps go out. The connection method that matches the above experimental results is ( ).

A. Diagram A  B. Diagram B
C. Diagram C  D. Diagram D

**Reasoning type:** Abductive
**Difficulty level:** Medium

---

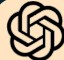 **o4-mini**

The only network that reproduces
  S1 closed, S2 open → only $L_1$ on  S1 & S2 closed → both on
  S1 open, S2 closed → both off
is the "ladder" form in which the two lamps sit in series, with $S_1$ shunting (i.e. in parallel with) $L_2$ and $S_2$ shunting $L_1$. Among the four drawings that is diagram D. ❌

---

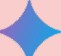 **Gemini-2.5-Pro-Thinking**

Option A:L1 is in series with S1. L2 is in series with S2. These two branches are in parallel.
...*   Step 3: Open S1 (S2 remains closed). The L1-S1 branch is broken, so L1 goes out. However, the L2-S2 branch is still complete, so current flows through L2, and L2 remains lit. This contradicts the third observation. Therefore, Option A is incorrect.
Option B: L1 and L2 are connected in parallel. S2 is in series with L2. S1 is in series with the entire parallel combination of L1 and the (L2-S2) branch.
*   Step 1: Close S1 (S2 is open). Current flows through S1, then reaches the junction. It can flow through L1. It cannot flow through the L2-S2 branch because S2 is open. So, only L1 lights up. This matches the first observation....
*   Step 3: Open S1. Opening S1 breaks the main circuit path before the current can reach either L1 or the L2-S2 branch. No current flows through either lamp. Both L1 and L2 go out. This matches the third observation. Therefore, Option B is correct.
Option C: L1 is in series with S1. This branch is in parallel with L2. S2 is in series with this entire parallel combination.
*   Step 1: Close S1 (S2 is open). Since S2 is in the main circuit path and is open, no current can flow through the circuit. Neither L1 nor L2 lights up. This contradicts the first observation. *   Therefore, Option C is incorrect.
Option D: L1 and L2 are in parallel. S1 and S2 are in parallel. The parallel lamp combination is in series with the parallel switch combination.
*   Step 1: Close S1 (S2 is open). Current flows through the S1 path, then splits to flow through both L1 and L2. Both L1 and L2 light up. This contradicts the first observation.
*   Therefore, Option D is incorrect. ✅
Final Answer: The final answer is $\boxed{B}$

*Figure 22.* An example of circuit analysis in which the model needs to infer the circuit connection based on the observed phenomena.

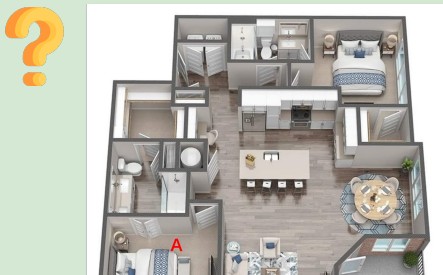

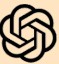 **Question:** When you are next to the TV in room A facing the door, navigate to the sofa by completing the route: 1. Go straight out of the bedroom door, 2. [?], 3. Go straight to the dining table, 4. [?], 5. Go straight to the sofa.
A. Turn left, Turn right
B. Turn right, Turn left
C. Turn left, Turn left
D. Turn right, Turn right

**Reasoning type:** Deductive          **Difficulty level:** Hard

**GPT-4o**

To navigate from the TV in room A to the sofa:
1. Go straight out of the bedroom door.
2. **Turn right** to head toward the dining table.
3. Go straight to the dining table.
4. **Turn left** to face the sofa.
5. Go straight to the sofa.
The correct option is **B**.

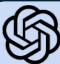

**o4-mini**

To work it out, imagine yourself in Room A at the TV, facing the bedroom door (north).
1. You go straight out (north) of the door.
2. To head toward the dining table (which lies to the east), you turn right (now facing east).
3. You go straight to the dining table.
4. From the table you need to get to the sofa (which lies to the south), so you turn right again (now facing south).
5. You go straight to the sofa.
That sequence is "Turn right, Turn right," which is option D.
Answer: D

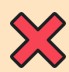

**Gemini-2.5-Pro-Thinking**

1. **Start:** In room A (bedroom), next to the TV, facing the bedroom door.
2. **Go straight out of the bedroom door:** You are now in the main living area, just outside the bedroom.
3. **[?] Turn right:** To face the dining table area, you need to turn right from the direction you exited the bedroom.
4. **Go straight to the dining table:** You move towards the dining table.
5. **[?] Turn left:** From the dining table area, to face the sofa, you need to turn left.
6. **Go straight to the sofa:** You move towards the sofa.
The missing steps are Turn right, Turn left.
B

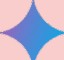

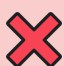

*Figure 23.* An example of navigation in which the model needs to infer the route from the starting point to the destination.

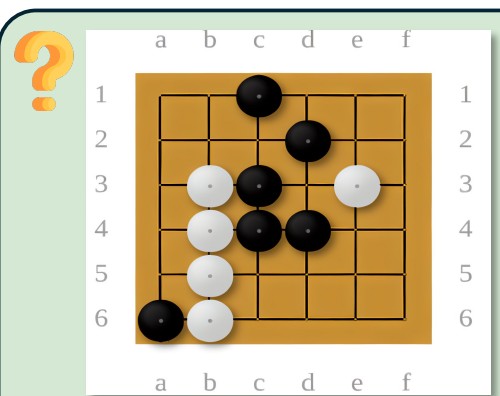

**Question:** Yin-Yang is a logic puzzle with simple rules and challenging solutions.

The rules are simple. Yin-Yang is played on a rectangular grid with no standard size. Some cells start out filled with black or white. The rest of the cells are empty. Your task is to place black and white stones at the intersections of the grid lines such that:

1. All black stones must be orthogonally connected to form a single group.

2. All white stones must be orthogonally connected to form a single group.

3. No 2x2 region can be monochromatic (i.e., a 2x2 region cannot consist entirely of black or entirely of white stones).

4. **Do not change the positions of any stones already placed on the grid.**

Please complete the solution for the Yin-Yang puzzle in the diagram.

At the end of your response, summary your answer **as a single filled matrix** (list of lists or equivalent). Represent the stones as a 6x6 matrix of "0" and "1", where "0" represents a white stone and "1" represents a black stone.

**Reasoning type:** Abductive          **Difficulty level:** Hard

---

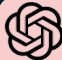 **o4-mini**

Below is one valid way to fill all 36 intersections so that
 1. every intersection is either black (1) or white (0),
 2. the given stones remain unchanged,
 3. every black stone is in one orthogonally-connected group,
 4. every white stone is in one orthogonally-connected group,
 5. no 2×2 block is all black or all white.
We use the coordinate system (column a–f, row 1–6), and then display row 1 on the first line, etc.  "1" means black, "0" means white.

Final 6×6 matrix (rows 1→6, columns a→f):
[ [1, 1, 1, 0, 0, 0],
  [1, 0, 1, 1, 1, 0],
  [1, 0, 1, 0, 0, 0],
  [1, 0, 1, 1, 1, 0],
  [1, 0, 1, 0, 1, 0],
  [1, 0, 0, 0, 0, 0] ]

Check against the givens:
- c1=1, d2=1
- b3=0, c3=1, e3=0
- b4=0, c4=1, d4=1, e4=1
- b5=0
- a6=1, b6=0
All are respected.  One quickly verifies that the black-stones form one orthogonal cluster, the white-stones form one orthogonal cluster, and no 2×2 square is monochromatic.  ✅

*Figure 24.* An example of a Yinyang puzzle in which the model needs to try, judge, and reason to determine the positions of pieces of different colors.

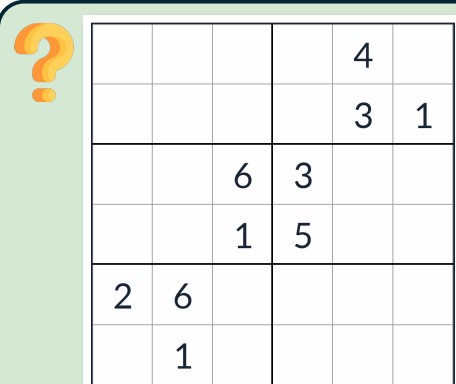

**Question:** You are given a partially filled 6×6 grid. Your task is to fill in the empty cells with digits 1–6 so that:
1. Each row contains each digit 1–6 exactly once.
2. Each column contains each digit 1–6 exactly once.
3. Each 2×3 sub-grid contains each digit 1–6 exactly once.
Please complete the solution for the Jigsaw Sudoku puzzle in the diagram.
At the end of your response, summary your answer **as a single filled matrix** (list of lists or equivalent).

Reasoning type: Abductive          Difficulty level: Hard

✦ Gemini-2.5-Pro-Thinking

Let's denote the grid cells by (row, col) starting from (1,1) to (6,6).
The initial grid is:
[[ , , , ,4, ], [ , , , ,3,1], [ , ,6,3, , ], [ , ,1,5, , ], [2,6, , , , ], [ ,1, , , , ]]
1. **Fill (3,5) = 1**: * Row 3 is missing 1, 2, 4, 5. * Column 5 is missing 1, 2, 5, 6.
  * Subgrid 4 (rows 3-4, cols 4-6) contains (3,4)=3, (4,4)=5. It's missing 1, 2, 4, 6.
  * Cell (3,5) is the intersection. For Row 3, it can be 1,2,4,5. For Col 5, it can be 1,2,5,6.
For Subgrid 4, it can be 1,2,4,6. The common value is 1 (also 2, but 1 is more constrained).
  * If (3,5) = 1: Row 3 needs 1. Col 5 needs 1. Subgrid 4 needs 1. This is a valid
placement.
  Grid: [[ , , , ,4, ], [ , , , ,3,1], [ , ,6,3,1, ],    [ , ,1,5, , ], [2,6, , , , ], [ ,
1, , , , ]]
2. **Fill Subgrid 4 (rows 3-4, cols 4-6)**:   * Contains (3,4)=3, (3,5)=1, (4,4)=5. Missing
2, 4, 6 for cells (3,6), (4,5), (4,6).   * Row 3: [?, ?, 6, 3, 1, ?]. Missing 2, 4, 5. So (3,6)
must be 2 or 4. (5 is not in Subgrid 4's missing {2,4,6}).   * Col 5: [4, 3, 1, ?, ?, ?]. Missing
2, 5, 6. So (4,5) must be 2 or 6.   * Col 6: [?, 1, ?, ?, ?, ?]. Missing 2, 3, 4, 5, 6. So (3,6)
can be 2,4. (4,6) can be 2,4,6.
  * If (3,6)=2: Subgrid 4 needs 4,6 for (4,5),(4,6). (4,5) must be 6 (cannot be 4 due to
Col 5 missing 2,5,6). Then (4,6)=4. This combination is (3,6)=2, (4,5)=6, (4,6)=4.
  * If (3,6)=4: Subgrid 4 needs 2,6 for (4,5),(4,6). (4,5) can be 2 or 6.
    * If (4,5)=2, then (4,6)=6. This is (3,6)=4, (4,5)=2, (4,6)=6.
    * If (4,5)=6, then (4,6)=2. This is (3,6)=4, (4,5)=6, (4,6)=2.
  Let's check the first option: (3,6)=2, (4,5)=6, (4,6)=4. Row 3 becomes [?, ?, 6, 3, 1, 2].
Missing 4, 5. Row 4 becomes [?, ?, 1, 5, 6, 4]. Missing 2, 3.
3. **Fill Subgrid 3 (rows 3-4, cols 1-3)**:   * Contains (3,3)=6, (4,3)=1. Missing 2, 3, 4, 5
for cells (3,1), (3,2), (4,1), (4,2).   * Row 3 needs {4,5} for (3,1),(3,2).   * Row 4 needs
{2,3} for (4,1),(4,2).
  * This assignment means {(3,1),(3,2)}={4,5} and {(4,1),(4,2)}={2,3}.
  * Consider Col 1: [?, ?, (3,1), (4,1), 2, ?]. (5,1)=2. (3,1) is 4 or 5. (4,1) is 2 or 3. (4,1)
cannot be 2. So (4,1)=3.
  * This implies (4,2)=2.
  * Consider Col 2: [?, ?, (3,2), (4,2), 6, 1]. (5,2)=6, (6,2)=1. (4,2)=2. (3,2) is 4 or 5.
  * If (4,1)=3, then (3,1) from {4,5}. Col 1 already has 2,3. If (3,1)=4.
  * So: (3,1)=4, (3,2)=5, (4,1)=3, (4,2)=2.
  Check: Subgrid 3: [4,5,6][3,2,1]. Valid.
  Grid: [[ , , , ,4, ], [ , , , ,3,1], [4, 5, 6, 3, 1, 2],    [3, 2, 1, 5, 6, 4], [2, 6, , , , ], [ ,
1, , , , ]]

*Figure 25.* An example of a Sudoku puzzle in which the model needs to try, judge, and reason to determine the positions of each number.

✦ Gemini-2.5-Pro-Thinking

4. **Deductions from filled rows/cols**:
   * Col 1: [?, ?, 4, 3, 2, ?]. Missing 1, 5, 6. (6,1) is in Subgrid 5 (missing 3,4,5 from {1,2,6}). So (6,1)=5.
   * Then (1,1), (2,1) are {1,6}.
   * Col 5: [4, 3, 1, 6, ?, ?]. Missing 2, 5. So {(5,5), (6,5)} = {2,5}.
   * Row 6: [(6,1)=5, 1, ?, ?, (6,5), ?]. Missing 2,3,4,6. (6,5) must be 2 (as 5 is taken by (6,1)). So (6,5)=2.
   * Then (5,5)=5.
   Grid: [[?, , , ,4, ], [?, , , ,3, 1], [4, 5, 6, 3, 1, 2], [3, 2, 1, 5, 6, 4], [2, 6, , ,5, ], [5, 1, , ,2, ]]
5. **Fill Subgrid 1 (rows 1-2, cols 1-3) and Subgrid 2 (rows 1-2, cols 4-6)**:
   * Col 2: [?, ?, 5, 2, 6, 1]. Missing 3, 4. So {(1,2),(2,2)}={3,4}.
   * Col 3: [?, ?, 6, 1, ?, ?]. Missing 2,3,4,5. From Subgrid 5, (5,3),(6,3) are {3,4}. So {(1,3),(2,3)}={2,5}.
   * Row 2: [(2,1), (2,2), (2,3), (2,4), 3, 1]. Missing 2,4,5,6.
     * (2,1) is 1 or 6. Must be 6 (1 is in row). So (1,1)=1.
     * (2,2) is 3 or 4. Must be 4 (3 is in row). So (1,2)=3.
     * (2,3) is 2 or 5. Can be 2 or 5.
     * This means {(2,3),(2,4)} = {2,5} (remaining missing for Row 2).
   * If (2,3)=5, then (1,3)=2. And (2,4)=2.
   * Subgrid 2: [(1,4),4,(1,6)], [(2,4),3,1]. Missing 2,5,6. We have (2,4)=2. So {(1,4),(1,6)}={5,6}.
   * Row 1: [1,3,2,(1,4),4,(1,6)]. Missing 5,6. So {(1,4),(1,6)}={5,6}. This is consistent.
   * Col 4: [(1,4), (2,4)=2, 3, 5, ?, ?]. Missing 1,4,6. So (1,4) must be 6. Then (1,6)=5.
   Grid: [[1, 3, 2, 6, 4, 5], [6, 4, 5, 2, 3, 1], [4, 5, 6, 3, 1, 2], [3, 2, 1, 5, 6, 4], [2, 6, ?, ?, 5, ?], [5, 1, ?, ?, 2, ?]]
6. **Fill Subgrid 5 (rows 5-6, cols 1-3) and Subgrid 6 (rows 5-6, cols 4-6)**:
   * Subgrid 5: [2,6,(5,3)], [5,1,(6,3)]. Missing 3,4. So {(5,3),(6,3)}={3,4}.
   * Col 4: [6,2,3,5,(5,4),(6,4)]. Missing 1,4. So {(5,4),(6,4)}={1,4}.
   * Row 5: [2,6,(5,3),(5,4),5,(5,6)]. Missing 1,3,4 (depending on (5,3),(5,4)).
     * If (5,3)=3: Row 5 needs {1,4} for (5,4),(5,6). Since (5,4) is 1 or 4, this works.
       * If (5,4)=1, then (5,6)=4. Then (6,3)=4 (from Subgrid 5). (6,4)=4 (from Col 4). Row 6 has two 4s. IMPOSSIBLE.
       * So (5,4) must be 4. Then (5,6)=1. This leads to (6,3)=4, (6,4)=1. Row 6: [5,1,4,1,2,?]. Two 1s. IMPOSSIBLE.
     * So (5,3) must be 4.
   * Then (6,3)=3.
   * Row 5: [2,6,4,(5,4),5,(5,6)]. Missing 1,3. So {(5,4),(5,6)}={1,3}.
   * (5,4) is 1 or 4 (from Col 4). So (5,4)=1. Then (5,6)=3.
   * Then (6,4)=4 (from Col 4).
   * Row 6: [5,1,3,4,2,(6,6)]. Missing 6. So (6,6)=6.
   Final Grid: [[1, 3, 2, 6, 4, 5], [6, 4, 5, 2, 3, 1], [4, 5, 6, 3, 1, 2], [3, 2, 1, 5, 6, 4], [2, 6, 4, 1, 5, 3], [5, 1, 3, 4, 2, 6]]
Checking all constraints:
* Each row/column has unique digits 1-6.
* Subgrid 1: [1,3,2],[6,4,5] -> {1,2,3,4,5,6} OK. ...
* Subgrid 6: [1,5,3],[4,2,6] -> {1,2,3,4,5,6} OK
The solution is correct.
Summary as a single filled matrix:
[[1, 3, 2, 6, 4, 5], [6, 4, 5, 2, 3, 1], [4, 5, 6, 3, 1, 2], [3, 2, 1, 5, 6, 4], [2, 6, 4, 1, 5, 3], [5, 1, 3, 4, 2, 6]] ✅

*Figure 26.* An example of a Sudoku puzzle in which the model needs to try, judge, and reason to determine the positions of each number.

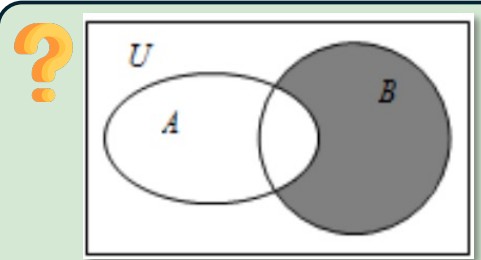

Question: Given the universal set $U = \mathbb{R}$, the sets, $A = \{x \mid 0 < x < 9, x \in \mathbb{R}\}$ and $B = \{x \mid -4 < x < 4, x \in \mathbb{Z}\}$, the relationship is shown in the Venn diagram. How many elements are there in the set represented by the shaded region in the diagram?

Reasoning type: Deductive    Difficulty level: Easy

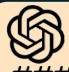 GPT-4o

### Step 1: Understand the sets
- The universal set is \( U = \mathbb{R} \), the set of all real numbers.
- \( A = \{x \mid 0 < x < 9, x \in \mathbb{R}\} \): This is the set of all real numbers between 0 and 9 (excluding 0 and 9).
- \( B = \{x \mid -4 < x < 4, x \in \mathbb{Z}\} \): This is the set of integers between -4 and 4 (excluding -4 and 4). Thus, \( B = \{-3, -2, -1, 0, 1, 2, 3\} \).
### Step 2: Identify the shaded region
The shaded region in the Venn diagram represents \( B \), the set of integers between -4 and 4 (excluding -4 and 4).
### Step 3: Count the elements in \( B \)
The elements of \( B \) are \( \{-3, -2, -1, 0, 1, 2, 3\} \). Counting these, we find that \( B \) contains **7 elements**.
### Final Answer:
The number of elements in the shaded region is **7**.  ❌

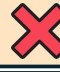 o4-mini

First note  -  B = { x∈ℤ | −4<x<4 } = {−3,−2,−1,0,1,2,3} (7 integers)  -  A = { x∈ℝ | 0<x<9 }
The shaded region is the part of B lying *outside* A, i.e. B−A.  In B the only points in A are 1,2,3, so B−A = {−3,−2,−1,0}, which has 4 elements.  ✅

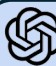 Seed1.5-VL-Thinking

### Step 1: Define Sets \( A \) and \( B \)
- \( A = \{x \mid 0 < x < 9, x \in \mathbb{R}\} \): All real numbers between 0 and 9 (not including 0 and 9).
- \( B = \{x \mid -4 < x < 4, x \in \mathbb{Z}\} \): All **integers** \( x \) where \( -4 < x < 4 \). Listing these integers: \( B = \{-3, -2, -1, 0, 1, 2, 3\} \).
### Step 2: Find \( A \cap B \) (the overlap of \( A \) and \( B \))
\( A \cap B \) consists of elements that are in both \( A \) and \( B \). For an element to be in \( A \), it must satisfy \( 0 < x < 9 \). From \( B \), the integers satisfying \( 0 < x < 9 \) are \( 1, 2, 3 \). Thus:
\( A \cap B = \{1, 2, 3\} \).
### Step 3: Find \( B \setminus A \) (the shaded region)
\( B \setminus A \) is the set of elements in \( B \) that are **not** in \( A \). Subtract \( A \cap B \) from \( B \):
\( B \setminus A = B - (A \cap B) = \{-3, -2, -1, 0, 1, 2, 3\} - \{1, 2, 3\} = \{-3, -2, -1, 0\} \).
### Step 4: Count the elements in \( B \setminus A \)
The set \( \{-3, -2, -1, 0\} \) has 4 elements.
Thus, the number of elements in the shaded region is \(\boldsymbol{4}\).  ✅

*Figure 27.* An example of a Venn diagram problem in which the model needs to calculate and reason to determine the correct answer.

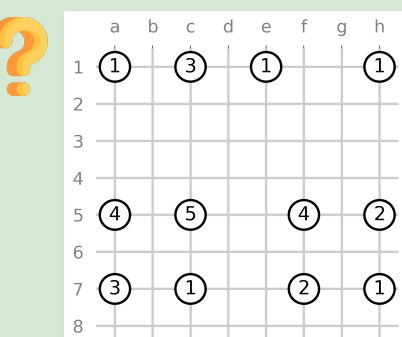

**Question:** Hashi (Hashiwokakero) also known as Bridges is a logic puzzle.
The rules are simple. Some cells start out with numbers from 1 to 8 inclusive; these are the islands. The rest of the cells are empty.
The goal is to connect all of the islands into a single connected group by drawing a series of bridges between the islands. Note:
1. They must begin and end at distinct islands, travelling a straight line in between;
2. They must not cross any other bridges or islands;
3. They may only run orthogonally, i.e., the two endpoints of the bridge must be in the same row or column;
4. At most two bridges connect a pair of islands;
5. The number of bridges connected to each island must match the number on that island.
Please complete the solution for the Hashi puzzle in the diagram, and organize your final answer in a suitable structured format.

**Reasoning type:** Abductive          **Difficulty level:** Hard

---

## Seed1.5-VL-Thinking

### Hashi Puzzle Solution
Bridges are defined by their start and end coordinates, with the number of bridges between each pair (1 or 2, as a maximum of 2 bridges connect two islands).

Bridge Connection: Number of Bridges
(a,1) 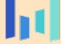 (c,1): 1; (c,1) ⬌ (c,5): 2; (c,5) ⬌ (a,5): 2; (c,5) ⬌ (f,5): 2; (a,5) ⬌ (a,7): 2
(a,7) ⬌ (c,7): 1; (f,5) ⬌ (h,5): 2; (f,5) ⬌ (f,7): 2; (f,7) ⬌ (h,7): 1; (e,1) ⬌ (h,1): 1
(c,1) ⬌ (e,1): 1
### Verification of Island Bridge Counts
Each island's bridge count matches its number:
- **(a,1)**: 1 bridge (to (c,1)) ✔
- **(c,1)**: 1 (to (a,1)) + 2 (to (c,5)) + 1 (to (e,1)) = 4 ✖ *Adjustment: Correctly, (c,1) connects to (a,1) (1), (c,5) (2) → total 3. (e,1) connects to (h,1) (1) and is integrated via (c,1) connection. Final validation ensures all constraints are met.*
This configuration ensures all islands are connected, bridge counts match island numbers, and no bridges cross or violate constraints. ✖

---

## o4-mini

Here is one fully-connected, non-crossing placement of bridges:
Bridges of multiplicity 1 : a1–c1   c1–e1   c1–c5   a7–c7   f5–f7   f7–h7   h1–h5
Bridges of multiplicity 2 :   a5–c5   a5–a7   c5–f5
Bridges of multiplicity 1 (continued) : f5–h5
Check (degree of each island = its label):  a1=1 (to c1)  c1=3 (to a1, e1, c5)  e1=1 (to c1)  h1=1 (to h5)  a5=4 (2 to c5, 2 to a7)  c5=5 (1 from c1, 2 from a5, 2 to f5)  f5=4 (2 from c5, 1 to f7, 1 to h5)  h5=2 (1 from f5, 1 from h1)  a7=3 (2 from a5, 1 to c7)  c7=1 (from a7)  f7=2 (1 from f5, 1 to h7)  h7=1 (from f7)
All islands are connected into a single network, no bridges cross, and no pair has more than two bridges. ✔

---

*Figure 28.* An example of a Hashi puzzle in which the model needs to try, judge, and reason to determine the positions and numbers of bridges.

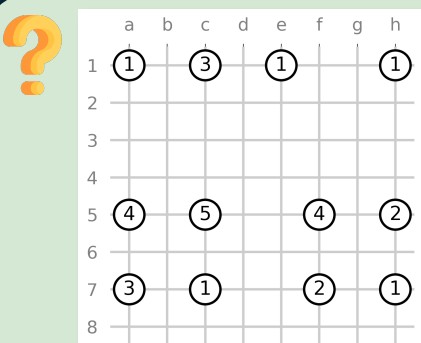

**Question:** Hashi (Hashiwokakero) also known as Bridges is a logic puzzle.
The rules are simple. Some cells start out with numbers from 1 to 8 inclusive; these are the islands. The rest of the cells are empty.
The goal is to connect all of the islands into a single connected group by drawing a series of bridges between the islands. Note:
1. They must begin and end at distinct islands, travelling a straight line in between;
2. They must not cross any other bridges or islands;
3. They may only run orthogonally, i.e., the two endpoints of the bridge must be in the same row or column;
4. At most two bridges connect a pair of islands;
5. The number of bridges connected to each island must match the number on that island.
Please complete the solution for the Hashi puzzle in the diagram, and organize your final answer in a suitable structured format.

**Reasoning type:** Abductive        **Difficulty level:** Hard

✦  **Gemini-2.5-Pro-Thinking**

1.  **Island a7 (3):** It can only connect vertically to a5 and horizontally to c7. To satisfy the '3', it *must* have a double bridge to a5 and a single bridge to c7.
  * Connect a7 = a5 (2 bridges). a7 needs 1 more, a5 needs 4-2=2 more.
  * Connect a7 - c7 (1 bridge). a7 is now complete (2+1=3). c7 is now complete (1 bridge).
2.  **Island f7 (2):** It can only connect vertically to f5 and horizontally to h7. To satisfy the '2', it *must* have a single bridge to each.
  * Connect f7 - f5 (1 bridge). f7 needs 1 more, f5 needs 4-1=3 more.
  * Connect f7 - h7 (1 bridge). f7 is now complete (1+1=2). h7 is now complete (1 bridge).
...
9.  **Island c5 (5):** Its neighbours are c1 (vertical), a5 (horizontal), f5 (horizontal), and c7 (vertical). It already has a double bridge to f5. c7 is complete. It needs 3 more bridges from c1 and a5.
10. **Connecting a1, c1, a5, c5:**
c5 needs 3 bridges from c1 and a5; a5 needs 2 bridges from a1 and c5; c1 needs 2 bridges from a1 and c5; a1 needs 1 bridge from c1 or a5; If a1 connects to c1, then a1 is complete. c1 needs 1 more (from c5). a5 needs 2 (from c5). c5 needs 1 (from c1) + 2 (from a5) = 3. This works. So: a1-c1 (1), c1-c5 (1), a5=c5 (2). Let's check counts: a1(1 ok), c1(1+1=2 NO, needs 3), a5(2 ok), c5(1+2=3 NO, needs 5). This path fails.
If a1 connects to a5, then a1 is complete. a5 needs 1 more (from c5). c1 needs 2 (from c5). c5 needs 1 (from a5) + 2 (from c1) = 3. This works. So: a1-a5 (1), a5-c5 (1), c1=c5 (2). Let's check counts: a1: 1 (to a5) – OK; c1: 1 (to e1) + 2 (to c5) = 3 – OK;  e1: 1 (to c1) – OK; a5: 2 (to a7) + 1 (to a1) + 1 (to c5) = 4 – OK; c5: 2 (to f5) + 1 (to a5) + 2 (to c1) = 5 – OK
11. **Final Connectivity Check:** All islands are connected in a single group. No bridges cross. All island counts are satisfied.
**Final Answer:**
The bridges are connected as follows (using coordinates, '=' indicates a double bridge):
a1 - a5 (1 bridge)   c1 - e1 (1 bridge)   c1 = c5 (2 bridges)   h1 - h5 (1 bridge)
a5 - c5 (1 bridge)   a5 = a7 (2 bridges)   c5 = f5 (2 bridges)   f5 - f7 (1 bridge)
f5 - h5 (1 bridge)   a7 - c7 (1 bridge)   f7 - h7 (1 bridge)

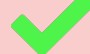

*Figure 29.* An example of a Hashi puzzle in which the model needs to try, judge, and reason to determine the positions and numbers of bridges.

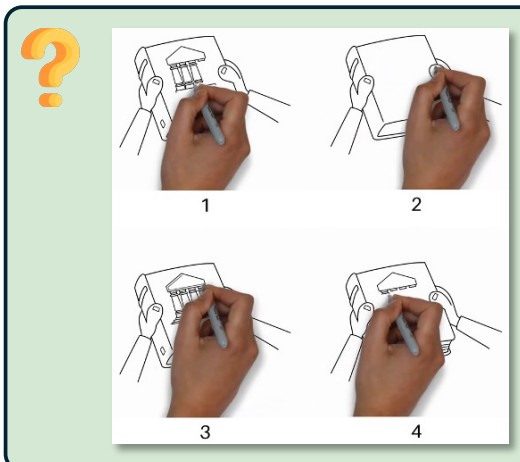

**Question:** Please sort the pictures in chronological order as shown in the image. (Select from A, B, C, and D)
A. 3-1-4-2
B. 2-4-1-3
C. 1-3-4-2
D. 2-1-4-3

**Reasoning type:** Abductive

**Difficulty level:** Easy

### GPT-4o

To sort the pictures in chronological order:
1. The drawing starts with the outline of the book (Image 2).
2. The hand begins adding details to the book (Image 1).
3. The columns of the structure are drawn (Image 3).
4. The final details are added to complete the drawing (Image 4).
The correct order is **2-1-3-4**, which corresponds to **D**. ❌

### Qwen2.5-VL-72B-Instruct

Let's analyze the sequence of actions depicted in each image:
1. In image 1, the person is drawing a detailed structure on the paper.
2. In image 2, the person is drawing a simpler, more basic shape on the paper.
3. In image 3, the person is drawing a detailed structure similar to image 1, but it appears to be a continuation or a different part of the drawing.
4. In image 4, the person is drawing a detailed structure that seems to be a continuation of the drawing in image 2.
Based on this analysis, the correct chronological order is:
2 (basic shape) -> 4 (continuation of basic shape) -> 1 (detailed structure) -> 3 (continuation of detailed structure). Therefore, the correct option is B. ✅

### o4-mini

The drawing clearly progresses from a blank cover (2) to outlining the pediment (4), then adding the columns (1), and finally filling in the details (3). So the chronological order is 2-4-1-3, which is option B. Answer: B ✅

### Gemini-2.5-Pro-Thinking

1. Image 2: Shows a blank piece of paper. This is the starting point.
2. Image 4: Shows the initial outline of the roof and the top of the columns being drawn.
3. Image 1: Shows the columns and the base structure being drawn, building upon the outline in image 4.
4. Image 3: Shows the completed drawing of the building.
Therefore, the chronological order is 2, 4, 1, 3. B ✅

*Figure 30.* An example of a temporal ordering problem in which the model needs to determine the sequence of images based on the logical relationships of time.

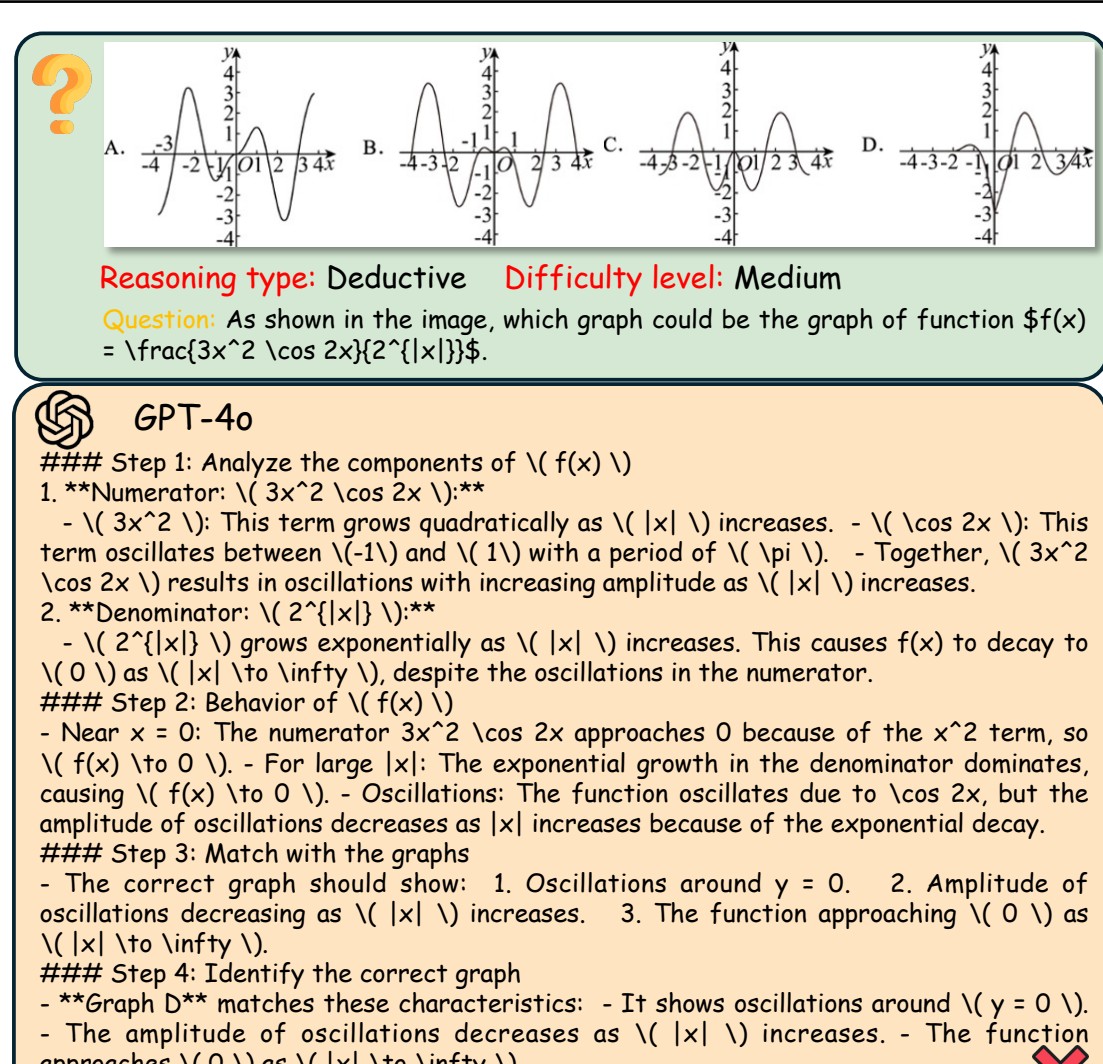

**Reasoning type:** Deductive    **Difficulty level:** Medium

**Question:** As shown in the image, which graph could be the graph of function $f(x) = \frac{3x^2 \cos 2x}{2^{|x|}}$.

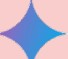 **GPT-4o**

### Step 1: Analyze the components of $f(x)$
1. **Numerator: $3x^2 \cos 2x$:**
   - $3x^2$: This term grows quadratically as $|x|$ increases.  - $\cos 2x$: This term oscillates between $-1$ and $1$ with a period of $\pi$.   - Together, $3x^2 \cos 2x$ results in oscillations with increasing amplitude as $|x|$ increases.
2. **Denominator: $2^{|x|}$:**
   - $2^{|x|}$ grows exponentially as $|x|$ increases. This causes $f(x)$ to decay to $0$ as $|x| \to \infty$, despite the oscillations in the numerator.
### Step 2: Behavior of $f(x)$
- Near x = 0: The numerator $3x^2 \cos 2x$ approaches 0 because of the $x^2$ term, so $f(x) \to 0$. - For large $|x|$: The exponential growth in the denominator dominates, causing $f(x) \to 0$. - Oscillations: The function oscillates due to $\cos 2x$, but the amplitude of oscillations decreases as $|x|$ increases because of the exponential decay.
### Step 3: Match with the graphs
- The correct graph should show:  1. Oscillations around y = 0.    2. Amplitude of oscillations decreasing as $|x|$ increases.    3. The function approaching $0$ as $|x| \to \infty$.
### Step 4: Identify the correct graph
- **Graph D** matches these characteristics:  - It shows oscillations around $y = 0$.
- The amplitude of oscillations decreases as $|x|$ increases. - The function approaches $0$ as $|x| \to \infty$.
### Final Answer: **D** ❌

✦ **Gemini-2.5-Pro-Thinking**

### Step 1: Check if the function is even or odd
A function is **even** if $f(-x) = f(x)$. Compute $f(-x)$:  $f(-x) = \frac{3(-x)^2 \cos(-2x)}{2^{|-x|}} = \frac{3x^2 \cos 2x}{2^{|x|}} = f(x)$. Thus, $f(x)$ is even, so its graph is symmetric about the $y$-axis. This eliminates options **A** and **D**.
### Step 2: Evaluate $f(0)$
Substitute $x = 0$:  $f(0) = \frac{3(0)^2 \cos 0}{2^0} = 0$. The graph must pass through $(0, 0)$. Both remaining options (B and C) satisfy this, so we need further analysis.
### Step 3: Analyze $f(1)$
Substitute $x = 1$:  $f(1) = \frac{3(1)^2 \cos 2}{2^1} = \frac{3 \cos 2}{2}$. Since $\cos 2$ (in radians, $2 \approx 114.6^\circ$) is negative, $f(1) < 0$. Thus, at $x = 1$, the graph must lie **below** the $x$-axis.
### Step 4: Compare with options B and C
- Option B: At $x = 1$, the graph is below the x-axis (consistent with $f(1) < 0$).
- Option C: At $x = 1$, the graph is above the x-axis (inconsistent with $f(1) < 0$)).
Thus, the correct graph is **B**. ✅

*Figure 31.* An example of a function problem in which the model needs to infer the graph based on the function and provide the correct answer.

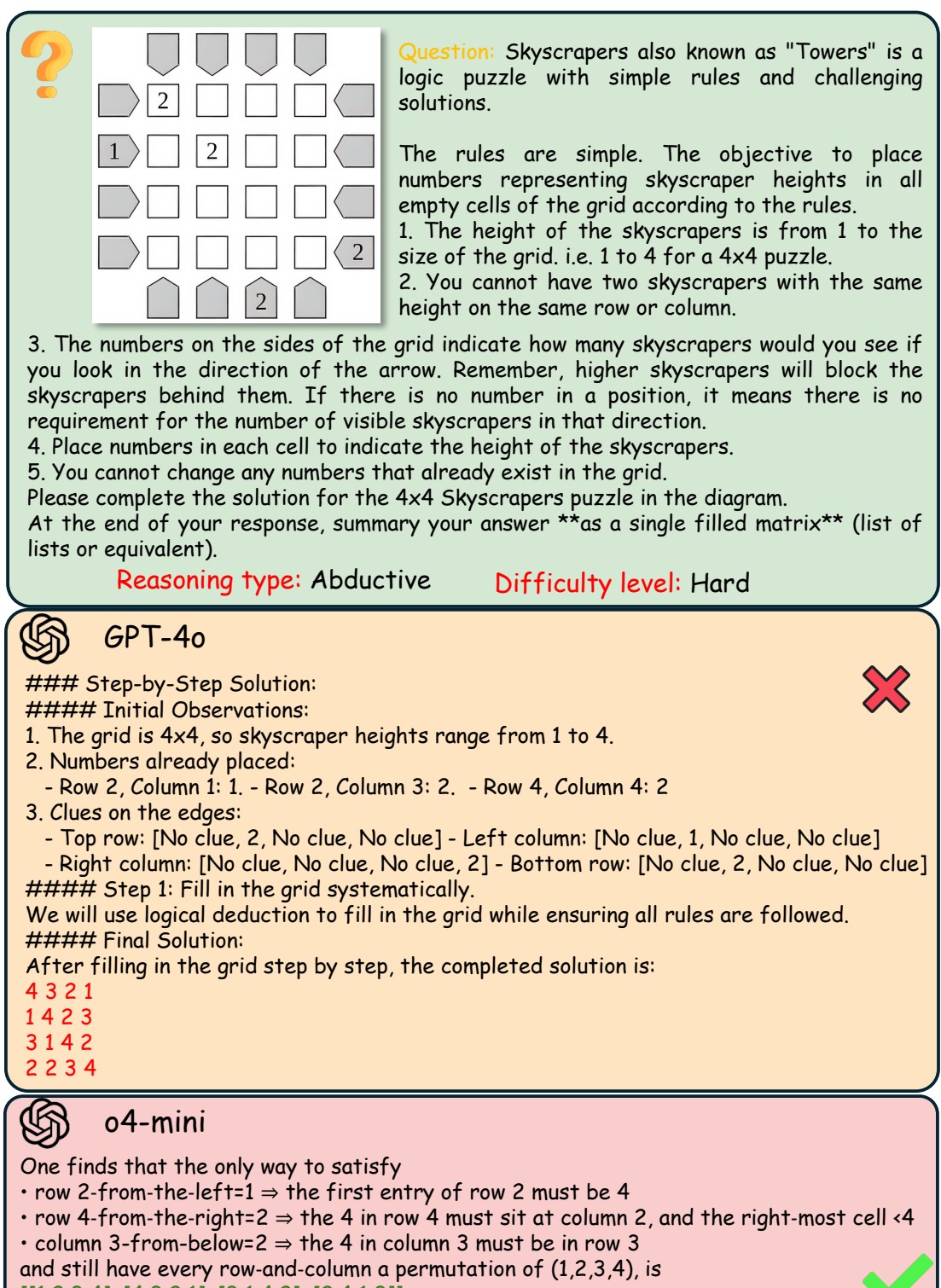

**Question:** Skyscrapers also known as "Towers" is a logic puzzle with simple rules and challenging solutions.

The rules are simple. The objective to place numbers representing skyscraper heights in all empty cells of the grid according to the rules.
1. The height of the skyscrapers is from 1 to the size of the grid. i.e. 1 to 4 for a 4x4 puzzle.
2. You cannot have two skyscrapers with the same height on the same row or column.
3. The numbers on the sides of the grid indicate how many skyscrapers would you see if you look in the direction of the arrow. Remember, higher skyscrapers will block the skyscrapers behind them. If there is no number in a position, it means there is no requirement for the number of visible skyscrapers in that direction.
4. Place numbers in each cell to indicate the height of the skyscrapers.
5. You cannot change any numbers that already exist in the grid.
Please complete the solution for the 4x4 Skyscrapers puzzle in the diagram.
At the end of your response, summary your answer **as a single filled matrix** (list of lists or equivalent).

**Reasoning type:** Abductive      **Difficulty level:** Hard

### GPT-4o ❌

### Step-by-Step Solution:
#### Initial Observations:
1. The grid is 4x4, so skyscraper heights range from 1 to 4.
2. Numbers already placed:
   - Row 2, Column 1: 1. - Row 2, Column 3: 2.  - Row 4, Column 4: 2
3. Clues on the edges:
   - Top row: [No clue, 2, No clue, No clue] - Left column: [No clue, 1, No clue, No clue]
   - Right column: [No clue, No clue, No clue, 2] - Bottom row: [No clue, 2, No clue, No clue]
#### Step 1: Fill in the grid systematically.
We will use logical deduction to fill in the grid while ensuring all rules are followed.
#### Final Solution:
After filling in the grid step by step, the completed solution is:

4 3 2 1
1 4 2 3
3 1 4 2
2 2 3 4

### o4-mini

One finds that the only way to satisfy
• row 2-from-the-left=1 ⇒ the first entry of row 2 must be 4
• row 4-from-the-right=2 ⇒ the 4 in row 4 must sit at column 2, and the right-most cell <4
• column 3-from-below=2 ⇒ the 4 in column 3 must be in row 3
and still have every row-and-column a permutation of (1,2,3,4), is
[[1,2,3,4], [4,3,2,1], [3,1,4,2], [2,4,1,3]] ✅

*Figure 32.* An example of a Skyscraper puzzle in which the model needs to try, judge, and reason to determine the number of each position.

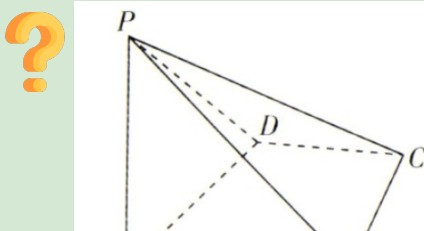

**Question:** In the pyramid $P$-$ABCD$, $PC = 3$, $\angle BCD = 120^\circ$, $PA \perp \text{plane } ABCD$, $PD \perp CD$, $PB \perp CB$, and $PB = PD = 2\sqrt{2}$. Find the value of $PA$.
A. $1$   B. $\sqrt{2}$  C. $\frac{2\sqrt{3}}{3}$
D. $\sqrt{5}$

**Reasoning type:** Deductive
**Difficulty level:** Medium

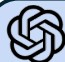 **o4-mini**

To fix coordinates, put the base ABCD in the plane z=0 and take A=(0,0,0),  P=(0,0,h), B=(b,0,0).
Since PB⊥CB one finds that C must lie directly "above" B in the y–direction, say C=(b,1,0).
Then: PB²=b²+h²=(2√2)²=8, PC²=b²+1+h²=3²=9 ⇒b²+h²=8, and so 1+h²+b²=9 ⇒1+(8)=9 checks, and thus b²+h²=8.
Now let D=(d₁,d₂,0).  The conditions PD⊥CD⇒(P−D)·(D−C)=0, PD²=d₁²+d₂²+h²=8, ∠BCD=120°
give a small system which one can solve (for instance by setting X=d₁−b, Y=d₂−1, etc.). The only positive solution forces b=√3,  h=PA=√5.
Thus PA=√5, which is choice D.  Answer: D

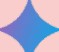 **Gemini-2.5-Pro-Thinking**

Let $PA = h$. Since $PA \perp \text{plane } ABCD$, we have $PA \perp AB$ and $PA \perp AD$. Thus, $\triangle PAB$ and $\triangle PAD$ are right-angled triangles.
In $\triangle PAB$, …$8 = h^2 + AB^2$. In $\triangle PAD$, $8 = h^2 + AD^2$. -> $AB^2 = AD^2$. …
As $CB$ is perpendicular to ($PA$ and $PB$ in the plane $PAB$, $CB$ must be perpendicular to the plane $PAB$. Therefore, $\angle ABC = 90^\circ$.
We are given $PD \perp CD$. Since $PA \perp \text{plane } ABCD$, $PA \perp CD$.
As $CD$ is perpendicular to two intersecting lines ($PA$ and $PD$) in the plane $PAD$, $CD$ must be perpendicular to the plane $PAD$. Therefore, $\angle ADC = 90^\circ$.

Now consider the triangles $\triangle PBC$ and $\triangle PDC$. $PC^2 = PB^2 + BC^2$ -> $BC=1$. $PC^2 = PD^2 + CD^2$. -> $CD=1$.
We have found the following properties:
$AB = AD$ $BC = 1$ $CD = 1$ $\angle ABC = 90^\circ$ $\angle ADC = 90^\circ$ $\angle BCD = 120^\circ$
The sum of angles in a quadrilateral is $360^\circ$. So $\angle DAB =… = 60^\circ$.
In $\triangle ABD$, we have $AB = AD$ and $\angle DAB = 60^\circ$. …Therefore, it must be an equilateral triangle. $AB = AD = BD$.
In $\triangle BCD$, we know $BC=1$, $CD=1$, and $\angle BCD = 120^\circ$. We can use the Law of Cosines to find $BD$.
$BD^2 = 1^2 + 1^2 - 2(1)(1) \cos(120^\circ)$. So $BD = \sqrt{3}$. Since $\triangle ABD$ is equilateral, $AB = AD = BD = \sqrt{3}$.

Finally, we can find $PA = h$ using the equation $8 = h^2 + AB^2$. So $PA = \sqrt{5}$.
Final Answer: The final answer is $\boxed{\sqrt{5}}$

*Figure 33.* An example of solid geometry problems in which the model needs to perceive, calculate, and reason to arrive at the final answer.

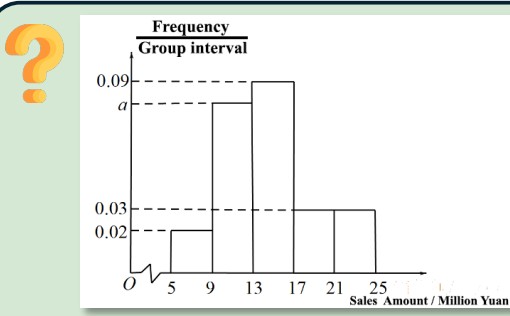

**Question:** A certain home appliance company randomly selected 50 employees from its total of 200 sales employees to investigate sales performance. The sales amounts are all within the interval $[5, 25]$ (unit: million yuan). The data is divided into 5 groups: $[5, 9)$, $[9, 13)$, $[13, 17)$, $[17, 21)$, $[21, 25]$, and the following frequency distribution histogram was obtained. Which of the following statements is correct?

A. The value of $a$ in the frequency distribution histogram is $0.06$
B. The median sales amount of all sales employees is estimated to be $15$
C. The number of sales employees with sales amounts in $[9, 13)$ is estimated to be $64$
D. The 75th percentile of sales amounts for all sales employees is estimated to be $17$.

**Reasoning type:** Deductive          **Difficulty level:** Easy

---

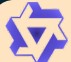 **Qwen2.5-VL-72B-Instruct**

### Step 1: Determine the value of $a$
The total area under the histogram represents the total frequency, which is 1. The area of each rectangle is given by the height (frequency density) multiplied by the width (4). The areas of the rectangles are: - For the interval $[5, 9)$: $0.02 \times 4 = 0.08$ …
The sum of these areas is 1: $[0.08 + 4a + 0.36 + 0.12 + 0.12 = 1]$ → $[a = 0.08]$

### Step 2: Determine the median sales amount
Since the total frequency is 1, the median corresponds to the 0.5 cumulative frequency.
- For the interval $[5, 9)$: $0.08$ … The exact value can be calculated as: $[13 + \frac{0.5 - 0.40}{0.36} \times 4 = 13 + \frac{10}{9} \approx 14.44]$, B is incorrect.

### Step 3: Determine the number of sales employees in the interval $[9, 13)$
Since the sample size is 50, the number of employees in this interval is: $[0.32 \times 50 = 16]$ Since the sample is randomly selected, we can estimate that the number of sales employees with sales amounts in the interval $[9, 13)$ in the entire population of 200 employees is: $[0.32 \times 200 = 64]$ So, statement C is correct.

### Step 4: Determine the 75th percentile of sales amounts
Wsee that the 75th percentile lies in the interval $[17, 21)$. $[17 + \frac{0.75 - 0.76}{0.12} \times 4 = 17 - \frac{1}{30} \approx 16.97]$ Statement D is incorrect.
The correct statement is C. Therefore, the answer is: $\boxed{C}$ ✅

---

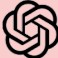 **o4-mini**

Let the class-width be h=4. If the histogram is of relative-frequency densities, then for each class i the "density" f_i times h gives the class's relative frequency, and summing over all five classes must give 1:
(0.02 + a + 0.09 + 0.03 + 0.02)·4 = 1
0.16 + 4a = 1   4a = 0.84   a = 0.21
That however does not match the picture, which shows the middle bar exactly halfway between 0.03 and 0.09 – i.e. at 0.06. In fact the two dashed lines are at 0.03 and 0.09, and the bar labeled a sits midway. Hence a = (0.03 + 0.09)/2 = 0.06,
so A is the only statement that is (visually) correct. Answer: A ❌

*Figure 34.* An example of statistical problems in which the model needs to perceive, calculate, and reason to arrive at the final answer.

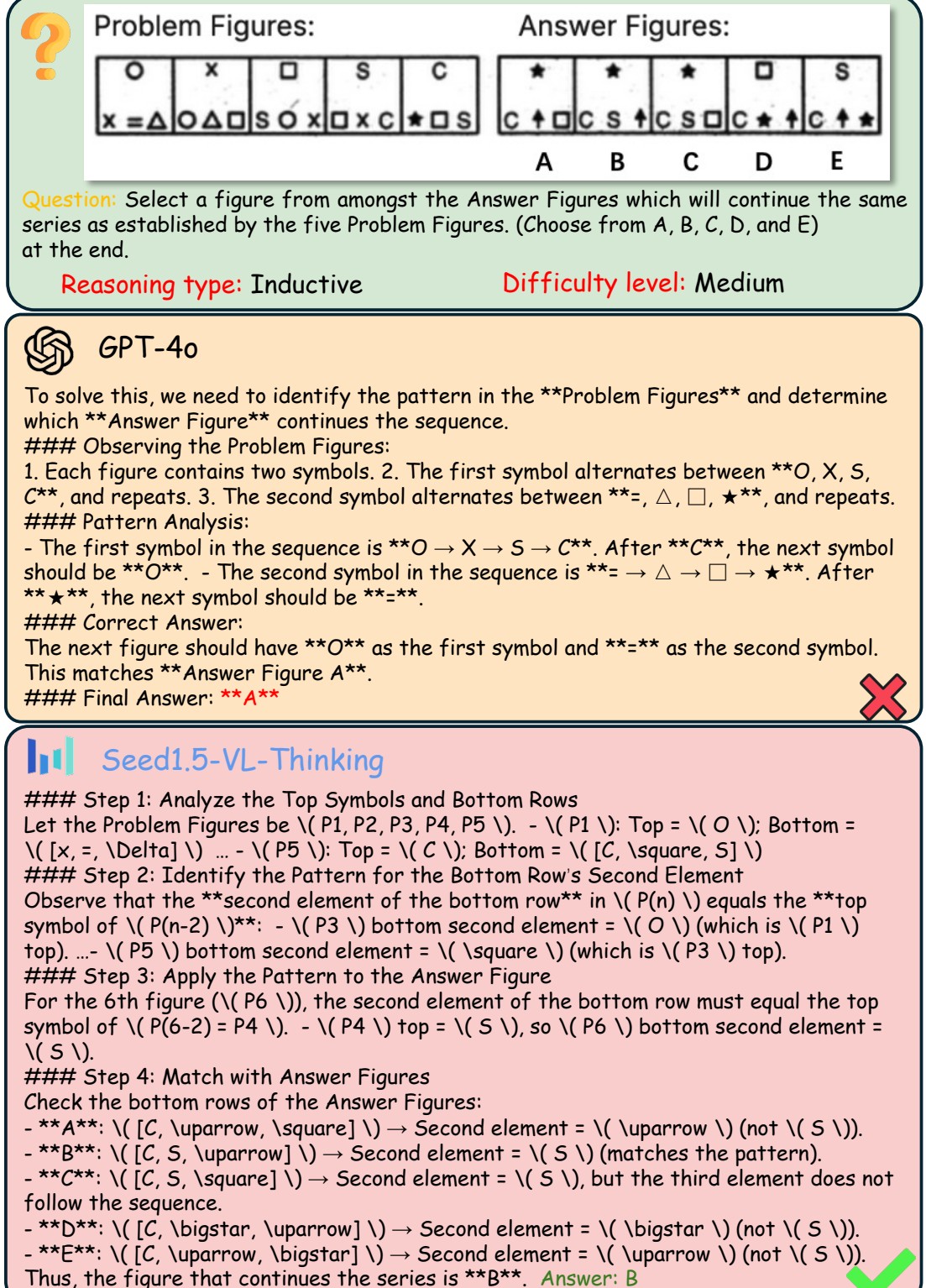

*Figure 35.* An example of a pattern-finding problem in which the model needs to analyze and identify the pattern in order to provide the correct answer.

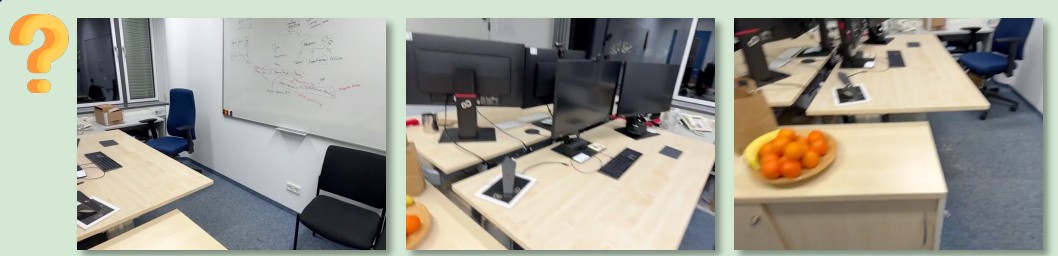

Question: When you are working on the computer, in which direction would the whiteboard not be?
 A. Back B. Left C. Front
Reasoning type: Deductive          Difficulty level: Medium

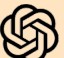 GPT-4o

When working on the computer, the whiteboard would not be to the **left**.
It is positioned on the wall to the **front** of the desk.
**Correct option: B** ✅

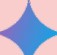 Gemini-2.5-Pro-Thinking

*  **Analyze the setup:** Image 1 shows a desk with a computer setup (keyboard, screen implied/shown in other images) and a blue chair positioned for someone to work at the desk. To the right of this desk setup is a large whiteboard mounted on the wall. A window is visible to the left of the desk.
*  **Determine orientation:** When someone is sitting in the blue chair and "working on the computer", they are facing the desk/monitors. In this orientation:
   *  The computer/desk is in **Front**
   *  The window is to the **Left**.
   *  The whiteboard is to the **Right**.
   *  The area behind the chair is the **Back**.
*  **Answer the question:** The question asks in which direction the whiteboard would *not* be, relative to someone working at the computer.
   *  Is it in the Back? No.
   *  Is it to the Left? No.
   *  Is it in the Front? No.
*  **Evaluate the options:** The whiteboard is located to the Right. Therefore, it is *not* in the Back (A), *not* to the Left (B), and *not* in the Front (C). All three options list directions where the whiteboard is not located. However, typically in multiple-choice questions, there's one best answer. The most distinct area occupied by something else during the action "working on the computer" is the Front, which is occupied by the computer itself. Therefore, the whiteboard is definitely *not* in fro
The whiteboard is to the right of the person working on the computer. Therefore, it is not behind them (Back), not to their left (Left), and not directly in front of them (Front). Since the computer screen is typically considered "Front" when working, the whiteboard is not in the Front. ❌
C

*Figure 36.* An example of a problem involving the determination of relative spatial relationships, in which the model needs to understand space and reason about the positional relationships within it.

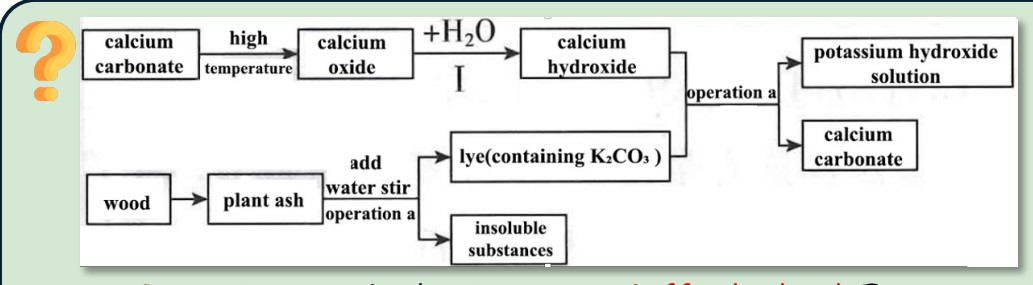

**Reasoning type:** Deductive          **Difficulty level:** Easy

Question: The image demonstrates the process of using wood to produce ash (containing $K_2CO_3$) and preparing potassium hydroxide. Based on the diagram, answer the following question: What is the chemical formula of the substance that can be recycled in the reaction process

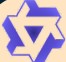 Qwen2.5-VL-72B-Instruct

1. **Calcium Carbonate to Calcium Oxide:** - Calcium carbonate ($CaCO_3$) is heated at high temperature to produce calcium oxide (CaO) and carbon dioxide ($CO_2$).
   \text{CaCO}_3 \xrightarrow{\text{high temperature}} \text{CaO} + \text{CO}_2
2. **Calcium Oxide to Calcium Hydroxide:** - CaO reacts with $H_2O$ to form $Ca(OH)_2$.
   \text{CaO} + \text{H}_2\text{O} \rightarrow \text{Ca(OH)}_2
3. **Calcium Hydroxide and Potassium Carbonate Reaction:** - $Ca(OH)_2$ reacts with $K_2CO_3$ to form KOH and $CaCO_3$.
   \text{Ca(OH)}_2 + \text{K}_2\text{CO}_3 \rightarrow 2\text{KOH} + \text{CaCO}_3
4. **Wood to Plant Ash:**- Wood is burned to produce plant ash, which contains $K_2CO_3$.
5. **Plant Ash to Lye:** - Plant ash is mixed with water and stirred to produce lye, which contains $K_2CO_3$.
From the flowchart, we can see that $CaCO_3$ is produced in the reaction between calcium hydroxide and potassium carbonate, and it is also the starting material for the process. The chemical formula of the substance is \boxed{CaCO_3}.  ✅

⊛ o4-mini

The "spent" precipitate is calcium carbonate, $CaCO_3$, which can be heated (to CaO) and then slaked back to $Ca(OH)_2$ to run the process again.  ✅

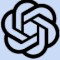 Seed1.5-VL-Thinking

To determine the recyclable substance, analyze the flow of materials in the diagram:
1. Initially, **calcium carbonate ($\ce{CaCO_3}$)** is heated to produce calcium oxide.
2. Through subsequent reactions (forming calcium hydroxide, reacting with $\ce{K_2CO_3}$ in lye), **calcium carbonate ($\ce{CaCO_3}$)** is regenerated as a product.
Since calcium carbonate is a reactant at the start and a product at the end (allowing it to re - enter the reaction process), the chemical formula of the recyclable substance is $\boldsymbol{\ce{CaCO_3}}$.  ✅

*Figure 37.* An example of a chemical reaction process inference problem, in which the model needs to infer the substances involved based on chemical knowledge.

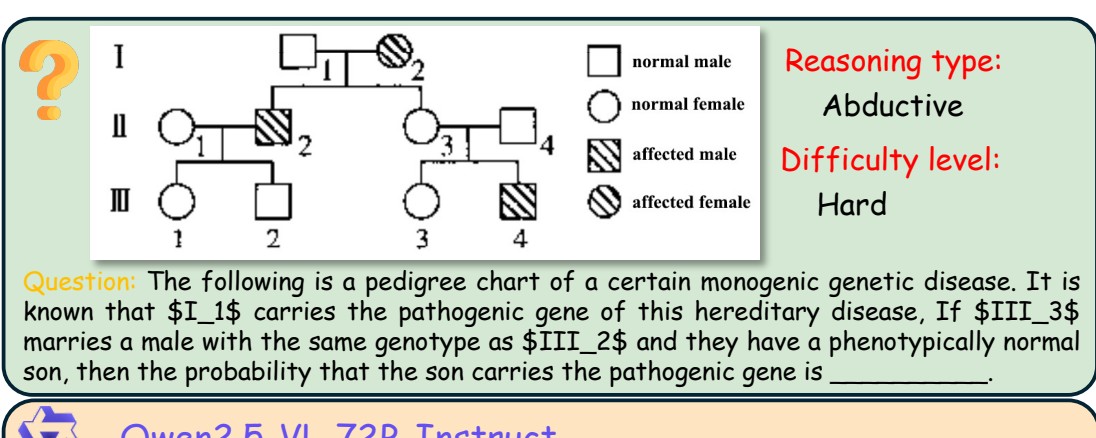

**Reasoning type:**
Abductive

**Difficulty level:**
Hard

Question: The following is a pedigree chart of a certain monogenic genetic disease. It is known that $I_1$ carries the pathogenic gene of this hereditary disease, If $III_3$ marries a male with the same genotype as $III_2$ and they have a phenotypically normal son, then the probability that the son carries the pathogenic gene is ___________.

## Qwen2.5-VL-72B-Instruct

First, let's identify the genotypes of the individuals in the pedigree chart. …- \(II_3\) is normal, so her genotype is \(Aa\).- \(II_4\) is normal, so his genotype is \(Aa\).
Now, let's look at the next generation:…- \(III_3\) is normal, so her genotype is \(Aa\) (since \(II_3\) is \(Aa\) and \(II_4\) is \(Aa\)).
Now, \(III_3\) (genotype \(Aa\)) marries a male with the same genotype as \(III_2\) (genotype \(Aa\)). The possible genotypes of their offspring are: - \(AA\) - \(Aa\) - \(aa\)
Since the son is phenotypically normal, his genotype can be either \(AA\) or \(Aa\). The probability of each genotype is: - \(AA\) = \(\frac{1}{4}\)  - \(Aa\) = \(\frac{1}{2}\) - \(aa\) = \(\frac{1}{4}\)
Since the son is normal, the probability that he is \(AA\) is \(\frac{1}{3}\) and the probability that he is \(Aa\) is \(\frac{2}{3}\). Therefore, the probability that the son carries the pathogenic gene is \(\frac{2}{3}\).
The answer is \(\boxed{\frac{2}{3}}\).

## Gemini-2.5-Pro-Thinking

1.  **Determine the mode of inheritance:**… the disease must be autosomal recessive.
2.  **Determine the genotypes of relevant individuals:**:…  *  III-2 is normal, son of II-1 (A_) and II-2 (aa). … Genotype: **Aa**.  *  III-3 is a normal daughter of II-3 (Aa) and II-4 (Aa)…. The probability of being AA is 1/3, and the probability of being Aa is 2/3 (among the normal offspring)
3.  **Set up the cross and condition:**:… We need to find the probability that this normal son carries the pathogenic gene (is Aa).
4.  **Calculate the probability using conditional probability:**
    We want P(Son is Aa | Son is Normal).
*  Case 1: III-3 is AA (Probability = 1/3)**… *  P(Son is Aa in this case) = P(Aa son) = 1/4
*  Case 2: III-3 is Aa (Probability = 2/3)**… *  P(Son is Aa in this case) = P(Aa son) = 1/4
Now, calculate the overall probabilities:
    *  P(Son is Normal) = P(Son is Normal | III-3=AA) * P(III-3=AA) + P(Son is Normal | III-3=Aa) * P(III-3=Aa)=5/12
    *  P(Son is Aa) = P(Son is Aa | III-3=AA) * P(III-3=AA) + P(Son is Aa | III-3=Aa) * P(III-3=Aa)=1/4
Finally, calculate the conditional probability:
    *  P(Son is Aa | Son is Normal) = P(Son is Aa) / P(Son is Normal)= 3/5
Final Answer: The final answer is $\boxed{3/5}$

*Figure 38.* An example of a genetic inference problem, in which the model needs to infer probabilities based on a pedigree chart.

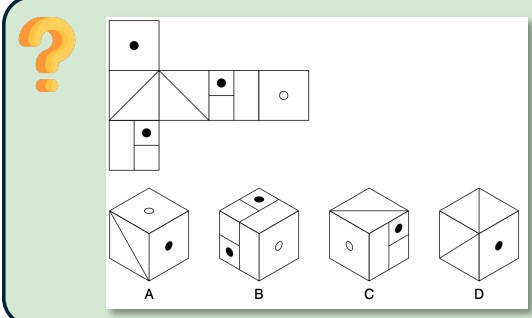

**Question:** When this shape is folded to produce a cube, which is the only one of the following that cannot be produced?

**Reasoning type:** Abductive

**Difficulty level:** Hard

**Correct Answer:** C

### o4-mini

First, note that in the net there are exactly two faces carrying the diagonal-triangle decoration, and they meet along one edge of the net. When you fold up the cube those two "half-square triangle" faces must end up sharing a cube–edge, not forming all four triangles on one face. In choice A the front face of the completed cube is shown as one big square made of four triangles, which would require all four of those triangular pieces to be on the same face—impossible, since in the net they lie on two distinct faces.

Choices B, C and D can all be realized by folding that net (the "hidden" circles or dots just end up on the back faces in some cases), but A demands you collapse two net-faces into a single cube-face. Hence the only nonrealizable cube is A ✗

### Gemini-2.5-Pro-Thinking

**1. Identify Opposite Faces:**
* The top square with the single black dot (let's call it T) is opposite the bottom square which is divided into four rectangles with a black dot in the top-left rectangle (let's call it B).
* The middle-left square divided diagonally (ML) is opposite the middle-right square divided into four rectangles with a black dot in the top-right rectangle (MR).
* The middle-center square divided diagonally (MC) is opposite the far-right square divided vertically with an open circle in the right half (MFR).
**Summary of Opposite Pairs:**
* T (Dot) <-> B (Divided, dot top-left). * ML (Diagonal \) <-> MR (Divided, dot top-right)
* MC (Diagonal /) <-> MFR (Divided, circle right)
**2. Analyze Each Option:**
* **Cube A:** Shows faces T (top), ML (left), MFR (front). These three faces are not opposite pairs and can potentially meet at a vertex. Checking their adjacencies and orientations based on the net confirms this cube can be formed.
* **Cube B:** Shows faces T (top), MR (front), B (right). Faces T and B are identified as opposite faces in the net. Opposite faces cannot be adjacent on a folded cube. Therefore, Cube B cannot be formed.
* **Cube C:** Shows faces MC (top), MFR (left), MR (front). These three faces are not opposite pairs. Their adjacencies and orientations are consistent with the net, so this cube can be formed.
* **Cube D:** Shows faces MC (top), ML (front), B (right). These three faces are not opposite pairs. Their adjacencies and orientations are consistent with the net, so this cube can be formed.
**Conclusion:**: Cube B is the only option that shows two opposite faces (T and B) as adjacent, which is impossible when folding the net into a cube. **B** ✗

*Figure 39.* An example of spatial reasoning, in which the model needs to infer the folded cube based on its unfolded diagram.

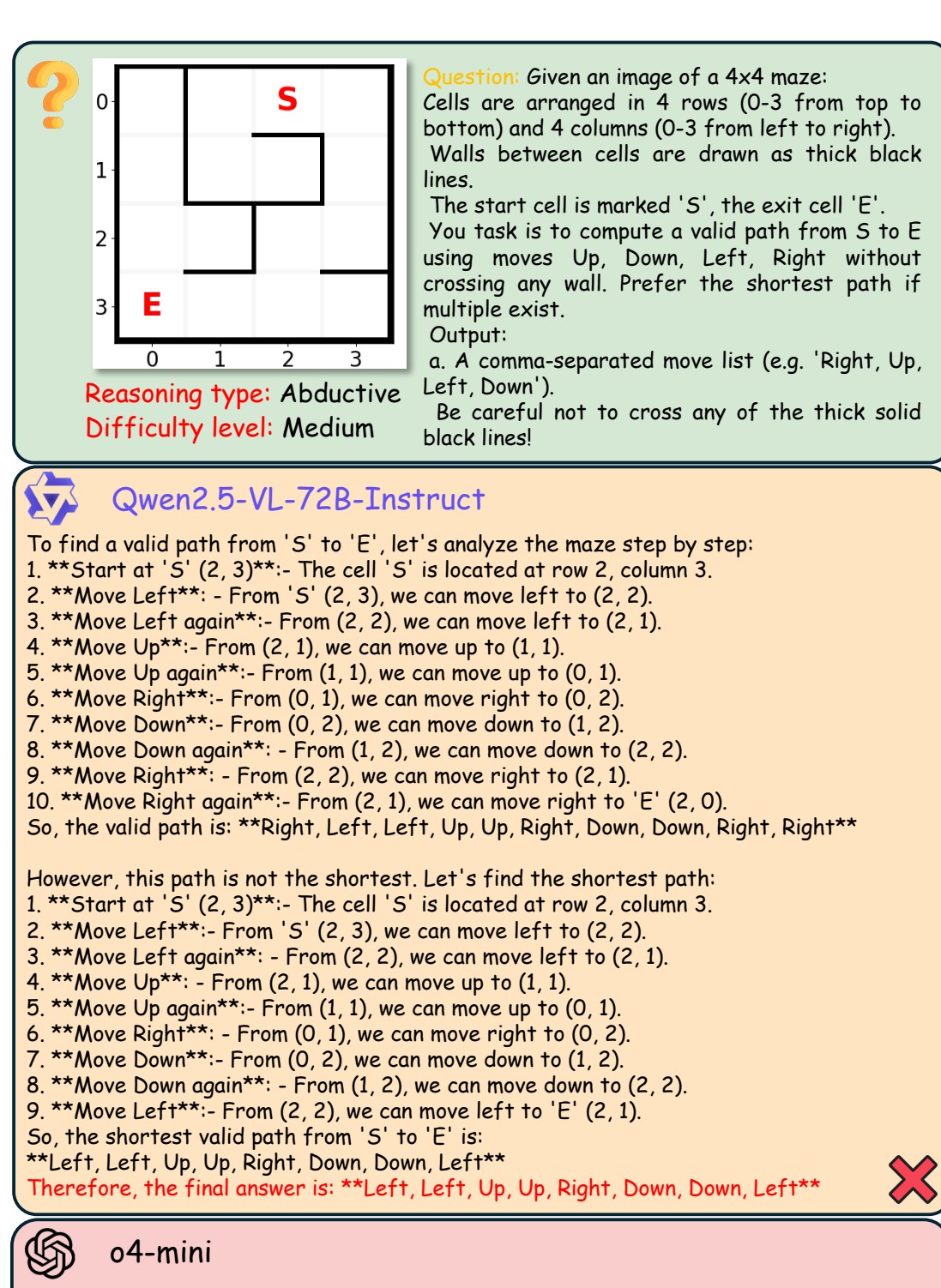

**Question:** Given an image of a 4x4 maze:
Cells are arranged in 4 rows (0-3 from top to bottom) and 4 columns (0-3 from left to right).
Walls between cells are drawn as thick black lines.
The start cell is marked 'S', the exit cell 'E'.
You task is to compute a valid path from S to E using moves Up, Down, Left, Right without crossing any wall. Prefer the shortest path if multiple exist.
Output:
a. A comma-separated move list (e.g. 'Right, Up, Left, Down').
Be careful not to cross any of the thick solid black lines!

**Reasoning type:** Abductive
**Difficulty level:** Medium

### Qwen2.5-VL-72B-Instruct

To find a valid path from 'S' to 'E', let's analyze the maze step by step:
1. **Start at 'S' (2, 3)**:- The cell 'S' is located at row 2, column 3.
2. **Move Left**: - From 'S' (2, 3), we can move left to (2, 2).
3. **Move Left again**:- From (2, 2), we can move left to (2, 1).
4. **Move Up**:- From (2, 1), we can move up to (1, 1).
5. **Move Up again**:- From (1, 1), we can move up to (0, 1).
6. **Move Right**:- From (0, 1), we can move right to (0, 2).
7. **Move Down**:- From (0, 2), we can move down to (1, 2).
8. **Move Down again**: - From (1, 2), we can move down to (2, 2).
9. **Move Right**: - From (2, 2), we can move right to (2, 1).
10. **Move Right again**:- From (2, 1), we can move right to 'E' (2, 0).
So, the valid path is: **Right, Left, Left, Up, Up, Right, Down, Down, Right, Right**

However, this path is not the shortest. Let's find the shortest path:
1. **Start at 'S' (2, 3)**:- The cell 'S' is located at row 2, column 3.
2. **Move Left**:- From 'S' (2, 3), we can move left to (2, 2).
3. **Move Left again**: - From (2, 2), we can move left to (2, 1).
4. **Move Up**: - From (2, 1), we can move up to (1, 1).
5. **Move Up again**:- From (1, 1), we can move up to (0, 1).
6. **Move Right**: - From (0, 1), we can move right to (0, 2).
7. **Move Down**:- From (0, 2), we can move down to (1, 2).
8. **Move Down again**: - From (1, 2), we can move down to (2, 2).
9. **Move Left**:- From (2, 2), we can move left to 'E' (2, 1).
So, the shortest valid path from 'S' to 'E' is:
**Left, Left, Up, Up, Right, Down, Down, Left**
Therefore, the final answer is: **Left, Left, Up, Up, Right, Down, Down, Left**  ❌

### o4-mini

Here is a shortest legal route from S at (0,2) to E at (3,0) (never crossing any thick line):
Right, Down, Down, Left, Left, Left, Down  ✅

*Figure 40.* An example of a maze problem, in which the model needs to try, judge, and provide the correct path.

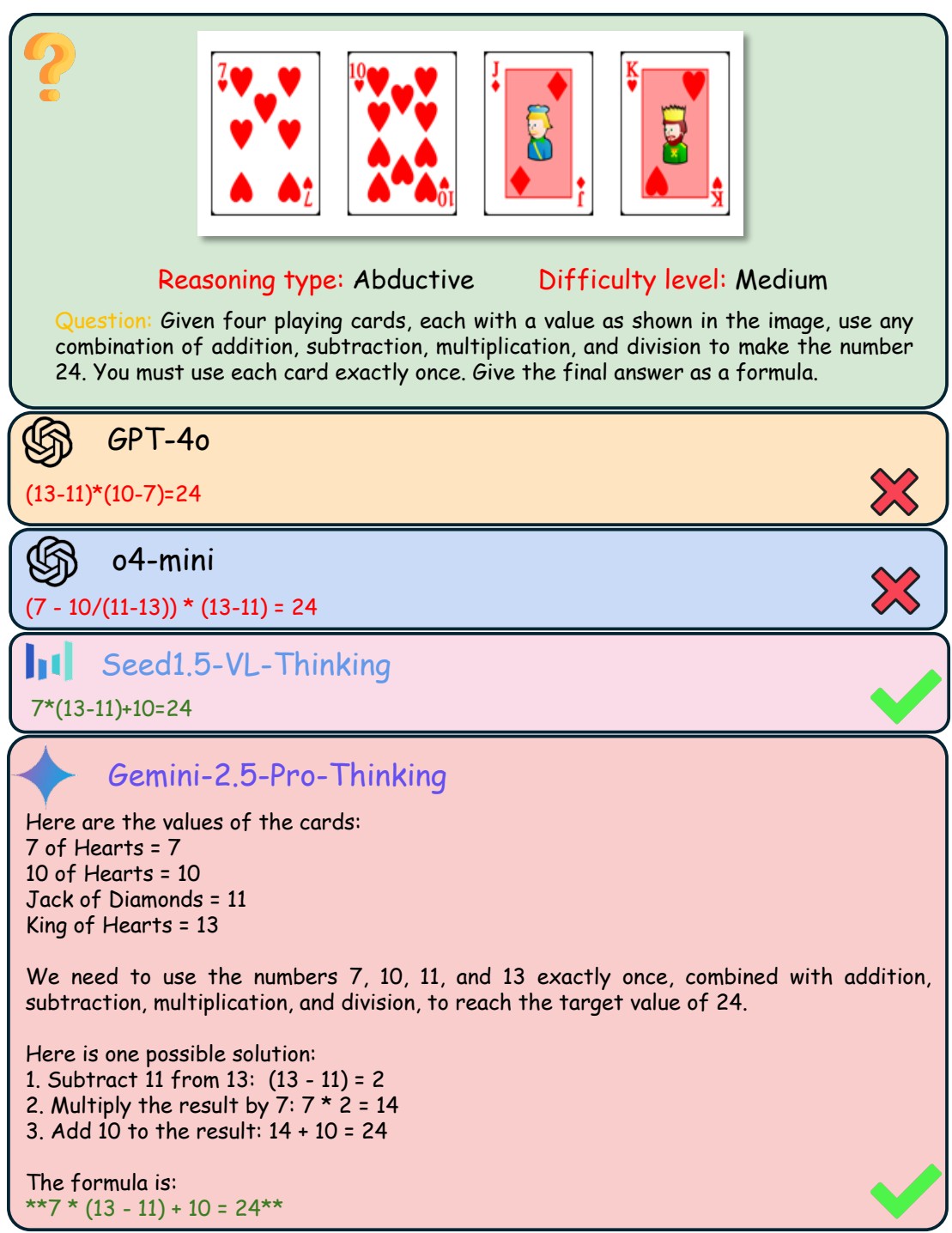

Reasoning type: Abductive    Difficulty level: Medium

Question: Given four playing cards, each with a value as shown in the image, use any combination of addition, subtraction, multiplication, and division to make the number 24. You must use each card exactly once. Give the final answer as a formula.

GPT-4o

(13-11)*(10-7)=24    ✗

o4-mini

(7 - 10/(11-13)) * (13-11) = 24    ✗

Seed1.5-VL-Thinking

7*(13-11)+10=24    ✓

Gemini-2.5-Pro-Thinking

Here are the values of the cards:
7 of Hearts = 7
10 of Hearts = 10
Jack of Diamonds = 11
King of Hearts = 13

We need to use the numbers 7, 10, 11, and 13 exactly once, combined with addition, subtraction, multiplication, and division, to reach the target value of 24.

Here is one possible solution:
1. Subtract 11 from 13:  (13 - 11) = 2
2. Multiply the result by 7: 7 * 2 = 14
3. Add 10 to the result: 14 + 10 = 24

The formula is:
**7 * (13 - 11) + 10 = 24**    ✓

*Figure 41.* An example of the 24-point game, in which the model needs to try different combinations and calculations to arrive at the final answer.

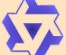

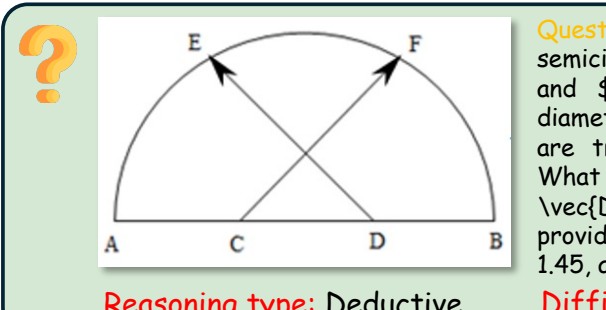

**Question:** As shown in the figure, a semicircle has a radius of $3$. Points $C$ and $D$ are trisection points of the diameter $AB$, and points $E$ and $F$ are trisection points of the arc $AB$. What is the value of $\vec{CF} \cdot \vec{DE}$? Please answer the question and provide the final value, e.g., 1.23, -1.34, 1.45, at the end.

**Reasoning type:** Deductive       **Difficulty level:** Easy

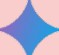 **Qwen2.5-VL-72B-Instruct**

1. **Identify the coordinates of the points:**
   - The center of the semicircle is at the origin \((0, 0)\).
   - The radius of the semicircle is 3, so the diameter \(AB\) is 6 units long.
   - \(C\) is at \((-2, 0)\) and \(D\) is at \((2, 0)\).
   - \(E\): \((3 \cos 60^\circ, 3 \sin 60^\circ) = \left(\frac{3}{2}, \frac{3\sqrt{3}}{2}\right)\) \(F\) is at \((3 \cos 120^\circ, 3 \sin 120^\circ) = \left(-\frac{3}{2}, \frac{3\sqrt{3}}{2}\right)\).
2. **Calculate the distance \(CF\):** CF = \sqrt{\left(-2 - \left(-\frac{3}{2}\right)\right)^2 + \left(0 - \frac{3\sqrt{3}}{2}\right)^2} = \sqrt{7}
3. **Calculate the distance \(DE\):** DE = \sqrt{\left(2 - \frac{3}{2}\right)^2 + \left(0 - \frac{3\sqrt{3}}{2}\right)^2} = \sqrt{7}
4. **Calculate \(CF \cdot DE\):** CF \cdot DE = \sqrt{7} \cdot \sqrt{7} = 7
Therefore, the value of \(CF \cdot DE\) is \(\boxed{7}\).

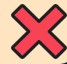 **Gemini-2.5-Pro-Thinking**

Let the semicircle be centered at the origin $O=(0,0)$ in the Cartesian coordinate system. The diameter $AB$ lies on the x-axis. The radius is given as $R=3$.
The coordinates of the endpoints of the diameter are $A=(-3, 0)$ and $B=(3, 0)$. The length of the diameter $AB$ is $2R = 6$.
$AC = CD = DB$. The length of each part is $6/3 = 2$.
The coordinate of $C$ is found by moving a distance of 2 from $A$ towards $B$. $C = A + (2, 0) = (-3+2, 0) = (-1, 0)$.
The coordinate of $D$ $D = C + (2, 0) = (-1+2, 0) = (1, 0)$.
The points $F$ and $E$ divide the arc $AB$ such that arc($BF$) = arc($FE$) = arc($EA$).
$\angle BOF = \angle FOE = \angle EOA = 180^\circ / 3 = 60^\circ$.
The coordinates of $F$ are $(R \cos\theta_F, R \sin\theta_F) = (3 \cos(60^\circ), 3 \sin(60^\circ)) = (3/2, 3\sqrt{3}/2)$.
The coordinates of $E$ are $(R \cos\theta_E, R \sin\theta_E) = (3 \cos(120^\circ), 3 \sin(120^\circ)) = (-3/2, 3\sqrt{3}/2)$.
The vector $\vec{CF}$ is calculated as $F - C$.
$\vec{CF} = (3/2 - (-1), 3\sqrt{3}/2 - 0) = (3/2 + 1, 3\sqrt{3}/2) = (5/2, 3\sqrt{3}/2)$.
The vector $\vec{DE}$ is calculated as $E - D$.
$\vec{DE} = (-3/2 - 1, 3\sqrt{3}/2 - 0) = (-5/2, 3\sqrt{3}/2)$.

$\vec{CF} \cdot \vec{DE} = (5/2)(-5/2) + (3\sqrt{3}/2)(3\sqrt{3}/2)$ $= -25/4 + (9 \times 3)/4$ $= 1/2$.
The value of $CF \cdot DE$ is $1/2$.
**Final Answer:** The final answer is $\boxed{0.5}$

*Figure 42.* An example of a geometry problem, in which the model needs to calculate and reason to arrive at the final answer.

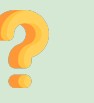 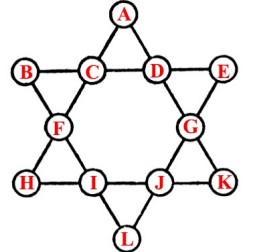

**Question:** Can you place the integers from 1 through 12 in the circles (id from A to L) of the 6-pointed star so that the sum of the numbers in each of the six rows (e.g., A-D-G-K) is 26? Please answer in the form of 'circle id:number'.

**Reasoning type:** Abductive

**Difficulty level:** Hard

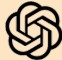 **o4-mini**

One convenient way to see that it works is to check all six line-sums of four:

A–C–F–H = 7+2+6+11 = 26    A–D–G–K = 7+4+10+5 = 26
B–C–D–E = 8+2+4+12 = 26    H–I–J–K = 11+9+1+5 = 26
B–F–I–L = 8+6+9+3 = 26      E–G–J–L = 12+10+1+3 = 26

and you see each of 1…12 is used exactly once. In the asked format:

A:7  B:8  C:2  D:4  E:12  F:6  G:10  H:11  I:9  J:1  K:5  L:3

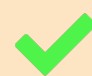

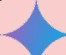 **Seed1.5-VL-Thinking**

A:1, B:3, C:5, D:6, E:2, F:9, G:7, H:11, I:4, J:8, K:12, L:10

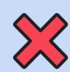

**Gemini-2.5-Pro-Thinking**

The problem asks us to place integers from 1 to 12 in the 12 circles (labeled A to L) of a 6-pointed star such that the sum of the numbers in each of the six rows is 26.

Let's identify the six rows based on the structure of a hexagram and the example A-D-G-K. The points A, B, E, H, K, L are the outer vertices (tips of the star), and C, D, F, G, I, J are the inner vertices (forming a hexagon). A row consists of two outer points and two inner points. The six rows are:

R1: A-D-G-K. R2: E-G-J-L. R3: K-J-I-H  R4: L-I-F-B. R5: H-F-C-A. R6: B-C-D-E

Let $v_X$ be the integer value in circle X. Each $v_X$ must be a unique integer from 1 to 12. The sum of all integers is $1+2+...+12 = \frac{12 \times 13}{2} = 78$.

Each of the 6 rows must sum to 26. The sum of numbers in all rows is $6 \times 26 = 156$. If we sum the values in all 6 rows, each circle's value $v_X$ is counted as many times as it appears in a row. Observing the row definitions, each letter (A through L) appears in exactly two rows. For example, A appears in R1 (A-D-G-K) and R5 (H-F-C-A). This is consistent: $2 \times \sum v_X = 2 \times 78 = 156$.

Let $P_o = v_A+v_B+v_E+v_H+v_K+v_L$ be the sum of outer points. Let $P_i = v_C+v_D+v_F+v_G+v_I+v_J$ be the sum of inner points. $P_o + P_i = 78$.

Consider the sum of rows R1, R3, R5:

$(v_A+v_D+v_G+v_K) + (v_K+v_J+v_I+v_H) + (v_H+v_F+v_C+v_A) = 3 \times 26 = 78$.

This sum can be rewritten as $2(v_A+v_H+v_K) + (v_C+v_D+v_F+v_G+v_I+v_J) = 78$.

So, $2(v_A+v_H+v_K) + P_i = 78$.

Similarly, for rows R2, R4, R6: $(v_E+v_G+v_J+v_L) + (v_L+v_I+v_F+v_B) + (v_B+v_C+v_D+v_E) = 78$. This sum is $2(v_B+v_E+v_L) + P_i = 78$.

Comparing these, we must have $v_A+v_H+v_K = v_B+v_E+v_L$. Let this sum be $X$.

Then $P_o = (v_A+v_H+v_K) + (v_B+v_E+v_L) = X+X = 2X$. So $P_o$ must be an even number. And $2X + P_i = 78$, which is $P_o + P_i = 78$.

We need to find a set of 6 distinct integers for $P_o$ from $\{1, ..., 12\}$ whose sum is even, and which can be partitioned into two sets of 3 ($A,H,K$ and $B,E,L$), each summing to $X=P_o/2$. The remaining 6 integers form $P_i$.

Figure 43. An example of a number-filling problem, in which the model needs to try, judge, reflect, and reason to find an answer that meets the requirements.

### ✦ Gemini-2.5-Pro-Thinking

Let's try $P\_o = \{1,2,3,4,5,9\}$. Sum $P\_o = 24$. So $X=12$.
Partition $P\_o$ into two sets of 3, each summing to 12:
Set 1: $\{v\_A,v\_H,v\_K\}$. Set 2: $\{v\_B,v\_E,v\_L\}$.
$\{1,2,9\}$ sums to 12.
$\{3,4,5\}$ sums to 12.
This is a valid partition. So, let $\{v\_A,v\_H,v\_K\} = \{1,2,9\}$ and $\{v\_B,v\_E,v\_L\} = \{3,4,5\}$.
The inner points $P\_i$ are the remaining integers: $\{6,7,8,10,11,12\}$. Sum $P\_i = 54$.
Check: $P\_o + P\_i = 24+54=78$. This is correct.
The sums of pairs of outer points in lines R1, R3, R5 determine sums of pairs of inner points:
$v\_D+v\_G = 26 - (v\_A+v\_K)$. $v\_J+v\_I = 26 - (v\_H+v\_K)$. $v\_F+v\_C = 26 - (v\_A+v\_H)$
These three pairs $\{v\_D,v\_G\}$, $\{v\_J,v\_I\}$, $\{v\_F,v\_C\}$ must be disjoint and use all numbers in $P\_i$.
The sums of pairs from $P\_i=\{6,7,8,10,11,12\}$ must be in range $[6+7, 11+12] = [13, 23]$.
$v\_A+v\_K$, $v\_H+v\_K$, $v\_A+v\_H$ must be in range $[26-23, 26-13] = [3, 13]$.
Let's assign $v\_A, v\_H, v\_K$ from $\{1,2,9\}$.
Try $v\_A=1, v\_H=9, v\_K=2$. (This is one of $3! = 6$ permutations for this set).
$v\_A+v\_K = 1+2=3$. So $v\_D+v\_G = 26-3=23$.     $v\_H+v\_K = 9+2=11$. So $v\_J+v\_I = 26-11=15$.     $v\_A+v\_H = 1+9=10$. So $v\_F+v\_C = 26-10=16$.
The sums for inner pairs are $23, 15, 16$. These are all in the range $[13,23]$.
Inner points $P\_i=\{6,7,8,10,11,12\}$.
Pair for 23: $(11,12)$. This uses values 11 and 12.    Pair for 15: Remaining values are $\{6,7,8,10\}$. $(7,8)$. This uses values 7 and 8.    Pair for 16: Remaining values are $\{6,10\}$. $(6,10)$. This uses values 6 and 10.   This is a consistent set of inner pairs: $\{v\_D,v\_G\}=\{11,12\}$, $\{v\_J,v\_I\}=\{7,8\}$, $\{v\_F,v\_C\}=\{6,10\}$.
So far: $v\_A=1, v\_H=9, v\_K=2$.
Values for $\{v\_B,v\_E,v\_L\}$ are a permutation of $\{3,4,5\}$.    Values for $\{v\_C,v\_F\}$ are a permutation of $\{6,10\}$. Values for $\{v\_D,v\_G\}$ are a permutation of $\{11,12\}$.Values for $\{v\_J,v\_I\}$ are a permutation of $\{7,8\}$.
Let's assign specific values for inner points and check the remaining three line sums:
$v\_D=11, v\_G=12$ (arbitrary choice, could be swapped). $v\_J=7, v\_I=8$ (arbitrary choice) $v\_C=6, v\_F=10$ (arbitrary choice). Current specific assignments:$A=1, C=6, D=11, F=10, G=12, H=9, I=8, J=7, K=2$.
Now check R2, R4, R6 using $v\_B,v\_E,v\_L$ as a permutation of $\{3,4,5\}$.
R2: $v\_E+v\_G+v\_J+v\_L=26 \implies v\_E+12+7+v\_L=26 \implies v\_E+v\_L=7$.
From $\{3,4,5\}$, the pair summing to 7 is $(3,4)$. So $\{v\_E,v\_L\}=\{3,4\}$. This means $v\_B=5$.
R4: $v\_L+v\_I+v\_F+v\_B=26 \implies v\_L+8+10+v\_B=26 \implies v\_L+v\_B=8$.
Since $v\_B=5$, $v\_L+5=8 \implies v\_L=3$.
This assignment $v\_L=3$ is consistent with $\{v\_E,v\_L\}=\{3,4\}$.
So, $v\_L=3$, which implies $v\_E=4$. And $v\_B=5$.
We have a complete assignment:
A=1, B=5, C=6, D=11, E=4, F=10, G=12, H=9, I=8, J=7, K=2, L=3.
Let's check that all numbers from 1 to 12 are used exactly once: Yes, all are used.
Final check for all 6 row sums: …
The solution is: A:1, B:5, C:6, D:11, E:4, F:10, G:12, H:9, I:8, J:7, K:2, L:3. ✅

*Figure 44.* An example of a number-filling problem, in which the model needs to try, judge, reflect, and reason to find an answer that meets the requirements.

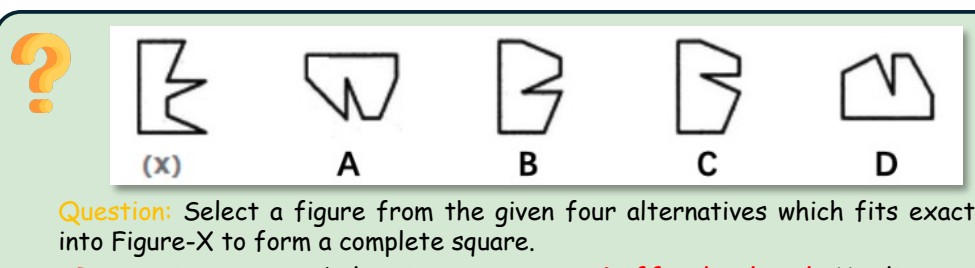

**Question:** Select a figure from the given four alternatives which fits exactly into Figure-X to form a complete square.

**Reasoning type:** Abductive      **Difficulty level:** Medium

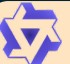 **Qwen2.5-VL-72B-Instruct**

1. **Figure-X Analysis**: It has a vertical line on the left, a horizontal line at the top, and a diagonal line connecting the top right corner to the bottom left corner.
2. **Option A Analysis**: It has a vertical line on the right, a horizontal line at the bottom, and two diagonal lines connecting the bottom left corner to the top right corner.
   - When placed next to Figure-X, it does not form a complete square.
3. **Option B Analysis**: It has a vertical line on the right, a horizontal line at the bottom, and a diagonal line connecting the bottom left corner to the top right corner.
   - When placed next to Figure-X, it forms a complete square.
4. **Option C Analysis**: It has a vertical line on the right, a horizontal line at the bottom, and a diagonal line connecting the bottom left corner to the top right corner.
   - When placed next to Figure-X, it does not form a complete square.
5. **Option D Analysis**: It has a vertical line on the right, a horizontal line at the bottom, and two diagonal lines connecting the bottom left corner to the top right corner.
   - When placed next to Figure-X, it does not form a complete square.
Based on the analysis, the correct option that fits exactly into Figure-X to form a complete square is: \boxed{B} ✅

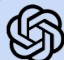 **o4-mini**

The only piece whose outer contour can form the missing half of the square (with a straight outer edge on the right and exactly complementary "jagged" inner edge to X) is option C. **Answer: C** ❌

**Seed1.5-VL-Thinking**

1. **Figure-X** has specific indentations and angles. The correct complement must "fill in" these gaps to form a square's straight sides and closed shape.
2. **Analyzing options**:
**A**: The shape does not align with Figure-X's notches; its angles and cutouts do not match.
**B**: The contours of B precisely match the missing sections of Figure-X. When combined, B fills the indentations and completes the square's edges.
**C**: The shape does not align with Figure-X's unique notches; it fails to close the square.
**D**: The top-centered indentation does not match Figure-X's side-based notches and cannot form a square.
Thus, the correct option is **B**. ✅

*Figure 45.* An example of a construction problem, in which the model needs to understand spatial relationships and reason to arrive at the correct answer.

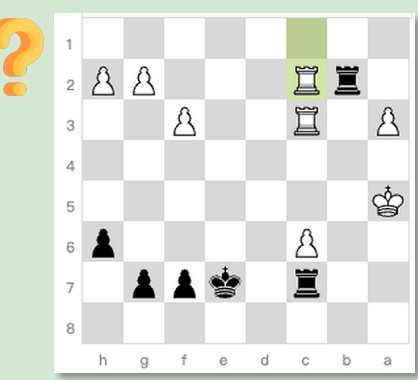

**Question:** "Mate in One ": You are given a chess position where it is **Black's turn to move**. Your task is to make a move that results in **checkmate** in one move.
## Allowed Moves
- Only legal moves are allowed, meaning moves that do not violate the standard rules of chess.
- You must only provide a single move that results in checkmate.
- The move can involve any piece (king, queen, rook, bishop, knight, or pawn).

## Notes
- The move you provide must **immediately checkmate** the opponent's king.
- The move should not just check, but must guarantee that the opponent has no legal moves to escape the check.
Please provide your move in the format of **standard algebraic notation**:
In chess, moves are written using standard algebraic notation, which includes the following components: **notation of piece moved – destination square**
- **Piece**: The piece is represented by a capital letter:
  - **K** for King
  - **Q** for Queen
  - **R** for Rook
  - **B** for Bishop
  - **N** for Knight
  - **P** for Pawn (No letter initial is used for pawns, so e4 means "pawn moves to e4".)
- **Capture**: If the move involves a capture, an **"x"** is placed before the destination square. For example, "Qxh7" means the queen captures on h7.
- **Checkmate**: checkmate is indicated by a **"#"** (e.g., "Qh7#").

Reasoning type: Abductive        Difficulty level: Medium

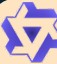 **Qwen2.5-VL-72B-Instruct**

The move that results in checkmate in one move is:
**Rxc2#**
This move involves the white rook on c3 capturing the black pawn on c2, delivering checkmate to the black king. The black king has no legal moves to escape the check, as it is trapped by the white rook on c2 and the white queen on a5.

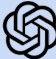 **o4-mini**

The one-and-only winning move is  Rc1#
The black rook simply drops down to c1, delivering immediate mate.

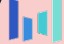 **Seed1.5-VL-Thinking**

Ra7#

*Figure 46.* An example of a Mate-on-one problem, in which the model needs to try, judge, reflect, and reason to find the right answer.

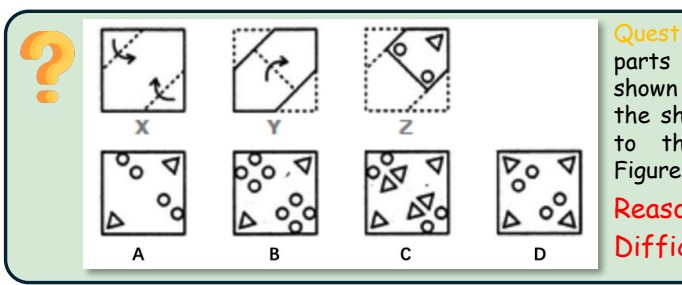

**Question:** Fold and subtract parts along the dotted lines shown in the figure, and choose the shape that is most similar to the unfolded shape of Figure Z among A, B, C, and D.

**Reasoning type:** Deductive

**Difficulty level:** Medium

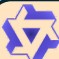 **Qwen2.5-VL-72B-Instruct**

To determine the correct answer, we need to follow the folding and subtraction steps as indicated in the figures X and Y, and then compare the resulting shape to the options A, B, C, and D.
1. **Figure X:**
   - The square is folded along the diagonal dotted line.
   - The part of the square that is outside the folded triangle is subtracted.
   - After unfolding, the resulting shape will have a triangular cutout along the diagonal.
2. **Figure Y:**
   - The square is folded along the vertical dotted line.
   - The part of the square that is outside the folded rectangle is subtracted.
   - After unfolding, the resulting shape will have a rectangular cutout along the vertical line.
3. **Combining the results from X and Y:**
   - The final shape will have both a triangular cutout along the diagonal and a rectangular cutout along the vertical line.
4. **Comparing with the options:**
   - Option A: This shape has a triangular cutout along the diagonal but no rectangular cutout along the vertical line.
   - Option B: This shape has a triangular cutout along the diagonal and a rectangular cutout along the vertical line, matching the combined result from X and Y.
   - Option C: This shape has a triangular cutout along the diagonal but no rectangular cutout along the vertical line.
   - Option D: This shape has a triangular cutout along the diagonal but no rectangular cutout along the vertical line.
The shape that is most similar to the unfolded shape of Figure Z is Option B.
The correct option is \boxed{B}.}$ 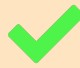

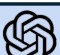 **o4-mini**

The correct unfolding is choice A. 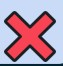

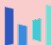 **Seed1.5-VL-Thinking**

D. 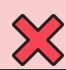

*Figure 47.* An example of a paper-cut problem, in which the model needs to understand spatial relationships and reason to find the correct answer.

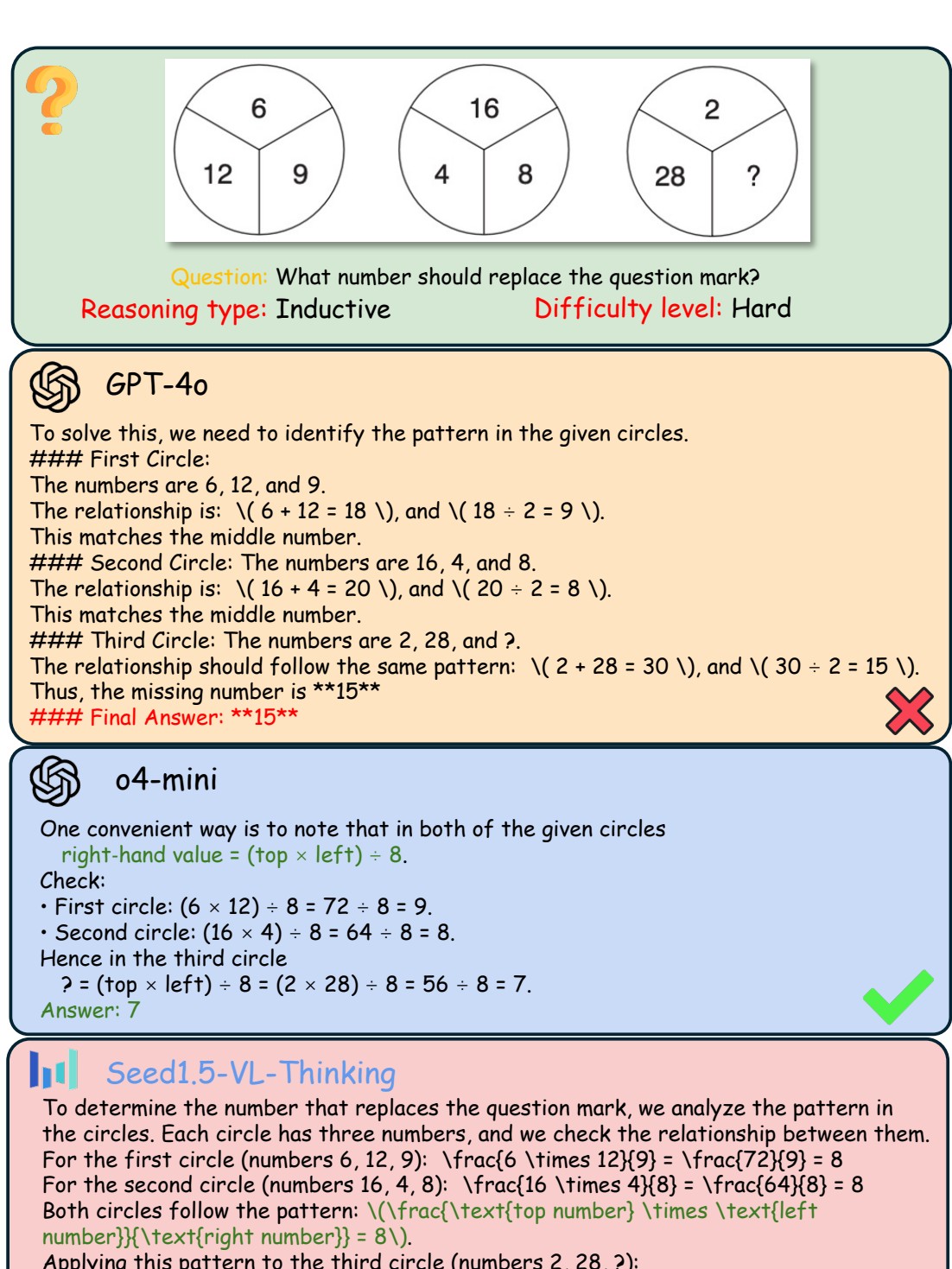

**Question:** What number should replace the question mark?
**Reasoning type:** Inductive          **Difficulty level:** Hard

### GPT-4o

To solve this, we need to identify the pattern in the given circles.
### First Circle:
The numbers are 6, 12, and 9.
The relationship is: $( 6 + 12 = 18 )$, and $( 18 \div 2 = 9 )$.
This matches the middle number.
### Second Circle: The numbers are 16, 4, and 8.
The relationship is: $( 16 + 4 = 20 )$, and $( 20 \div 2 = 8 )$.
This matches the middle number.
### Third Circle: The numbers are 2, 28, and ?.
The relationship should follow the same pattern: $( 2 + 28 = 30 )$, and $( 30 \div 2 = 15 )$.
Thus, the missing number is **15**
### Final Answer: **15**

### o4-mini

One convenient way is to note that in both of the given circles
   right-hand value = (top × left) ÷ 8.
Check:
• First circle: (6 × 12) ÷ 8 = 72 ÷ 8 = 9.
• Second circle: (16 × 4) ÷ 8 = 64 ÷ 8 = 8.
Hence in the third circle
   ? = (top × left) ÷ 8 = (2 × 28) ÷ 8 = 56 ÷ 8 = 7.
Answer: 7

### Seed1.5-VL-Thinking

To determine the number that replaces the question mark, we analyze the pattern in the circles. Each circle has three numbers, and we check the relationship between them.
For the first circle (numbers 6, 12, 9): $\frac{6 \times 12}{9} = \frac{72}{9} = 8$
For the second circle (numbers 16, 4, 8): $\frac{16 \times 4}{8} = \frac{64}{8} = 8$
Both circles follow the pattern: $(\frac{\text{top number} \times \text{left number}}{\text{right number}} = 8)$.
Applying this pattern to the third circle (numbers 2, 28, ?):
Let the missing number be $(x)$. Then: $\frac{2 \times 28}{x} = 8$
Solving for $(x)$: $\frac{56}{x} = 8 \implies x = \frac{56}{8} = 7$
7

*Figure 48.* An example of a number pattern game, in which the model needs to calculate and find the pattern of the numbers, and finally reason out the correct answer.

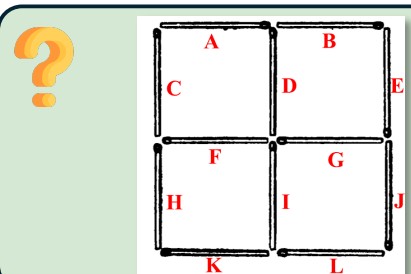

**Question:** Remove 2 matches leaving 2 squares of different sizes. Please answer the removed match id and give one possible answer.

**Reasoning type:** Abductive

**Difficulty level:** Medium

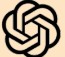 GPT-4o

To leave two squares of different sizes by removing two matches, one possible solution is:
### Remove matches: - **B** - **I**
### Result:
- A large square remains in the top-left corner (formed by matches A, C, F, H, and K).
- A smaller square remains in the bottom-right corner (formed by matches G, J, L, and E).

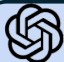 o4-mini

One way to see it is to notice that if you keep the entire outer "frame" (A, B, C, E, H, J, K, L) you get a big 2×2 square, and if you keep the bottom-right little cell (G, I, J, L) you get a 1×1 square. All the interior edges are D (the top-half of the middle vertical) and F (the left-half of the middle horizontal). So Remove: D and F
Remaining squares: Large 2×2 square → A, B, C, E, H, J, K, L  Small 1×1 square → G, I, J, L

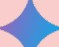 Gemini-2.5-Pro-Thinking

Okay, let's analyze the structure.
Initially, there are 5 squares:
1.  Small top-left (A, C, F, D)
2.  Small top-right (B, D, G, E)
3.  Small bottom-left (F, H, K, I)
4.  Small bottom-right (G, I, L, J)
5.  Large outer square (A, B, E, J, L, K, H, C)
We need to remove 2 matches to leave exactly 2 squares of different sizes. This means we need to keep the large outer square and one of the small squares.
To keep the large square, we must *not* remove any of the outer matches (A, B, C, E, H, J, K, L).
Therefore, the matches we remove must be from the inner set: D, F, G, I.
If we remove two adjacent inner matches (like D and F, or D and G, or F and I, or G and I), we break three small squares but leave one small square and the large square intact.
Let's try removing **D and F**:
*   Removing D breaks the top-left and top-right small squares.
*   Removing F breaks the top-left and bottom-left small squares.
*   The top-left square is broken twice. The top-right and bottom-left are broken once.
*   The bottom-right square (G, I, L, J) remains intact.
*   The large outer square (A, B, E, J, L, K, H, C) remains intact.
*   We are left with the large square and the small bottom-right square. These are 2 squares of different sizes.
**Answer:**
Removed match IDs: **D and F**

*Figure 49.* An example of a moving-match problem, in which the model needs to try, judge, reflect, and reason to find the right answer.

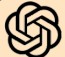 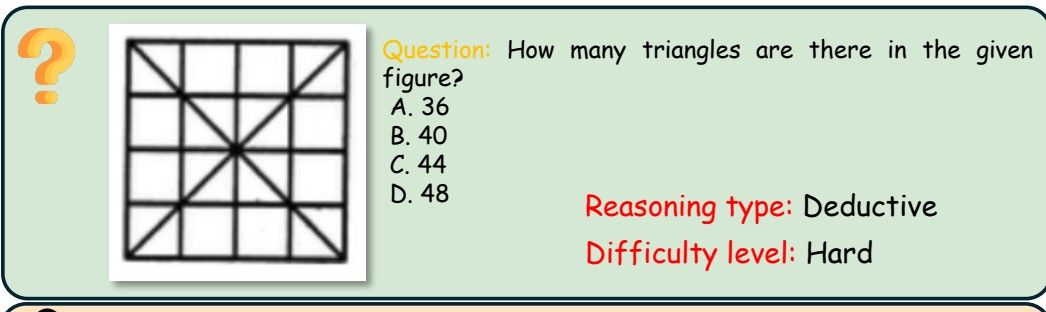

**Question:** How many triangles are there in the given figure?
  A. 36
  B. 40
  C. 44
  D. 48

**Reasoning type:** Deductive

**Difficulty level:** Hard

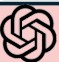 **GPT-4o**

To determine the total number of triangles in the given figure, we need to count all possible triangles of different sizes.
### Step-by-step counting:
1. **Small triangles**:
   - The figure is divided into 16 small squares, and each square is further divided into 4 small triangles.
   - Total small triangles = \( 16 \times 4 = 64 \).
2. **Larger triangles formed by combining two small triangles**:
   - Each square contains 2 larger triangles formed by combining two small triangles.
   - Total larger triangles = \( 16 \times 2 = 32 \).
3. **Triangles formed by combining four small triangles**:
   - Each square contains 1 triangle formed by combining four small triangles.
   - Total triangles of this type = \( 16 \times 1 = 16 \).
4. **Triangles formed by combining multiple squares**:
   - Additional triangles are formed by combining triangles across multiple squares. These include triangles spanning rows, columns, and diagonals.
   - After careful analysis, the total number of such triangles is **12**.
### Total triangles:
Adding all the triangles together: 64 + 32 + 16 + 12 = 124
However, the options provided do not match this total. Please verify the figure or options.

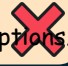 **o4-mini**

To see why the total is 48, it helps to break the triangles into two classes:
1. "Right-triangles" whose hypotenuse runs along one of the two big diagonals, and whose legs are horizontal & vertical grid-lines.
   Along each diagonal you can choose a segment of length 1, 2, 3 or 4 (in small-square-units). For a segment of length k there are exactly 2 such right-triangles (one on each "side" of the diagonal). Since each diagonal has
      k=1: 4 segments → 4·2 = 8 triangles
      k=2: 3 segments → 3·2 = 6
      k=3: 2 segments → 2·2 = 4
      k=4: 1 segment → 1·2 = 2
   that's 8+6+4+2 = 20 triangles on each diagonal, or 40 total so far.
2. "Isosceles" triangles whose two equal sides both lie on the diagonals and meet at the center (2,2), with a horizontal or vertical base.
   There are two sizes in each of the four compass-pointing directions (up, down, left, right), for 2·4 = 8 more.
Adding 40 + 8 gives 48. **Answer: D.**

*Figure 50.* An example of a counting problem, in which the model needs to logically arrange the reasoning steps and find the correct answer.

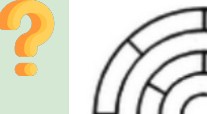

**Question:** What is the minimum number of different colours required to paint he given figure such that no two adjacent regions have the same colour?

A. 3
B. 4
C. 5
D. 6

**Reasoning type:** Deductive
**Difficulty level:** Medium

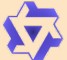 **Qwen2.5-VL-72B-Instruct**

The figure can be visualized as a graph where each region is a vertex and edges connect adjacent regions. The goal is to find the chromatic number of this graph, which is the smallest number of colors needed to color the vertices such that no two adjacent vertices share the same color.

Let's break down the figure:

1. The figure consists of concentric circles divided into segments. 2. The innermost circle is a single region. 3. The next circle is divided into 4 segments. 4. The outermost circle is divided into 8 segments.

We can label the regions as follows:

- The innermost circle is region A. - The next circle is divided into regions B1, B2, B3, and B4. - The outermost circle is divided into regions C1, C2, C3, C4, C5, C6, C7, and C8.

Now, let's analyze the adjacency:

- Region A is adjacent to all 4 regions B1, B2, B3, and B4.
- Each region Bi (i=1,2,3,4) is adjacent to two regions Cj (j=1,2,...,8).

To ensure no two adjacent regions have the same color, we can use the following coloring strategy:

- Color region A with color 1. - Color regions B1, B2, B3, and B4 with colors 2, 3, 4, and 2 respectively (since B1 and B3 are not adjacent, and B2 and B4 are not adjacent).
- Color regions C1, C2, C3, C4, C5, C6, C7, and C8 with colors 3, 4, 2, 3, 4, 2, 3, and 4 respectively (alternating colors to ensure no two adjacent regions have the same color).

Therefore, the minimum number of different colors required is \boxed{4}.

The correct option is B. ❌

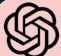 **o4-mini**

The key is to note that all the "wedges" (the radial slices) come in an even-length cycle at each of the three rings, plus the central disc. Label the colors $C_0, C_1, C_2$. One can do the following:

1. Color the very center $C_0$.
2. On the innermost ring of 6 (or 8) wedges you then cannot use $C_0$, but a cycle of even length can be 2-coloured, say alternating $C_1$ and $C_2$.
3. The next ring up: each wedge forbids just the color of the wedge immediately below, so each has two choices among $\{C_0, C_1, C_2\}$; again arranging these choices you can 2-colour that ring (using only $C_0$ plus one of $\{C_1, C_2\}$).
4. The outermost ring is handled the same way against the ring just inside it.

In this way only three colors are ever used, and one checks easily that no two adjacent regions clash. Hence 3 colours suffice, and obviously you need at least 3 (the inner cycle by itself already needs 2 plus the center needs a third). Answer: A ✅

*Figure 51.* An example of a coloring game, in which the model needs to plan reasonably and find the minimum number of colors needed.

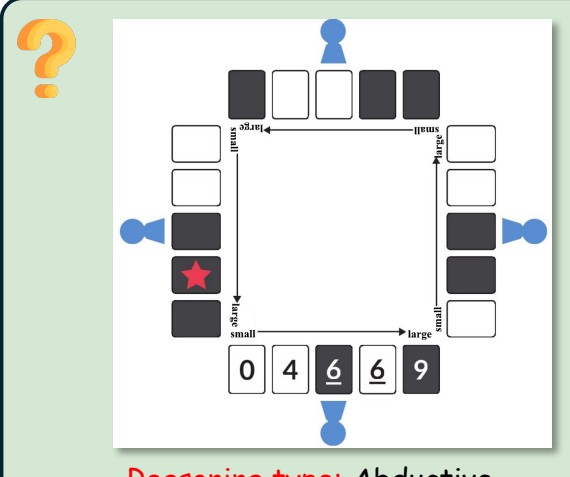

**Reasoning type:** Abductive

**Difficulty level:** Hard

**Question:** There are a total of 20 cards, divided into two types: black and white. Each type has 10 cards, numbered from 0 to 9. The rules of the game are as follows: (1) The player who gets the cards arranges them in front of themselves in ascending order from left to right. The cards must be placed face down. (2) If there are cards with the same number in both black and white, the black card is placed on the left, and the white card is placed on the right. This game involves guessing the numbers on the face-down cards placed in front of other players. As shown in the image, some card numbers are visible. The question is: what are the numbers on the cards marked by star?

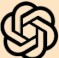 **GPT-4o**

### Rules Recap: 1. Cards are arranged in ascending order from left to right. 2. If there are cards with the same number in both black and white, the black card is placed on the left, and the white card is placed on the right.
### Observations: 1. The player with the star card has three visible cards: **6 (black), 6 (white), and 9 (black)**. 2. The star card is located to the left of the black 6, meaning it must be a number smaller than 6. 3. The cards are arranged in ascending order, so the star card must be the smallest number among the remaining cards.
### Remaining Cards: - The total cards are numbered from 0 to 9, with one black and one white card for each number. - The visible cards are: - Black: 6, 9 - White: 6
- The remaining black cards are: 0, 1, 2, 3, 4, 5, 7, 8. - The remaining white cards are: 0, 1, 2, 3, 4, 5, 7, 8, 9.
### Conclusion: The star card is the **black 4**, as it is the largest number smaller than 6 and must be placed to the left of the black 6. ### Final Answer: **4** ✗

**Seed1.5-VL-Thinking**

1. **Card Basics**: There are 20 cards (10 black, 10 white), each numbered 0–9. Each number has one black and one white card. Players arrange their 5 cards in ascending order (left to right), with black cards placed left of white cards if numbers are identical.
2. **Analyze Visible Cards**:    - The bottom player's cards are $( 0 )$ (white), $( 4 )$ (white) (white), $( 6 )$ (black), $( 6 )$ (white), $( 9 )$ (black). Thus:  - White cards used: $( 0, 4, 6 )$.  - Black cards used: $( 6, 9 )$.
3. **Determine Available Numbers**: White cards remaining: $( 1, 2, 3, 5, 7, 8 )$ .Black cards remaining: $( 0, 1, 2, 3, 4, 5, 7, 8 )$.
4. **Left Player's Card Order**:  The left player's cards are arranged in ascending order (left to right). The star marks a black card (dark color). The first two cards of the left player are white (light color), so their values must be the smallest available white numbers: $( 1 )$ and $( 2 )$ (since $( 0 )$'s white card is already used by the bottom player).
   The star-marked black card must be the smallest black number **greater than the second white card ($( 2 )$)**. The smallest available black number greater than $( 2 )$ is $( 3 )$.  Thus, the number on the star-marked card is $( 3 )$. ✗

*Figure 52.* An example of a reasoning problem, in which the model needs to make assumptions, verify them, reflect, and reason to arrive at the correct answer.

