# OpenReview forum: "MME-Reasoning: A Broad-Spectrum Benchmark for Evaluating Logical Reasoning in MLLMs"
_ICML.cc/2026/Conference — ICML 2026 regular_

### Official Review · Reviewer_Dp3d · 2026-03-06

**Soundness:** 3
**Presentation:** 3
**Significance:** 3
**Originality:** 3
**Overall Recommendation:** 5
**Confidence:** 3

**Summary:**

Most existing multimodal reasoning benchmarks lack a clear categorization of logical reasoning types and tend to conflate "visual perception capabilities" or "complex domain knowledge" with genuine "logical reasoning abilities." To address these issues, the research team constructed the MME-Reasoning benchmark, which contains over 1,000 carefully curated questions comprehensively covering three classic logical reasoning types: Deductive, Inductive, and Abductive reasoning.

**Compliance With Llm Reviewing Policy:**

Affirmed.

**Final Justification:**

My concerns have been adequately addressed. I improved my score.

**Key Questions For Authors:**

Same as weaknesses.

**Limitations:**

No, need to discuess.

**Strengths And Weaknesses:**

### Strengths

- Unlike previous benchmarks that primarily focus on deductive and inductive reasoning, MME-Reasoning systematically incorporates abductive reasoning into its evaluation framework, thereby measuring the models' logical capabilities more comprehensively.

- To ensure the purity of the evaluation, the benchmark excludes questions that can be answered solely through simple visual recognition. Meanwhile, it limits the domain knowledge threshold to the K-12 level or below, preventing models from being penalized for lacking specific advanced professional knowledge.

- The paper not only provides a leaderboard but also delves into the practical utility and cost-effectiveness of the currently trending "Thinking Models" and Rule-based Reinforcement Learning in multimodal reasoning.

### Weaknesses

- The paper rigidly and exclusively categorizes all questions as "Deductive," "Inductive," or "Abductive." However, in real-world complex reasoning, these three types are often intertwined. Therefore, could this forced categorization approach introduce bias into the evaluation of model capabilities?

- The evaluation data might have already been extensively scraped and is highly likely to exist in the pre-training corpora of major closed- and open-source models (e.g., GPT, Qwen). How do the authors prevent this issue, or how can they prove that no data leakage/contamination has occurred?

- With the emergence of recent "thinking with image" technologies, have the deficiencies in model capabilities highlighted by this benchmark already been resolved?

- For newer models, such as Gemini 3-Pro, has the performance already reached a level comparable to that of humans?

- The authors should supplement the evaluation results of fully open-source models, like Bee [1], which would help illustrate the performance gaps among different types of models within the open-source community.

[1] Zhang, Y., Ni, B., Chen, X. S., Zhang, H. R., Rao, Y., Peng, H., ... & Hu, S. M. (2025). Bee: A High-Quality Corpus and Full-Stack Suite to Unlock Advanced Fully Open MLLMs. arXiv preprint arXiv:2510.13795.

---

> ### Author Rebuttal · Authors · 2026-03-31
>
> **Q1: Does the strict Deductive/Inductive/Abductive categorization introduce evaluation bias since real-world reasoning is often intertwined?**
>
> **A1**: 1. Theoretical Basis: Our categorization is grounded in the classical syllogism framework (Charles Sanders Peirce's classification), which structures reasoning around three components: Premises, Rules, and Conclusions (Deductive: Premises + Rules -> Conclusion; Inductive: Premises + Conclusion -> Rules; Abductive: Rules + Conclusion -> Premises).
> 2. Rigorous Annotation Process: To ensure that this categorization does not introduce subjective bias, we implemented a strict, multi-annotator consensus mechanism rather than relying on a single individual's judgment:
>    - Solution to Disagreements: Each problem was independently evaluated by at least two annotators. Based on our annotation principles (Appendix C.3), annotators were instructed to identify the reasoning type according to the theoretical basis. If the first two annotators assigned different reasoning types to the same question, the problem was forwarded to a third annotator for a final decision. The final reasoning type was only assigned if at least two out of the three annotators reached an agreement.
>    - Disagreement Rate:  The initial agreement rate was 78.6%. For the remaining disagreements, a third annotator was introduced. Notably, no single question received three different labels, proving that the boundaries between reasoning types are clear enough for human experts to reach a strict majority consensus.
>
> **Q2: How do you ensure there is no data contamination/leakage from the pre-training corpora of major models?**
>
> **A2**: We address this through three aspects:
> 1. New & Synthetic Data: 84.85% of the benchmark is newly collected. Crucially, many logic puzzles and spatial tasks are procedurally generated via custom scripts or manually redesigned, making them completely absent from historical corpora.
> 2. Empirical Evidence: SoTA models still struggle massively compared to human experts (83.4%). If data were leaked, model accuracy would be unusually high.
> 3. No Memorization Patterns: Our CoT analysis (case studies provided in Appendix F) found no instances of "memorization symptoms" (*i.e.*, flawed/nonsensical reasoning processes that magically output the exact correct answer).
>
> **Q3: Have recent "thinking with image" technologies resolved the reasoning deficiencies highlighted by this benchmark?**
>
> **A3**: We would like to address this from the following perspectives:
> 1. Current "thinking with image" primarily enhances perception and domain-specific tasks: Recent visual thinking technologies focus on fine-grained perception (e.g., dynamic zooming) or specific image manipulations (e.g., drawing auxiliary lines). While helpful for math-heavy tasks, they address only a fraction of the comprehensive reasoning spectrum.
> 2. MME-Reasoning evaluates a much broader spectrum of logical reasoning: Unlike perception-centric benchmarks, MME-Reasoning comprehensively evaluates deductive, inductive, and abductive reasoning across diverse scenarios. Enhancing perception does not inherently improve performance in abstract pattern discovery or hypothesis formulation.
>
> Therefore, MME-Reasoning remains a highly challenging and necessary testbed to guide the next generation of multimodal models toward true, general-purpose logical reasoning.
>
> **Q4: Have the latest models reached human-level performance?**
>
> **A4:** We evaluated the newest models (Gemini-3.1-Pro, GPT-5.4, Claude-4.6-Sonnet):
> - Gap Remains: While Gemini-3.1-Pro establishes a new SoTA (68.3%), it still falls significantly short of Human Experts (83.4%).
> - Bottlenecks: Newer models show a more balanced cognitive profile, but Abductive reasoning (forming hypotheses) and Spatial & Temporal (S&T) reasoning remain major weaknesses across all models.
>
> |Model|AVG.|CAL.|P&E.|PA.|S&T.|CCA.|DED.|IND.|ABD.|
> |-|-|-|-|-|-|-|-|-|-|
> |Human Expert|83.4|84.4|84.9|80.3|88.1|85.8|85.8|76.9|85.6|
> |Gemini-3.1-Pro-T|68.3|77.3|77.2|72.5|54.8|84.1|69.0|65.1|69.4|
> |GPT-5.4|52.2|62.6|47.5|50.0|48.9|62.5|61.0|45.0|48.0|
> |Claude-4.6-Sonnet|51.2|59.3|53.9|49.7|43.3|74.3|54.5|43.8|53.5|
> |Bee-8B-SFT|37.0|46.5|27.1|32.1|34.0|45.1|46.8|36.3|26.1|
> |Bee-8B-RL|38.2|47.4|26.0|34.8|36.5|43.1|48.6|38.7|25.9|
>
> **Q5: Please supplement evaluation results for fully open-source models like Bee.**
>
> **A5:** We evaluated Bee-8B (RL and SFT versions) and will add the results to the revised manuscript. Key findings include:
> 1. Universal Abductive Bottleneck: Like closed models, Bee models show a massive performance drop in Abductive reasoning compared to Deductive reasoning (a gap of over 20 points).
> 2. RL > SFT: The RL version (38.2%) slightly outperforms the SFT version (37.0%), showing RL effectively enhances multimodal logical reasoning.
> 3. Weakness in Exploration: Both models struggle significantly in open-ended "Planning & Exploring" scenarios (scoring ~26-27%).

---

> > ### Author Rebuttal · Reviewer_Dp3d · 2026-04-02
> >
> > My concerns have been addressed.

---

### Official Review · Reviewer_d6ds · 2026-03-12

**Soundness:** 2
**Presentation:** 3
**Significance:** 3
**Originality:** 3
**Overall Recommendation:** 4
**Confidence:** 3

**Summary:**

The paper introduced a novel benchmarks focusing on the evaluation of the logic reasoning ability of multi-modal large language models. The proposed benchmarks introduced three types of reasoning (i.e., inductive, deductive, and abductive) from multiple source. The benchmark includes questions with varying levels of difficulty, as well as diverse types and disciplines. The evaluation covers a wide range of both open-source and closed-source models.

**Compliance With Llm Reviewing Policy:**

Affirmed.

**Final Justification:**

Most of the concerns have been addressed. I would maintain my positive score.

**Key Questions For Authors:**

1. The evaluation heavily relies on ChatGPT for answer extraction and verification, which may limit the practical usability of the benchmark. Could the authors consider alternative evaluation methods that do not depend on external tools? For example, in open-ended reasoning tasks for LLMs (e.g., math benchmarks), it is common practice to instruct models to present the final answer in the format \\box{ans}. This allows the answer to be easily extracted from the output, thereby simplifying the evaluation process.

**Limitations:**

Yes

**Strengths And Weaknesses:**

Strengths:

1. Compared to prior counterpart benchmarks, the proposed benchmarks covers all three types of reasoning (i.e., inductive, deductive, and abductive).

2. The questions in the benchmarks are sampled from multiple sources.

3. The proposed benchmarks covers multiple disciplines.

4. The evaluation is comprehensive.

Weaknesses:
1. It seems that the evaluation process heavily requires ChatGPT to extract and verify the answers, which would increase the cost for practical use.

2. The questions in the benchmarks are divided into easy, medium and hard problems. But the paper doesn't provide the details on the partitiion of the difficulties.

3. When evaluating whether the benchmark is reasoning-centric, the paper does not provide clear details about the criteria used to determine whether a question requires reasoning.

---

> ### Author Rebuttal · Authors · 2026-03-31
>
> **Q1: Relying on ChatGPT for evaluation increases practical costs.**
>
> **A1**: We sincerely thank the reviewer for the question.
> 1. LLM-based evaluation is widely adopted in complex multimodal benchmarks (e.g., MathVista) to handle diverse, open-ended outputs.
> 2. Highly Affordable: While our evaluation pipeline relies on an LLM judge, the actual financial cost is highly optimized. We specifically employ `GPT-4o-mini` as the extractor and judge. Due to its high efficiency and low pricing, the total cost for evaluating a model on the entire MME-Reasoning benchmark is less than $1 per run (extracting and judging the response of Qwen2.5-VL-72B costs ~0.1).
> 3. Open-source Alternative: For users who prefer to eliminate API costs or evaluate in offline environments, our framework is flexible and supports locally deployed open-source models as the judge. To validate this alternative, we evaluated 5 MLLMs using GPT-4o-mini and the open-source Qwen3.5-35B-A3B as judges. The negligible score difference (within ±0.3 points) proves that researchers can reliably use local open-source models for accurate, zero-cost evaluations. We will add this cost analysis and open-source instructions to the revised appendix.
>
> **Q2: Missing details on the partition of difficulty levels (easy, medium, hard).**
>
> **A2**: The detailed criteria are provided in **Appendix C.2**, based on the cognitive load and time required for a human expert:
> - Easy: The question typically has a straightforward and quick solution that can be correctly answered by a human expert within 2 minutes.
> - Medium: The question generally requires some reasoning steps and one to two rounds of trial and reflection, and can be correctly answered by a human expert within 2 to 5 minutes.
> - Hard: The question usually requires more than two attempts and reflections, or involves the use of tools such as auxiliary lines or drafts to support the thought process. It may or may not be solved by a human expert within 10 minutes.
>
> We will add a clear pointer to Appendix C.2 in the main text to prevent future confusion.
>
> **Q3: Unclear criteria for determining whether a question requires reasoning.**
>
> **A3**: We sincerely thank the reviewer for this comment.
>
> 1. In our benchmark, we define a question as requiring reasoning if the correct answer cannot be directly obtained through simple visual perception (*e.g.*, object recognition, OCR, basic counting) or direct factual knowledge retrieval. Instead, the question must necessitate a multi-step logical reasoning process.
>
> 2. Based on these criteria, to quantitatively validate that MME-Reasoning is primarily focused on reasoning and cannot be solved only based on perception, we conduct human and LLM-based evaluations (Tab. 4 in our paper). We enlisted six PhD-level experts, who independently classified questions from our benchmark. As shown in the table below, three separate human expert groups determined that 96.1%, 97.3%, and 95.8% of the questions in MME-Reasoning require reasoning.
>
>  We also used GPT-4o for the same classification task under the same criteria, which found that 94.4% of the questions require reasoning. These results, consistently showing that over 95% of the benchmark requires reasoning, provide strong evidence that MME-Reasoning is indeed a reasoning-centric benchmark, especially when compared to another benchmark like MathVista, which scored significantly lower in the same evaluation.
>
> | | Human Group 1 | Human Group 2 | Human Group 3 | GPT-4o |
> |-| -| - | - | - |
> | MathVista | 55.7% | 65.3% | 52.2% | 60.7% |
> | MME-Reasoning | 96.1% | 97.3% | 95.8% | 94.4% |
>
> **Q4: Can you use rule-based extraction (e.g., instructing models to use `\box{ans}`) instead of ChatGPT?**
>
> **A4**: We sincerely thank the reviewer for this constructive suggestion.
>
> 1. Using LLMs for answer extraction and verification has become a widely adopted method in recent multimodal reasoning benchmarks (*e.g.*, MathVista). Unlike pure math problems, MME-Reasoning contains diverse question types. These tasks often require structured outputs (*e.g.*, matrices or coordinate lists) that are difficult to perfectly constrain or parse using a simple `\box{ans}` rule.
> 2. Low Cost: The practical cost of using an API like GPT-4o-mini is extremely low and highly manageable for researchers. As detailed in Appendix D.1, extracting and judging the entire set of responses for a model (e.g., Qwen2.5-VL-72B) across our benchmark costs only around $0.1.
> 3. Viability of Open-Source Alternatives: Open-source LLMs (such as the Qwen3.5 series) also demonstrate excellent performance in instruction following and answer extraction.
> 4. Limitations of Rule-Based Matching: Rule-based methods are brittle for open-ended reasoning, often causing false negatives when answers are semantically correct but structurally varied [A].
>
> [A] Huang Y, et al. From Accuracy to Robustness: A Study of Rule-and Model-based Verifiers in Mathematical Reasoning.

---

> > ### Author Rebuttal · Reviewer_d6ds · 2026-04-04
> >
> > Thanks for the rebuttal. My concerns have been addressed. I would maintain my score.

---

### Official Review · Reviewer_KxwX · 2026-03-13

**Soundness:** 2
**Presentation:** 3
**Significance:** 3
**Originality:** 3
**Overall Recommendation:** 4
**Confidence:** 4

**Summary:**

This paper introduces MME-Reasoning, a benchmark for evaluating logical reasoning in multimodal large language models (MLLMs). The central contribution is coverage of all three classical reasoning types, inductive, deductive, and abductive, whereas prior benchmarks largely neglect abductive reasoning. The benchmark contains ~1,200 carefully curated questions spanning multiple question formats (multiple-choice, free-form, and rule-based), annotated for reasoning type, difficulty, and required capability. The authors evaluate a broad set of models and find that even state-of-the-art MLLMs fall well short of human expert performance, with abductive reasoning emerging as a consistent bottleneck. Additional analyses examine token efficiency, the effect of "thinking mode," and rule-based RL training.

**Compliance With Llm Reviewing Policy:**

Affirmed.

**Final Justification:**

I raised my score from weak reject to weak accept. The construct validity and token efficiency concerns are adequately addressed. My remaining concern about the theoretical grounding of the abductive category is only partially resolved. The inter-annotator agreement shows the labeling criteria can be applied consistently, but this does not show whether those criteria correctly identify abductive reasoning as a logical form rather than a cognitive solving strategy. I encourage the authors to acknowledge this distinction explicitly in the revision. The benchmark makes a good contribution by systematically evaluating all three classical reasoning types, and the finding that abductive reasoning is a consistent bottleneck across model families is a valuable result for the community.

**Key Questions For Authors:**

How was inter-annotator agreement handled specifically for the reasoning-type labels (deductive/inductive/abductive)? Given that the abductive category appears to be defined partly by solving strategy rather than logical form, what was the disagreement rate on this dimension, and how were disagreements resolved?

The paper argues that rule-based RL models fail to generalize beyond math-heavy training, but it is also possible that smaller-scale RL models are simply weaker overall. How do you disentangle the effect of training distribution from the effect of model capacity when interpreting this finding?

Several of the benchmark's "abductive" tasks have unique correct answers (e.g., Sudoku has exactly one solution). If abductive reasoning is defined as involving a large hypothesis space and exploratory verification, does a task with a unique, verifiable answer truly require abductive reasoning in any meaningful sense, or is this more accurately characterized as constraint satisfaction?

**Limitations:**

Yes

**Strengths And Weaknesses:**

Strengths

The paper's most valuable contribution is its inclusion of abductive reasoning as a first-class evaluation target. This is a genuine gap in the existing benchmark landscape, and the authors make a credible case both theoretically (via Peirce's tripartite classification) and empirically (via the consistent performance drop on abductive tasks across model families) that this omission has been masking an important limitation of current MLLMs.

The diversity of question formats is also commendable. Rule-based evaluation for puzzles like Sudoku and Hashi is non-trivial to implement and makes it possible to evaluate question types that resist multiple-choice framing. The evaluation pipeline, including the use of GPT-based answer extraction with structured intermediate formats, is described in sufficient detail to be reproducible.

The breadth of model coverage is a practical strength. Evaluating chat models, thinking models, and rule-based RL models in parallel allows the paper to make comparisons that are genuinely informative to the field, particularly the finding that rule-based RL training on math-heavy data does not reliably generalize to broader reasoning tasks.

Weaknesses

The definition of abductive reasoning is under-theorized. The paper's central claim rests on distinguishing abductive from deductive and inductive reasoning, but the operationalization of this distinction in the annotation process is not fully convincing. The appendix notes that Sudoku and mate-in-one chess problems are labeled abductive because humans tend to use hypothesis-verification strategies, but this is a claim about human cognitive process, not about the logical structure of the problem. A skilled solver can approach Sudoku deductively. Without a clearer formal criterion that is independent of solving strategy, the reasoning-type labels risk conflating task structure with one particular solving heuristic, which weakens the interpretive value of performance breakdowns by reasoning type.

The curation methodology raises questions about construct validity. The authors state that questions requiring "complex domain knowledge" were excluded, limiting domain expertise to K12 level. This is a reasonable goal, but the boundary between domain knowledge and reasoning is notoriously difficult to draw, and the paper does not provide enough detail about disagreement rates among annotators or how borderline cases were resolved. Without this, it is hard to assess whether the benchmark is actually measuring reasoning ability independent of knowledge, or whether it is doing so inconsistently across question subtypes.

The analysis of token efficiency and reasoning length is suggestive but not conclusive. The observation that longer outputs correlate with better performance is interesting, but the paper does not adequately disentangle whether longer reasoning chains are causally beneficial or whether harder questions simply elicit both more tokens and more errors, with the correlation being driven by question difficulty as a confounder. The diminishing-returns claim is similarly informal. A cleaner analysis controlling for difficulty level within the model would strengthen these conclusions considerably.

---

> ### Author Rebuttal · Authors · 2026-03-31
>
> **Q1: Labeling tasks like Sudoku as "abductive" conflates task structure with human solving heuristics rather than formal logical criteria.**
>
> **A1**:  We would like to clarify in the following two aspects:
> 1. We agree that for simple instances of these puzzles, pure deductive heuristics are sufficient. However, as the difficulty increases, the search space grows exponentially. For harder problems, relying solely on forward deductive rules becomes highly inefficient. To navigate this open-ended search space, both humans and MLLMs are practically forced to adopt abductive strategies.
> 2. Our benchmark is specifically designed to evaluate the reasoning capabilities of MLLMs. Through observation of the reasoning patterns (thinking models with long CoT, shown in case studies), we found that the pattern models employed to solve these problems are rarely a straightforward deductive chain. Instead, they heavily rely on a **hypothesis$\rightarrow$verification $\rightarrow$reflection** paradigm. This process of exploration aligns precisely with the core characteristics of abductive reasoning.
>
> Therefore, classifying these tasks as abductive is the result of the consideration of both theoretical analysis and empirical phenomena.
>
> **Q2: The disagreement rates and boundary of domain knowledge**
>
> **A2**: 1. Quantifying the Reliance on Domain Knowledge: We conducted experiments to quantify the amount of knowledge required (Tab. 5 in our paper). Following VisualPuzzles, we prompted GPT-4o to generate knowledge concept checklists for each question. The results show MME-Reasoning requires only 1.4 knowledge concepts per question on average (vs. 3.9 in MMMU), effectively isolating reasoning from heavy factual knowledge.
> 2. Annotation Process and Disagreement Rates: We implement a rigorous multi-stage review pipeline to ensure consistency across all question subtypes:
>   - Strict Guidelines:  To minimize subjectivity, explicitly permissible K-12 knowledge concepts were listed as a reference for the annotators.
>   - Review Pipeline: Questions were reviewed by two annotators, with a third resolving ties.
>   - Agreement Rate: The initial agreement rate between the first two annotators reached 91%, indicating a high level of consistency in our curation process.
>
> We will add these details to the revised version.
>
> **Q3: The analysis of token efficiency lacks control for question difficulty**
>
> **A3**:
> - Disentangling Factors: Controlling for difficulty, long-thinking models consistently outperform others (Fig. 4). We clarify that raw token length does not cause better performance; rather, effective reasoning patterns (exploration and reflection) naturally generate more tokens. It is the activation of these reasoning patterns that effectively solves complex logical problems.
> - Diminishing Returns: Our semi-log plot (Fig. 6) demonstrates that performance gains scale logarithmically with token expenditure. Our primary motivation for highlighting this phenomenon is to draw the community's attention to token utilization efficiency. While long-output models achieve superior results, they sometimes exhibit overthinking or repetitive reflection loops (Section 4.5) that consume massive compute without yielding proportional performance gains.
>
> **Q4: Inter-annotator disagreement rate for reasoning-type labels**
>
> **A4:**
> - Each problem was evaluated by two annotators. Disagreements were resolved by a third annotator, requiring a majority consensus.
> - Disagreement Rate: The initial agreement rate was 78.6%. The remaining 21.4% were resolved by the third annotator. Notably, no single problem received three different labels, indicating sufficiently clear boundaries.
>
> **Q5: The failure of rule-based RL models to generalize beyond math might be due to smaller model capacity rather than training distribution.**
>
> **A5:**
> We do not claim that rule-based RL inherently fails beyond math. Rather, we observe that *existing* open-source RLVR models primarily focus on math and struggle with broad-spectrum tasks. We fully agree that limited performance is a combined result of capacity, data, and methodology (Sec. 4.3). Our goal is to encourage the community to scale up RLVR in both model capacity and task diversity.
>
> **Q6: Do tasks with unique verifiable answers truly require abductive reasoning?**
>
> **A6:**
> While the final solution is unique, the solving process heavily relies on an abductive paradigm: formulating local hypotheses, conducting exploratory verification against constraints, and reflecting upon contradictions. This "hypothesize-verify-reflection" mechanism is a typical characteristic of abductive reasoning. We included such tasks to assess models' ability to execute this fundamental iterative process under strict and verifiable constraints. If a model is currently unable to sustain this reasoning loop, it is highly likely to struggle when confronted with the more complex and open-ended abductive reasoning tasks encountered in the real world.

---

> > ### Author Rebuttal · Reviewer_KxwX · 2026-04-03
> >
> > Thank you for the detailed reply. The responses on construct validity (Q2) and the difficulty-controlled token analysis (Q3) are satisfactory, the inter-annotator agreement rate and the knowledge concept analysis address my concerns, and Figure 4 does provide some difficulty-stratified evidence for the token efficiency claim.
> >
> > My remaining concern is Q1/Q4: the classification of tasks like Sudoku as abductive based on observed solving behavior rather than logical form. The authors argue this is grounded in empirical observation of model reasoning patterns, which I find partially convincing. However, high inter-annotator agreement shows the labeling criteria can be applied consistently, but does not show whether those criteria correctly capture abductive reasoning as a logical category rather than merely a cognitive solving strategy. I would encourage the authors to explicitly discuss this distinction in the revised paper.
> >
> > Given the overall contribution, I am willing to raise my score to weak accept.

---

### Official Review · Reviewer_KKG6 · 2026-03-14

**Soundness:** 3
**Presentation:** 3
**Significance:** 3
**Originality:** 3
**Overall Recommendation:** 4
**Confidence:** 4

**Summary:**

This paper introduces the MME-Reasoning evaluation benchmark, which systematically covers inductive, deductive, and abductive reasoning to assessing logical reasoning capabilities of multimodal large language models (MLLMs). Through carefully designed diverse questions, the study reveals significant challenges and imbalances in the logical performance of current state-of-the-art models.

**Compliance With Llm Reviewing Policy:**

Affirmed.

**Final Justification:**

My concerns have been adequately addressed. And I decide to give the weak accepted score.

**Key Questions For Authors:**

Please see the weakness.

**Limitations:**

Yes.

**Strengths And Weaknesses:**

Strengths:
1. The benchmark covers three reasoning types, with abductive reasoning included for the first time (as far as I know).
2. The question types are diverse, incorporating not only traditional multiple-choice questions, but also *free-form* and *rule-based* problems (such as Sudoku and 24-point).
3. The types of MLLMs selected for evaluation are comprehensive.
4. The scale of the dataset and the scope of the evaluation’s experimental work are both extensive and impressive.


Weaknesses:
1.    With only 1,188 examples, the dataset is relatively small. It would be helpful to describe methods for further scaling it.
2.    Manual annotation is relatively costly. Further exploration of automated annotation methods is recommended.
3.    Although the paper strives to cover a wide range of types, as it acknowledges that, due to the complexity of logical reasoning problems, it remains challenging to collect many real-world reasoning scenarios.

---

> ### Author Rebuttal · Authors · 2026-03-31
>
> **Q1: Data scale and further scaling method.**
>
> **A1**: We thank the reviewer for this feedback.
> 1. Comparison with Existing Reasoning Benchmarks: MME-Reasoning is comparable to or larger than recent benchmarks (*e.g.*, LogicVista has 0.4K). More importantly, it provides a much more comprehensive evaluation:
>     - Broader Reasoning Types: We incorporate abductive reasoning and diverse fine-grained subtypes, effectively making the scope of existing benchmarks a subset of ours.
>     - Diverse Formats: Beyond MCQs, we include free-form and rule-based questions to evaluate true problem-solving skills rather than simple option elimination.
>     - Diverse Domains: We comprehensively cover STEM, charts, multi-view images, temporal sequences, and logic games.
>
>     |Benchmark|Size|Question Types|Reasoning Types|Task Types|
>     |-|-|-|-|:-:|
>     |LogicVista|0.4K|MCQ|Inductive, Deductive|Spatial, Scientific, Puzzles|
>     |VisualPuzzles|1.2K|MCQ|Inductive|Puzzles (Quantitative, Spatial)|
>     |VisuLogic|1K|MCQ|Inductive|Puzzles (Quantitative, Spatial, Positional, Attribute, Stylistic, Others)|
>     |MME-Reasoning|1.2K|MCQ, Free-form, Rule-based|Inductive, Deductive, Abductive|Spatial, Temporal, Puzzle (Quantitative, Spatial, Positional, Attribute, Stylistic, Others), Geometry, Algebraic, Statistical, Scientific|
>
>    Our strict manual curation limits the initial scale but guarantees the absolute correctness required for a rigorous evaluation benchmark.
>
> 2. Methods for Further Scaling: We fully agree that outlining methods for further scaling is helpful for the community. We have established the following strategies to expand the dataset in the future:
>     - Programmatic Generation: For rule-based/synthetic tasks (*e.g.*, Sudoku), we use code-based pipelines to generate an unlimited number of instances with strictly controlled difficulty levels.
>     - Multi-Agent Extraction: For documents and textbooks, we will employ OCR/PDF parsing combined with a multi-agent system (including a judge agent) to automatically extract and filter high-quality questions at scale.
>
> **Q2: Exploration of automated annotation methods.**
>
> **A2**: We highly appreciate this insightful suggestion. To scale efficiently while maintaining consistency, one highly feasible approach is to employ an automated annotation pipeline driven by Multi-agent Systems, which mirrors our rigorous human protocol:
>   1. Rubric-Based Prompting: Formalize manual criteria (Difficulty, Reasoning Type, Capability) into strict rubrics for initial LLM annotation.
>   2. Multi-Agent Voting: To mitigate the hallucination and bias of a single LLM, a multi-agent system can be introduced, like a human annotation process: $N$ distinct LLM agents independently analyze the image and question, outputting their annotations and rationales based on the rubric. A summarizing agent then receives the annotations from all agents and determines the final answer based on the majority voting principle.
>   3. Human-in-the-Loop QA: Human annotators will intervene only when the voting yields no clear majority or the judge agent reports low confidence.
>
> In our future work, we will further expand the scale of the MME-Reasoning dataset to cover an even broader range of reasoning scenarios. Adopting such an automated or semi-automated annotation pipeline will be a crucial step in achieving this expansion efficiently while maintaining high consistency.
>
> **Q3: Challenging to collect many real-world reasoning scenarios.**
>
> **A3**: We sincerely thank the reviewer for this comment.  We would like to highlight that despite these challenges, MME-Reasoning already represents a significant step forward in terms of comprehensiveness. As shown in the table above and detailed in our appendix, it includes a much broader range of reasoning types, question formats, and task domains compared to existing benchmarks.
>
> However, despite our best efforts, we acknowledge that it is not feasible to cover every real-world reasoning task. There are two primary bottlenecks in benchmarking certain highly complex, real-world reasoning scenarios:
>
>   1. Data Collection Challenges: Gathering authentic, open-ended reasoning tasks along with definitive correct answers is exceedingly difficult. For instance, the logical reasoning involved in a judge's legal ruling is highly complex, nuanced, and extremely hard to collect and assign a single, absolute ground-truth answer to.
>   2. Evaluation Challenges: Many real-world reasoning tasks are inherently difficult to score objectively or programmatically using standardized protocols. For example, in a real scientific discovery scenario, evaluating the validity or brilliance of a novel research idea cannot be easily automated or assessed through current standardized evaluation frameworks.
>
> We appreciate your understanding of these inherent challenges and hope it will inspire future research to explore and formalize even more complex, real-world reasoning scenarios.

---

> > ### Author Rebuttal · Reviewer_KKG6 · 2026-04-03
> >
> > Thanks authors for their responses. My concerns have been adequately addressed.

---

### Decision · Program_Chairs · 2026-04-30

**Decision:**

Accept (regular)

**Comment:**

This submission proposes a new benchmark MME Reasoning for measuring different multimodal reasoning capabilities across deductive, inductive, and abductive settings.

Some concerns on the dataset building protocol is given to abductive setting. One reviewer pointed out that some tasks labeled as abductive could be better understood as constraint satisfaction problem (or possibly as abductive strategies than truly abductive logics). Reviewers also raised related concerns regarding construction validity and unwanted bias injected from classification.

Nonetheless, most reviewers agree that MME reasoning benchmark includes all three types of reasoning, covering a wide range of task formats evaluated by thinking models and RL-based variants. Most of the concerns were addressed during the author-reviewers discussion phase (e.g., dataset scale and additional annotation with agreement rates).

All in all, the proposed benchmark makes a reasonable contribution. We encourage the authors better frame abductive labels better distinguishing logical structure and observed solving behavior. Addressing additional concerns clarified during the rebuttal phase should be included in the main draft.